# Site-specific synergy in heterogeneous single atoms for efficient oxygen evolution

Peiyu Ma [1,5], Jiawei Xue [1,5], Ji Li[1,5], Heng Cao [1], Ruyang Wang[1], Ming Zuo[2], Zhirong Zhang [2] ✉ & Jun Bao [1,3,4] ✉

Heterogeneous single-atom systems demonstrate potential to break performance limitations of single-atom catalysts through synergy interactions. The synergy in heterogeneous single atoms strongly dependes on their anchoring sites. Herein, we reveal the site-specific synergy in heterogeneous single atoms for oxygen evolution. The $Ru_TIr_V/CoOOH$ is fabricated by anchoring Ru single atoms onto three-fold facial center cubic hollow sites and Ir single atoms onto oxygen vacancy sites on CoOOH. Moreover, $Ir_TRu_V/CoOOH$ is also prepared by switching the anchoring sites of single atoms. Electrochemical measurements demonstrate the $Ru_TIr_V/CoOOH$ exhibits enhanced OER performance compared to $Ir_TRu_V/CoOOH$. In-situ spectroscopic and mechanistic studies indicate that Ru single atoms at three-fold facial center cubic hollow sites serve as adsorption sites for key reaction intermediates, while Ir single atoms at oxygen vacancy sites stabilize the *OOH intermediates via hydrogen bonding interactions. This work discloses the correlation between the synergy in heterogeneous single atoms and their anchoring sites.

Single-atom catalysts (SACs) combining atomically dispersed metal centers and unique electronic structures have exhibited potential applications in various energy conversion reactions[1–7]. Nevertheless, the performances of the SACs remain unsatisfactory in many cases, especially for the multistep and multielectron reactions. Heterogeneous single-atom systems integrated the advantages of multiple metal species, demonstrating potential to break the performance limitations of SACs through the synergy interactions[8–10]. The interactions between heterogeneous single atoms could optimize single-atom electronic structures to enhance the catalytic activities[11–14]. In addition, heterogeneous single atoms can simultaneously serve as active sites, synergistically regulating the adsorption behaviors of intermediates[15–20]. Therefore, developing effective strategies to modulate the synergy interactions in heterogeneous single atoms is urgently required.

Selectively anchoring heterogeneous single atoms to diverse sites is an effective strategy to modulate their synergy interactions. The topology structures or defects on the surface of transition metal oxides provided a variety of single-atom anchoring sites. As a result of the different electronegativities between anchoring sites and single-atom precursors, heterogeneous single atoms can selectively anchor onto the diverse anchoring sites of transition metal oxides[21–23]. Differences in the anchoring sites result in diverse electronic structures and configurations of heterogeneous single atoms, differentiating their functions in the synergetic catalytic process. However, atomic-level insight into the site-specific synergy in heterogeneous single atoms is still lacking.

In this work, we provided an in-depth understanding of the site-specific synergy in heterogeneous single atoms for oxygen evolution reaction (OER). The heterogeneous single-atom catalyst $Ru_TIr_V/CoOOH$ was fabricated by selectively anchoring Ru single atoms onto three-fold facial center cubic (fcc) hollow sites of oxygens and Ir single atoms onto oxygen vacancy ($V_O$) sites. Moreover, $Ir_TRu_V/CoOOH$ was

[1]National Synchrotron Radiation Laboratory, University of Science and Technology of China, Hefei, Anhui, PR China. [2]Hefei National Research Center for Physical Sciences at the Microscale, University of Science and Technology of China, Hefei, Anhui, PR China. [3]Key Laboratory of Precision and Intelligent Chemistry, University of Science and Technology of China, Hefei, Anhui, PR China. [4]iChEM (Collaborative Innovation Center of Chemistry for Energy Materials), University of Science and Technology of China, Hefei, Anhui, PR China. [5]These authors contributed equally: Peiyu Ma, Jiawei Xue, Ji Li.
✉e-mail: zzhirong@ustc.edu.cn; baoj@ustc.edu.cn

also prepared by switching the anchoring sites of Ru and Ir single atoms. Electrochemical measurements demonstrated that the $Ru_T Ir_V$/CoOOH exhibited enhanced OER performance compared to $Ir_T Ru_V$/CoOOH. Specifically, the $Ru_T Ir_V$/CoOOH showed an overpotential of 180 mV at a current density of 10 mA cm$^{-2}$ for oxygen evolution, which expressively decreased compared with 270 mV of $Ir_T Ru_V$/CoOOH. A series of in-situ spectroscopic characterizations and mechanistic studies exhibited that the Ru single atoms at the three-fold fcc hollow sites serve as adsorption sites for key reaction intermediates. Meanwhile, Ir single atoms at $V_O$ sites stabilized the key reaction intermediates on the Ru single atoms via hydrogen bonding interactions. This work not only proposed a synthesis strategy for constructing heterogeneous single atoms but also deepened the understanding of the synergy interaction in heterogeneous single atoms at the atomic scale.

## Results

### Fabrication of heterogeneous single atoms at diverse sites

On account of the presence of multiple topologies and defects, the surface of transition metal oxides (e.g., CoOOH) features two kinds of anchoring sites, including three-fold fcc hollow sites of oxygen atoms and $V_O$ sites (Fig. 1a). The single-atom precursors formed metal cations ($Ru^{3+}$) were able to adsorbed on the negatively charged three-fold fcc hollow sites (Fig. 1b). Subsequently, the negatively charged single-atom precursors ($Ir(OH)_6^{2-}$) existing in the alkaline electrolyte can anchored onto the positively charged $V_O$ sites via electrostatic adsorption (Fig. 1c). The selective combination of single-atom precursors and anchoring sites provides opportunities to fabricate heterogeneous single atoms anchored at diverse sites.

Experimentally, CoOOH support was synthesized via an electrochemical deposition method with modifications[24]. Transmission

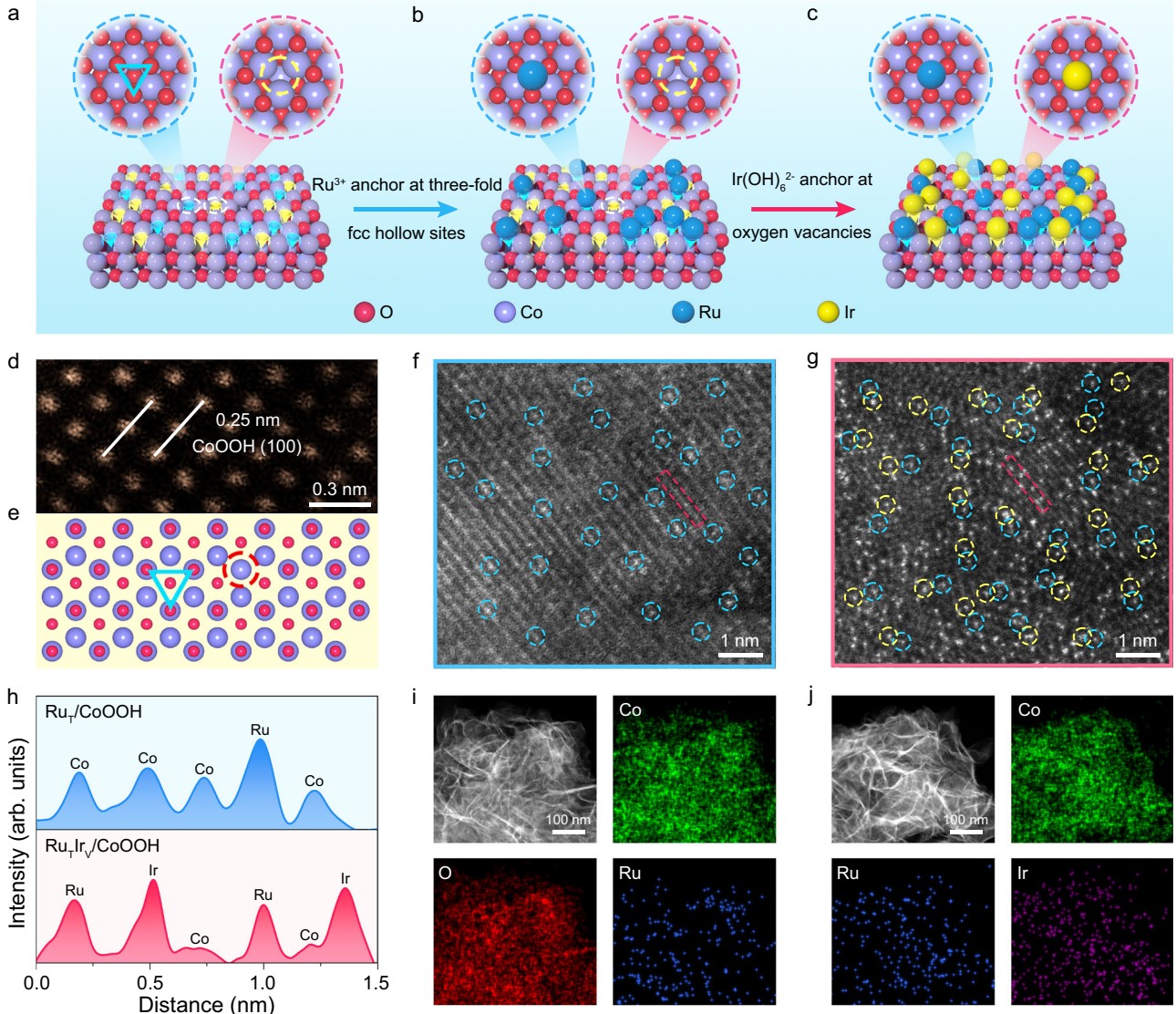

**Fig. 1 | Fabrication and spatial distribution of heterogeneous single atoms at diverse sites. a** Three-fold fcc hollow sites (indicated by the blue triangle) and $V_O$ sites (indicated by the yellow circle) on the surface of transition metal oxide. **b** Selectively anchor single-atom precursors $Ru^{3+}$ onto the three-fold fcc hollow sites. **c** Selectively anchor single-atom precursors $Ir(OH)_6^{2-}$ onto the $V_O$ sites to fabricate heterogeneous single atoms. Blue and yellow spheres represent Ru and Ir single atoms. Red and purple spheres represent the oxygen and Co atoms, respectively. HAADF-STEM image (**d**) and corresponding atomic structure models of CoOOH (**e**). HAADF-STEM image of $Ru_T$/CoOOH (**f**) and $Ru_T Ir_V$/CoOOH (**g**). Singly-dispersed Ru and Ir atoms are indicated by blue and yellow circles, respectively. **h** Line intensity profile obtained from the selected atomic column in the HAADF-STEM images of $Ru_T$/CoOOH and $Ru_T Ir_V$/CoOOH. EDX elemental mapping of $Ru_T$/CoOOH (**i**) and $Ru_T Ir_V$/CoOOH (**j**). Source data are provided as a Source Data file.

electron microscopy (TEM) image and X-ray diffraction (XRD) pattern demonstrated the as-obtained nanosheets attributed to the CoOOH (PDF #26-1107) (Supplementary Fig. 1a, b). The aberration-corrected high-angle annular dark-field scanning TEM (HAADF-STEM) image presented a lattice spacing of 0.25 nm, corresponding to the (100) facet of CoOOH (Fig. 1d). Besides, the deconvoluted O 1$s$ XPS spectrum of CoOOH displayed four characteristic peaks at 532.6 eV, 531.4 eV, 530.7 eV, and 529.3 eV, which were attributed to the adsorbed $H_2O$, $V_O$, Co-OH, and Co-O, respectively (Supplementary Fig. 2)[25,26]. The above results demonstrated that the surface of CoOOH contained both ordered atomic arrangement and defective sites, which provides three-fold fcc hollow sites and $V_O$ sites for anchoring single atoms (Fig. 1e).

Ru species were anchored onto the surface of CoOOH using a modified wet-chemical synthesis strategy[27]. During synthesis process, the positively charged $Ru^{3+}$ ions in the solution were selectively anchored onto negatively charged three-fold fcc hollow sites via electrostatic adsorption. TEM image showed that $Ru_T$/CoOOH displayed similar nanosheet morphologies relative to CoOOH (Supplementary Fig. 3a). The characteristic peaks in the XRD pattern of $Ru_T$/CoOOH were attributed to CoOOH, suggesting the absence of Ru-based metals or metal oxides (Supplementary Fig. 3b). HAADF-STEM image of $Ru_T$/CoOOH identified individual bright spots due to $Z$-contrast relative to the support, revealing the isolated dispersion of Ru atoms (Fig. 1f).

In the following step, Ir species were anchored onto the surface of $Ru_T$/CoOOH via an electrochemical deposition method[28]. In the synthesis process, the negatively charged $Ir(OH)_6^{2-}$ ions in the electrolyte were selectively anchored onto the positively charged $V_O$ sites by electrostatic adsorption. TEM image and XRD pattern of $Ru_T Ir_V$/CoOOH demonstrated the absence of Ru- or Ir-based metals or metal oxides (Supplementary Fig. 4a, b). HAADF-STEM image of $Ru_T Ir_V$/CoOOH identified isolated bright spots relative to the support, corresponding to Ru or Ir atoms (Fig. 1g). Based on element-specific electron scattering cross-sections of Ru ($Z = 44$) and Ir ($Z = 77$), the bright spots with lower brightness were Ru atoms and that with higher brightness were Ir atoms. Quantitative intensity analyses of the HAADF-STEM images showed two distinct intensities in the selected atomic column of $Ru_T$/CoOOH, with the lower intensity representing Co atoms and the higher intensity representing Ru atoms (Fig. 1h). For $Ru_T Ir_V$/CoOOH, three different intensities were derived in the selected atomic column, with the lowest intensity indicating Co atoms, the higher intensity describing Ru atoms, and the highest intensity representing Ir atoms (Fig. 1h). Furthermore, energy-dispersive X-ray (EDX) elemental mapping images exhibited the uniform distribution of Ru elements on the $Ru_T$/CoOOH (Fig. 1i), while both Ru and Ir elements across the $Ru_T Ir_V$/CoOOH (Fig. 1j). For comparison, $Ir_V$/CoOOH was fabricated by anchoring Ir single atoms onto the $V_O$ sites on CoOOH (Supplementary Fig. 5a–d). In addition, the $Ir_T Ru_V$/CoOOH was also prepared by anchoring Ir single atoms onto the three-fold fcc hollow sites and Ru single atoms onto the $V_O$ sites, respectively (Supplementary Fig. 6a–d).

To clearly identify the anchoring sites of Ru and Ir single atoms, we performed simulated HAADF-STEM images of $Ru_T$/CoOOH and $Ir_V$/CoOOH. In the simulated HAADF-STEM image of $Ru_T$/CoOOH, Ru atoms at three-fold fcc hollow sites almost overlap with the Co column (Supplementary Fig. 7a). Conversely, the Ir atoms at $V_O$ sites were located at the interstice of three triangular Co columns (Supplementary Fig. 7b). Therefore, we can precisely identify the anchoring sites of Ru and Ir single atoms based on the experimental HAADF-STEM images. In the experimental HAADF-STEM image of $Ru_T$/CoOOH, Ru single atoms nearly overlap with the Co column, suggesting the Ru single atoms were anchored at the three-fold fcc hollow sites (Supplementary Fig. 7c). Moreover, bright spots can be discerned in the interstice of three triangular lattice sites in the experimental HAADF-STEM image of $Ir_V$/CoOOH, which were ascribed to Ir single atoms at the $V_O$ sites

(Supplementary Fig. 7d). The above results provided direct evidence that the Ru and Ir single atoms were anchored at three-fold fcc hollow sites and $V_O$ sites, respectively. Quantitative analysis by inductively coupled plasma-atomic emission spectrometry (ICP-AES) indicated that the contents of Ru and Ir elements were about 4.6 wt% and 4.7 wt% for $Ru_T$/CoOOH and $Ir_V$/CoOOH, respectively. Furthermore, the contents of Ru and Ir elements were measured to be approximately 4.4 wt% and 4.9 wt% for $Ru_T Ir_V$/CoOOH, whereas the Ir and Ru elements were about 3.9 wt% and 4.0 wt% for $Ir_T Ru_V$/CoOOH.

## Atomic structural analysis of the heterogeneous single atoms at diverse sites

The detailed electronic structures and coordination environments of Ru and Ir single atoms were investigated by X-ray absorption near-edge spectroscopy (XANES) and extended X-ray absorption fine structure (EXAFS). As shown in Fig. 2a, the absorption edge of $Ru_T$/CoOOH and $Ru_T Ir_V$/CoOOH were located between Ru foil and $RuO_2$, indicating the valence state of Ru single atoms was between 0 and +4[29–31]. Meanwhile, the absorption edge of $Ru_T Ir_V$/CoOOH overlapped with that of $Ru_T$/CoOOH, presenting similar valence states of Ru single atoms. In the Ru $K$-edge EXAFS spectra, only one prominent peak was exhibited at about 1.5 Å for $Ru_T$/CoOOH and $Ru_T Ir_V$/CoOOH, which was ascribed to first-shell Ru-O coordination (Fig. 2b). The default of Ru-Ru bonding at about 2.4 Å substantiated the atomic dispersion of individual Ru atoms in both samples[32,33]. Wavelet transform (WT) of Ru $K$-edge EXAFS oscillations was conducted to further confirm the atomic dispersion of Ru species. The WT contour plot of $Ru_T$/CoOOH and $Ru_T Ir_V$/CoOOH showed a maximum intensity at around 4.5 Å$^{-1}$, corresponding to Ru-O scattering (Fig. 2c and Supplementary Fig. 8a). The absence of Ru-Ru scattering at around 7.4 Å$^{-1}$ evidenced of the isolated dispersion of Ru species (Supplementary Fig. 8b). By fitting the EXAFS spectra, the first-shell coordination of $Ru_T$/CoOOH and $Ru_T Ir_V$/CoOOH was both determined to be Ru-O with coordination numbers ($CN$s) of about 5.0 (Fig. 2g and Supplementary Table 1).

The electronic structures and coordination environments of Ir single atoms at the $V_O$ sites of $Ir_V$/CoOOH and $Ru_T Ir_V$/CoOOH were also investigated. In the Ir $L_3$-edge XANES spectra, the white line intensity of $Ir_V$/CoOOH and $Ru_T Ir_V$/CoOOH was near that of $IrO_2$, indicating the valence state of Ir single atoms was close to +4 (Fig. 2d)[34,35]. Moreover, $Ir_V$/CoOOH and $Ru_T Ir_V$/CoOOH showed an overlapped white line intensity, manifesting a similar valence state of Ir single atoms. In the EXAFS spectra, the samples both demonstrated only one characteristic peak at about 1.6 Å assigned to Ir-O bonding, which substantiated the isolated dispersion of Ir atoms (Fig. 2e)[36,37]. WT of Ir $L_3$-edge EXAFS oscillations exhibited only one maximum intensity at around 4.7 Å$^{-1}$ attributed to Ir-O scattering, confirming the atomic dispersion of Ir species on $Ir_V$/CoOOH and $Ru_T Ir_V$/CoOOH (Fig. 2f and Supplementary Fig. 8c, d). EXAFS fitting results demonstrated that the $CN$s of Ir-O for $Ir_V$/CoOOH and $Ru_T Ir_V$/CoOOH were about 6.0 (Fig. 2h and Supplementary Table 2). Therefore, the Ir single atoms of $Ir_V$/CoOOH and $Ru_T Ir_V$/CoOOH were coordinated with six oxygen atoms in the nearest neighbor to form an $IrO_6$ octahedral structure. Based on the synthesis mechanism and EXAFS results, Ru single atoms were stabilized by the three oxygen atoms at the three-fold fcc hollow sites, while the remaining two coordinated oxygen atoms were suspended at its surface as dangling bonds (Supplementary Fig. 9a). For Ir single atoms, one apex oxygen of the $IrO_6$ octahedral structure was inserted into the $V_O$ sites, while four side OH$^-$ of the octahedra formed hydrogen bonding with adjacent oxygen atoms on the CoOOH surface to stabilize the structure (Supplementary Fig. 9b). The above results demonstrated the configurations of Ru and Ir single atoms were site-specific.

The detailed electronic structures and coordination environments of Ir and Ru single atoms on $Ir_T Ru_V$/CoOOH were also investigated. In the Ir $L_3$-edge XANES spectra, the white line intensity of $Ir_T Ru_V$/CoOOH was higher than that of $Ru_T Ir_V$/CoOOH, indicating that

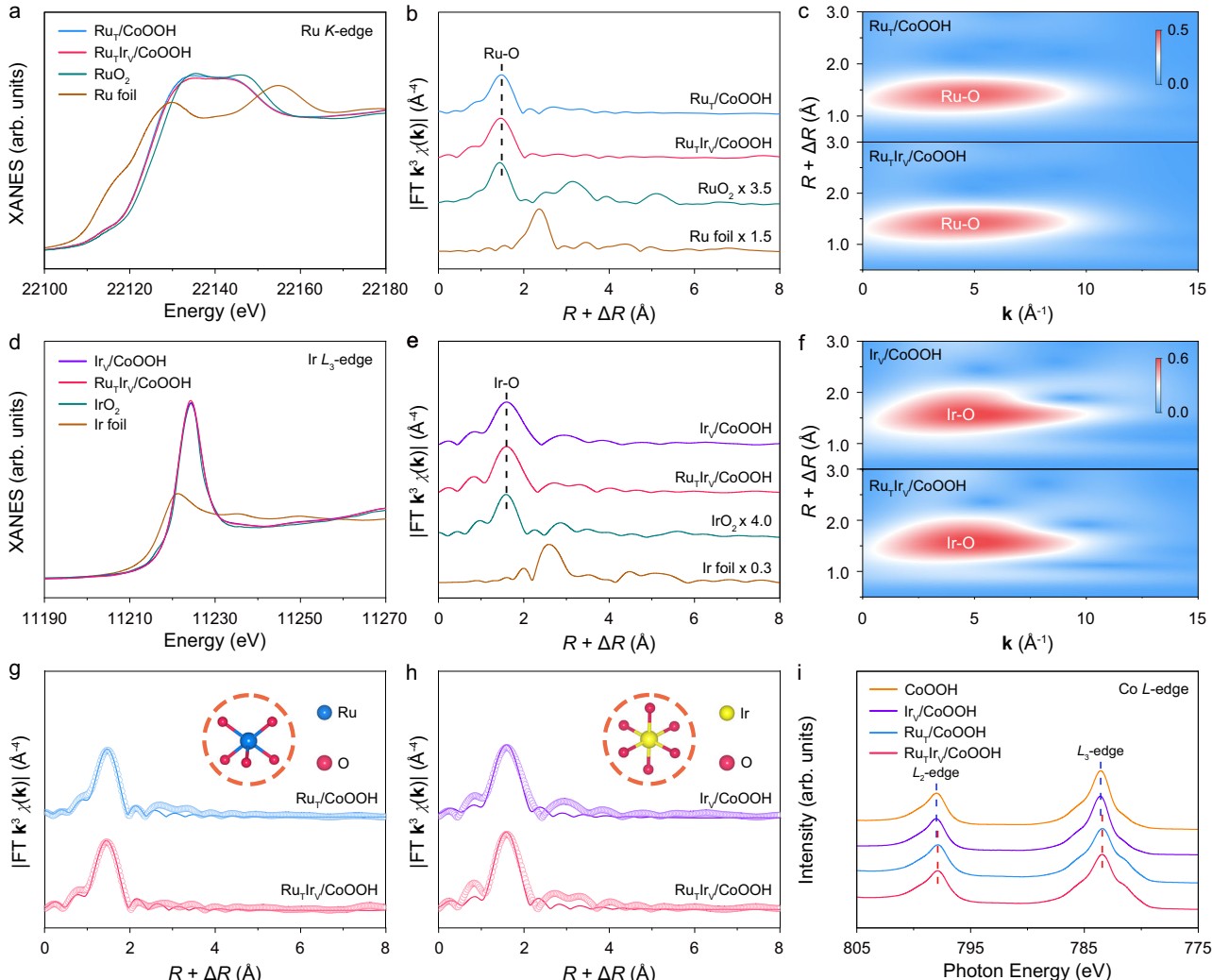

**Fig. 2 | Atomic structural analysis of the heterogeneous single atoms at diverse sites.** Normalized XANES (**a**) and EXAFS (**b**) spectra at the Ru $K$-edge of Ru$_T$/CoOOH and Ru$_T$Ir$_V$/CoOOH. $R$ and **k** denoted radial distance and wave vector, respectively. **c** WT of the **k**$^3$-weighted Ru $K$-edge EXAFS signals of Ru$_T$/CoOOH and Ru$_T$Ir$_V$/CoOOH. Normalized XANES (**d**) and EXAFS (**e**) spectra at the Ir $L_3$-edge of Ir$_V$/CoOOH and Ru$_T$Ir$_V$/CoOOH. **f** WT of the **k**$^3$-weighted Ir $L_3$-edge EXAFS signals of Ir$_V$/CoOOH and Ru$_T$Ir$_V$/CoOOH. **g** Experimental and fitting EXAFS spectra at the Ru $K$-edge of Ru$_T$/CoOOH and Ru$_T$Ir$_V$/CoOOH. **h** Experimental and fitting EXAFS spectra at the Ir $L_3$-edge of Ir$_V$/CoOOH and Ru$_T$Ir$_V$/CoOOH. The experimental and fitting results are indicated as circles and solid lines, respectively. The inset atomic models are the first-shell coordination of Ru and Ir atoms, respectively. The red, blue, and yellow spheres represent O, Ru, and Ir atoms, respectively. **i** Co $L$-edge XAS spectra. Source data are provided as a Source Data file.

the valence state of Ir single atoms anchored at three-fold fcc hollow sites was higher than that of anchored at V$_O$ sites (Supplementary Fig. 10a). In the Ir $L_3$-edge EXAFS spectra, the Ir$_T$Ru$_V$/CoOOH exhibited only one characteristic peak at about 1.6 Å assigned to Ir-O bonding, similar to that of Ru$_T$Ir$_V$/CoOOH, which substantiated the isolated dispersion of Ir atoms (Supplementary Fig. 10b). EXAFS fitting results demonstrated that the $CN$s of Ir-O for Ir$_T$Ru$_V$/CoOOH was about 6.0 (Supplementary Table 3). Therefore, the Ir single atoms at three-fold fcc hollow sites were stabilized by three oxygen atoms of the sites, while the remaining three coordinated oxygen atoms were suspended at its surface as dangling bonds (Supplementary Fig. 11a, b).

Subsequently, we investigated the electronic structures and coordination environments of Ru single atoms at the V$_O$ sites of Ir$_T$Ru$_V$/CoOOH. In the Ru $K$-edge XANES spectra, the absorption edge of Ir$_T$Ru$_V$/CoOOH shifted to a higher energy than that of Ru$_T$Ir$_V$/CoOOH, indicating an elevated valence state of Ru single atoms (Supplementary Fig. 12a). In the Ru $K$-edge EXAFS spectra, only one prominent peak was exhibited at about 1.5 Å for Ir$_T$Ru$_V$/CoOOH, confirming the atomic dispersion of individual Ru atoms (Supplementary Fig. 12b). By fitting

the EXAFS spectra, the first-shell coordination of Ir$_T$Ru$_V$/CoOOH was determined to be Ru-O with $CN$s of about 6.0, which differs from the Ru single atoms at three-fold fcc hollow sites that are coordinated with five oxygen atoms (Supplementary Table 4). Accordingly, for Ru single atoms at V$_O$ sites, one apex oxygen of the RuO$_6$ octahedral structure was inserted into the V$_O$ sites, while four side OH$^-$ of the octahedra formed hydrogen bonding with adjacent oxygen atoms on the CoOOH surface to stabilize the structure (Supplementary Fig. 13a, b). The above results demonstrated that the differences in the anchoring sites result in distinct electronic structures and configurations of Ru and Ir single atoms on Ir$_T$Ru$_V$/CoOOH and Ru$_T$Ir$_V$/CoOOH (Supplementary Fig. 14a, b).

To reveal the influence of heterogeneous Ru and Ir single atoms at diverse sites on the electronic structure of Co species, X-ray absorption spectroscopy (XAS) and XPS were conducted. As shown in Fig. 2i, all samples exhibited $L_3$- and $L_2$-edge absorption features at around 783.5 and 797.9 eV, which arise from Co $2p_{3/2}$ and Co $2p_{1/2}$ to Co $3d$ transitions, respectively. The Co $L_3$- and $L_2$- edges of Ru$_T$/CoOOH were shifted to a lower photon energy position than that of CoOOH,

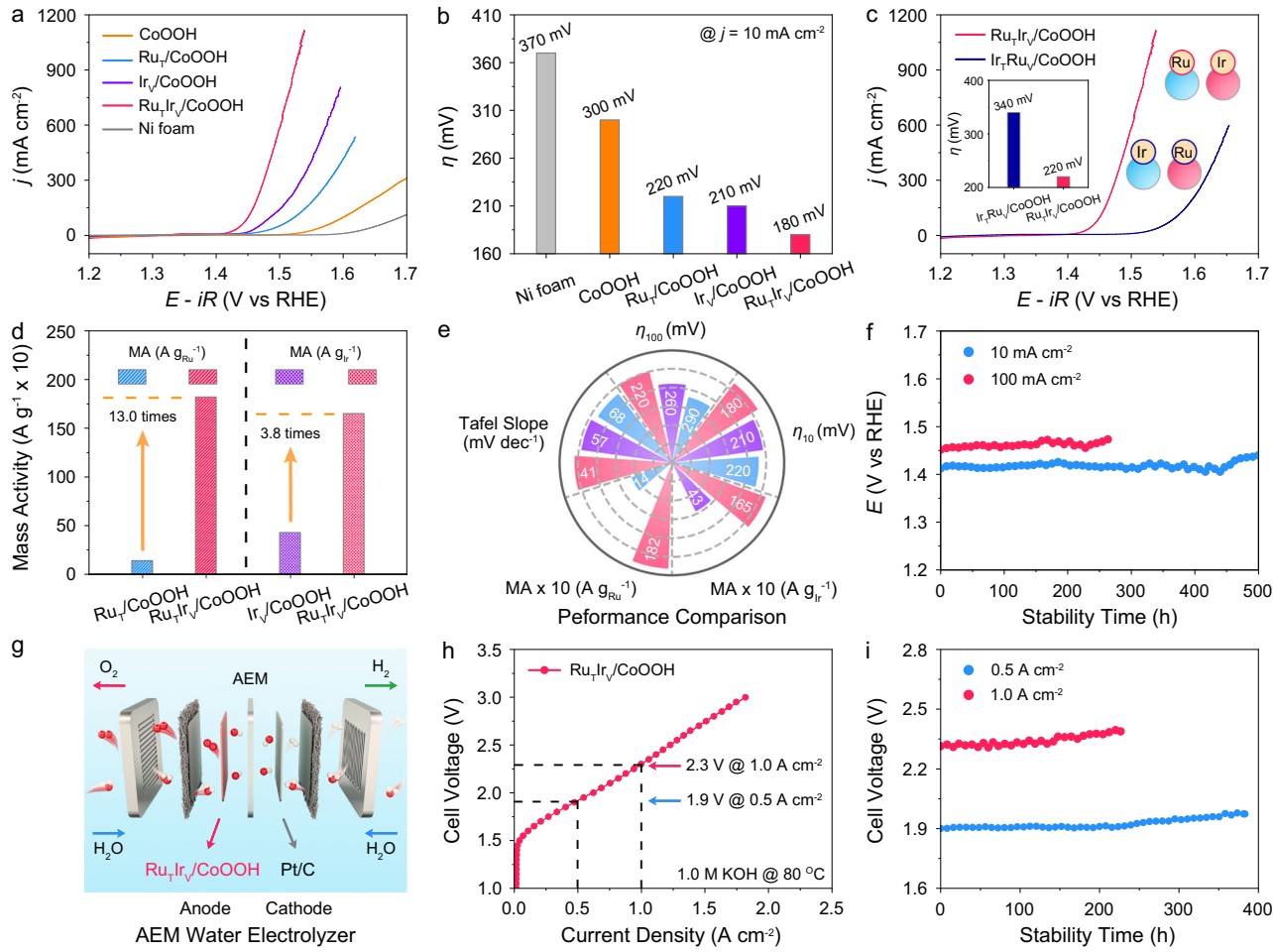

**Fig. 3 | Electrocatalytic performance towards oxygen evolution. a** Polarization curves of catalysts towards oxygen evolution in 1.0 M KOH electrolyte with *iR*-compensation. *R* was measured to be 0.75 Ω. **b** Overpotentials of Ni foam, CoOOH, $Ru_T$/CoOOH, $Ir_V$/CoOOH, and $Ru_TIr_V$/CoOOH at a current density of 10 mA cm$^{-2}$ in a single experiment. **c** Polarization curves of $Ru_TIr_V$/CoOOH and $Ir_TRu_V$/CoOOH towards oxygen evolution in 1.0 M KOH electrolyte. The inset figure is the overpotentials of $Ru_TIr_V$/CoOOH and $Ir_TRu_V$/CoOOH at a current density of 100 mA cm$^{-2}$. **d** Mass activities of samples against the mass loadings of Ru and Ir single atoms at an overpotential of 250 mV. **e** Intrinsic OER activity comparison. The blue, purple, and red columns indicated $Ru_T$/CoOOH, $Ir_V$/CoOOH, and $Ru_TIr_V$/CoOOH, respectively. **f** Chronopotentiometry curves of $Ru_TIr_V$/CoOOH towards OER at a current density of 10 mA cm$^{-2}$ and 100 mA cm$^{-2}$. Schematic diagram of the AEM water electrolyzer (**g**) and polarization curve of $Ru_TIr_V$/CoOOH in the AEM water electrolyzer without *iR*-compensation (**h**). The white and red spheres represent H and O atoms, respectively. **i** Chronopotentiometry test of $Ru_TIr_V$/CoOOH at a current density of 0.5 A cm$^{-2}$ and 1.0 A cm$^{-2}$ in the AEM water electrolyzer. Source data are provided as a Source Data file.

indicating the Ru single atoms at three-fold fcc hollow sites reduced the valence state of Co species. Remarkably, the photon energy position of the Co *L*-edge for $Ru_TIr_V$/CoOOH exhibited an inappreciable change compared to that of $Ru_T$/CoOOH, demonstrating an unaltered Co valence state after anchoring Ir single atoms at $V_O$ sites. Moreover, the XANES spectra at the Co *K*-edge and Co 2*p* XPS spectra further verified these results (Supplementary Figs. 15, 16)[38,39]. The above results indicated the interaction between heterogeneous single atoms and Co species was site-specific. Specifically, Ru single atoms at three-fold fcc hollow sites exhibited stronger interaction with Co species than Ir single atoms anchored at $V_O$ sites.

## Electrocatalytic performance towards oxygen evolution
To investigate the synergy in heterogeneous Ru and Ir single atoms, we evaluated their catalytic performance toward oxygen evolution, one of the most essential reactions for energy conversion[40–42]. The experiments were conducted in a standard three-electrode system with a 1.0 M KOH electrolyte. The Ni foam exhibited limited electrocatalytic activity towards OER, indicating the primary contributor to catalyst performance was the CoOOH-supported SACs (Fig. 3a and

Supplementary Fig. 17). In addition, $Ru_T$/CoOOH, $Ir_V$/CoOOH, and $Ru_TIr_V$/CoOOH showed dramatically improved current densities relative to the original CoOOH. Especially for $Ru_TIr_V$/CoOOH, the overpotential (*η*) required to reach a current density of 10 mA cm$^{-2}$ was 180 mV, which was 40 mV and 30 mV lower than that of $Ru_T$/CoOOH and $Ir_V$/CoOOH (Fig. 3b). Notably, it achieved a current density of 100 mA cm$^{-2}$ at an overpotential of 220 mV, which was 70 mV and 40 mV lower than that of $Ru_T$/CoOOH and $Ir_V$/CoOOH (Supplementary Fig. 18).

The catalytic activity of $Ir_T$/CoOOH and $Ir_TRu_V$/CoOOH was also evaluated to explore the intrinsic correlation between the synergy in heterogeneous single atoms and their anchoring sites. OER performance evaluation demonstrated that the $\eta_{10}$ of $Ir_T$/CoOOH and $Ir_TRu_V$/CoOOH were 280 mV and 270 mV, which were 100 mV and 90 mV higher than that of $Ru_TIr_V$/CoOOH, respectively (Supplementary Fig. 19a–c). In addition, the *η* required to reach a current density of 100 mA cm$^{-2}$ was 340 mV for $Ir_TRu_V$/CoOOH, which was 120 mV higher than that of $Ru_TIr_V$/CoOOH (Fig. 3c). The above results demonstrated the synergy in heterogeneous Ru and Ir single atoms was site-specific. The Ru single atoms at the three-fold fcc hollow sites and Ir single

atoms at $V_O$ sites synergistically enhanced the OER activity of CoOOH. In contrast, exchanging the anchoring sites of Ru and Ir single atoms resulted in a decreased activity.

To validate the reproducibility of electrochemical performance, the samples were tested using Hg/HgO as the reference electrode. The polarization curves showed that the performance of samples using Hg/HgO closely overlapped with those using Ag/AgCl (Supplementary Fig. 20a–d). In addition, three independent electrochemical performance evaluations were conducted to further confirm the reproducibility of the experiments. The results exhibited inconspicuous differences in the polarization curves across the three independent tests for each sample (Supplementary Fig. 21a, b). Furthermore, the minimal differences in $\eta_{10}$ across the three measurements indicated the reproducibility of electrochemical performance (Supplementary Fig. 21c, d).

To compare the intrinsic activity, the current densities of the samples were normalized against the mass loadings of Ru and Ir species, respectively. The results showed that $Ru_TIr_V$/CoOOH exhibited values of 1816.0 A $g_{Ru}^{-1}$ and 1646.3 A $g_{Ir}^{-1}$ at $\eta$ of 250 mV, which were 13.0 and 3.8 times higher than $Ru_T$/CoOOH and $Ir_V$/CoOOH, respectively (Fig. 3d and Supplementary Fig. 22a, b). Moreover, the $Ir_TRu_V$/CoOOH exhibited mass activities of 31.4 A $g_{Ru}^{-1}$ and 33.0 A $g_{Ir}^{-1}$ at $\eta$ of 250 mV. In comparison, the mass activities of $Ru_TIr_V$/CoOOH were 57.8 and 49.9 times higher than those of $Ir_TRu_V$/CoOOH at $\eta$ of 250 mV, respectively (Supplementary Fig. 23a-c). The specific activities of the samples were evaluated by normalizing the current densities to their electrochemical active surface areas (ECSAs) (Supplementary Fig. 24a–j). At $\eta$ of 250 mV, the $Ru_TIr_V$/CoOOH exhibited a specific activity value of 625.2 mA cm$^{-2}$, which was 111.6, 12.6, and 4.1 times higher than that of CoOOH, $Ru_T$/CoOOH, $Ir_V$/CoOOH, respectively (Supplementary Fig. 25a, b). Additionally, $Ir_TRu_V$/CoOOH exhibited a specific activity of 11.7 mA cm$^{-2}$ at $\eta$ of 250 mV, which was 53.4 times lower than that of $Ru_TIr_V$/CoOOH (Supplementary Fig. 25c, d).

The reaction kinetics of the samples were assessed by Tafel slopes and electrochemical impedance spectroscopy (EIS). CoOOH, $Ru_T$/CoOOH, $Ir_V$/CoOOH, and $Ru_TIr_V$/CoOOH showed Tafel slope values of 76, 68, 57, and 41 mV dec$^{-1}$, respectively (Supplementary Fig. 26a). In addition, the $Ir_T$/CoOOH and $Ir_TRu_V$/CoOOH exhibited the Tafel slope values of 67 and 64 mV dec$^{-1}$, respectively, which were 26 and 23 mV dec$^{-1}$ higher than that of $Ru_TIr_V$/CoOOH (Supplementary Fig. 26b). The lowest Tafel slope of $Ru_TIr_V$/CoOOH indicated its fastest kinetics among the catalysts. The result was further reflected by the EIS measurements (Supplementary Fig. 27). The smallest semicircle diameter of $Ru_TIr_V$/CoOOH suggested its fastest charge transfer at the interface, which was beneficial to accelerate OER kinetics. Comparatively, the $\eta_{10}$ and the Tafel slope of $Ru_TIr_V$/CoOOH were comparable to those of the currently reported high-performance OER catalysts. (Supplementary Fig. 28 and Supplementary Table 5). The results demonstrated the efficient synergy between Ru single atoms at three-fold fcc hollow sites and Ir single atoms at $V_O$ sites promoted the OER performance (Fig. 3e).

Durability tests were carried out to estimate the stability of $Ru_TIr_V$/CoOOH. As shown in Fig. 3f, $Ru_TIr_V$/CoOOH achieved a lifetime of 500 h at a current density of 10 mA cm$^{-2}$. Furthermore, the OER current density of $Ru_TIr_V$/CoOOH showed no apparent attenuation for over 260 h at a current density of 100 mA cm$^{-2}$. The dissolved Ru and Ir species during the stability test were quantified using inductively coupled plasma-mass spectrometry (ICP-MS). The results demonstrated merely 5.9 and 7.0 wt% of Ru and Ir species were dissolved during 500 h stability test, respectively (Supplementary Fig. 29). The morphology and structure of $Ru_TIr_V$/CoOOH after OER were also characterized. No obvious metal clusters or particles were identified in the TEM image and XRD pattern (Supplementary Fig. 30a, b) In the HAADF-STEM image, the atomic dispersion of Ru and Ir single atoms were preserved after the test (Supplementary Fig. 30c). EDX elemental

mapping images still showed the uniform elements distribution of Ru and Ir elements across the $Ru_TIr_V$/CoOOH (Supplementary Fig. 30 d). The above results demonstrated the impressive stability of $Ru_TIr_V$/CoOOH for oxygen evolution.

To further evaluate the application potential of $Ru_TIr_V$/CoOOH for industrial water splitting, an anion-exchange membrane (AEM) water electrolyzer was assembled, with cathodic reaction supported by commercial Pt/C for hydrogen evolution (Fig. 3g). The polarization curves showed the $Ru_TIr_V$/CoOOH exhibit a cell voltage of 1.9 V and 2.3 V at current densities of 0.5 A cm$^{-2}$ and 1.0 A cm$^{-2}$ for AEM water electrolyzer, respectively (Fig. 3h). Furthermore, the AEM water electrolyzer was continuously operated for over 380 h and 220 h at a current density of 0.5 A cm$^{-2}$ and 1.0 A cm$^{-2}$, respectively (Fig. 3i). The overpotential at 1.0 A cm$^{-2}$ and stability time of $Ru_TIr_V$/CoOOH were comparable to those of recently reported catalysts for AEM water electrolyzers (Supplementary Fig. 31 and Supplementary Table 6). The activity and stability of $Ru_TIr_V$/CoOOH were also evaluated by assembling the sample into a membrane electrode assembly (MEA) water electrolyzer (Supplementary Fig. 32a, b). The polarization curves showed the $Ru_TIr_V$/CoOOH exhibited a cell voltage of 2.3 V at current densities of 1.0 A cm$^{-2}$ (Supplementary Fig. 32c). Stability evaluation demonstrated the MEA incorporating $Ru_TIr_V$/CoOOH as the anode catalysts operated for over 87 h at a current density of 1.0 A cm$^{-2}$ (Supplementary Fig. 32d). The above results highlighted the potential of $Ru_TIr_V$/CoOOH for industrial water splitting applications.

## In-situ spectroscopic analysis

To reveal the intrinsic reason for the site-specific synergy in heterogeneous Ru and Ir single atoms, in-situ X-ray absorption fine structure (XAFS) measurements were carried out. The measurements were conducted in a specialized electrolytic cell using a standard three-electrode system (Supplementary Fig. 33a, b). The in-situ Ru $K$-edge XANES spectra and the corresponding local magnification exhibited the absorption edge shift to higher energy when the applied voltage increased from open circuit potential (OCP) to 1.65 V, indicating an elevated Ru valence state (Fig. 4a, e)[43–45]. Notably, when the applied potential was reversed back to OCP, the absorption edge at the Ru $K$-edge of $Ru_TIr_V$/CoOOH shifted to an energy position close to the OCP, which may originate from the desorption of oxygenated intermediates (Supplementary Fig. 34a, b). In-situ EXAFS spectra demonstrated only one prominent peak at about 1.5 Å assigned to Ru-O bonding, confirming the single-atom structure of Ru species (Fig. 4b and Supplementary Fig. 34c). The EXAFS fitting results revealed that the coordination numbers of Ru-O increased from 5.0 to 5.4 as the applied voltage increased from OCP to 1.65 V, indicating the coordinated oxygen of Ru single atoms underwent dynamic evolution (Fig. 4f, Supplementary Fig. 35, and Supplementary Table 7). Remarkably, when the applied potential was reversed to OCP, the coordination numbers of Ru-O decreased from 5.4 to 5.1, in response to the desorption of oxygenated intermediates out of OER conditions (Supplementary Fig. 34d). The above results proved the Ru single atoms serve as the active sites for adsorption of oxygenated reaction intermediates.

In-situ Ir $L_3$-edge XAFS spectra of $Ru_TIr_V$/CoOOH were also performed to investigate Ir species' potential coordination environment evolution. The relative energy of in-situ Ir $L_3$-edge XANES absorption positions showed negligible shifts during the experiments (Fig. 4c, e). In addition, the in-situ Ir $L_3$-edge XANES spectra exhibited an increased white line intensity with applied potentials from OCP to 1.65 V, corresponding to an elevated valence state of Ir species under oxidation potentials (Fig. 4c)[34,46]. The EXAFS spectra showed only one prominent peak at about 1.6 Å, which was attributed to Ir-O bonding, validating the single-atom structure of Ir species (Fig. 4d). Further EXAFS fitting results demonstrated the Ir single atoms consistently coordinated with six oxygen as oxidation potential increasing, indicating no detectable

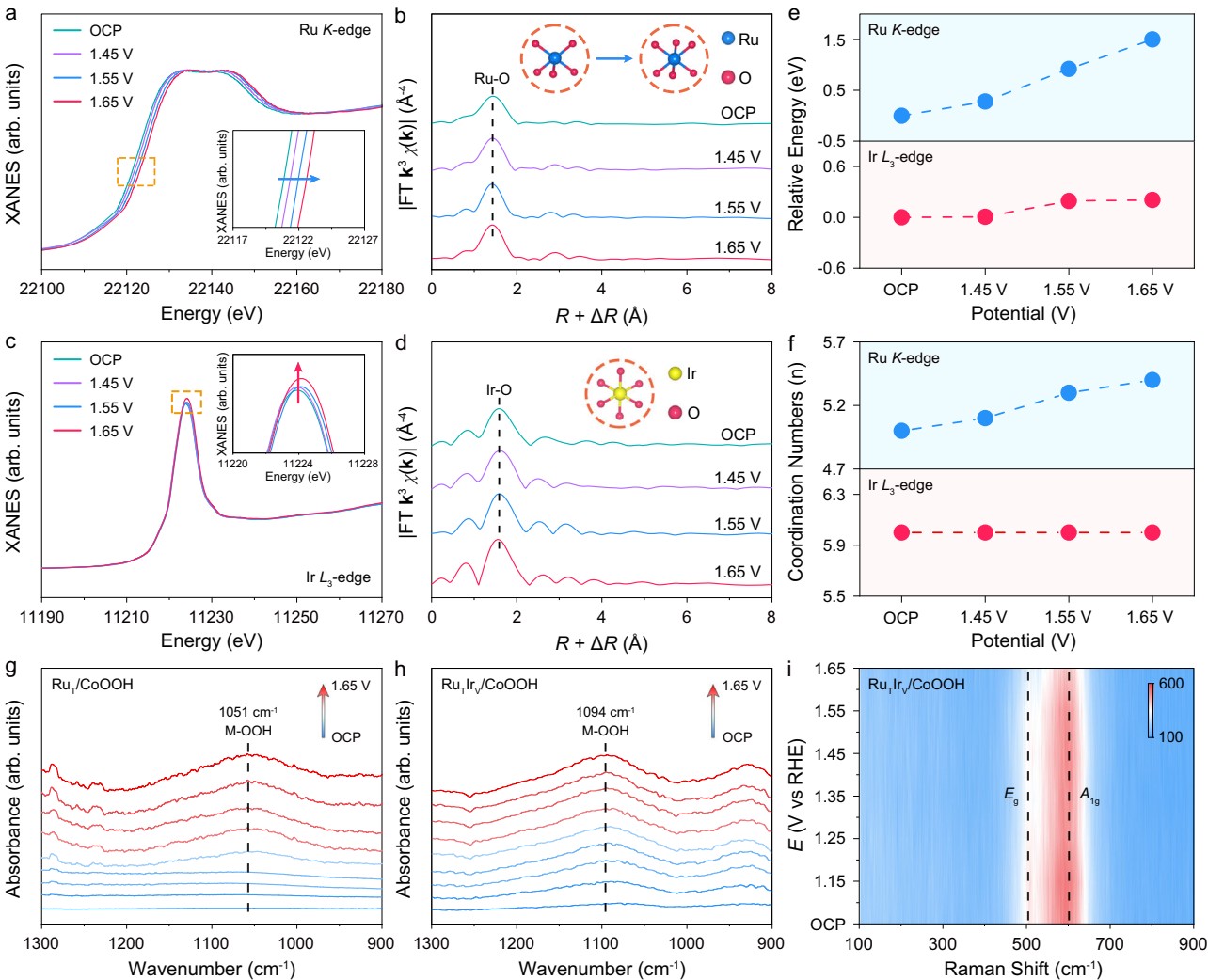

**Fig. 4 | In-situ spectroscopic characterizations of Ru$_T$Ir$_V$/CoOOH.** In-situ Ru $K$-edge XANES (**a**) and EXAFS (**b**) spectra of Ru$_T$Ir$_V$/CoOOH. $R$ and **k** denoted radial distance and wave vector, respectively. In-situ Ir $L_3$-edge XANES (**c**) and EXAFS (**d**) spectra of Ru$_T$Ir$_V$/CoOOH. **e** The relative energy of absorption edge in Ru $K$-edge XANES and Ir $L_3$-edge XANES. **f** The coordination numbers of Ru and Ir single atoms during the in-situ XAFS measurements. In-situ ATR-SEIRAS of Ru$_T$/CoOOH (**g**) and Ru$_T$Ir$_V$/CoOOH (**h**). **i** In-situ Raman spectra of Ru$_T$Ir$_V$/CoOOH. Source data are provided as a Source Data file.

coordination environments evolution (Fig. 4f, Supplementary Fig. 36 and Supplementary Table 8). The negligible change in the Ir-O coordination numbers suggests the enhanced activity of Ru$_T$Ir$_V$/CoOOH compared to Ru$_T$/CoOOH may originate from other factors.

In-situ attenuated total reflection surface-enhanced infrared absorption spectroscopy (ATR-SEIRAS) was performed to sensitively detect the key reaction intermediates and identify the reaction mechanism (Supplementary Fig. 37a, b). As shown in Fig. 4g, with the applied potential on Ru$_T$/CoOOH increased from OCP to 1.65 V, an absorption band at about 1051 cm$^{-1}$ showed a potential-dependent behavior, which can be assigned to the adsorption of *OOH species[47–51]. The production of key *OOH species under OER conditions suggested the Ru$_T$/CoOOH follows the adsorbate evolution mechanism[52–54]. Meanwhile, the *OOH wavenumber of Ru$_T$Ir$_V$/CoOOH was located at about 1094 cm$^{-1}$ (Fig. 4h). The blue shift of *OOH characteristic peak of Ru$_T$Ir$_V$/CoOOH illustrated an enhanced O-O bond vibration, which stems from weaker adsorption of *OOH intermediates modulated by Ir single atoms. Furthermore, in-situ ATR-SEIRAS spectra of Ir$_T$/CoOOH and Ir$_T$Ru$_V$/CoOOH were recorded. The results demonstrated the *OOH wavenumber of Ir$_T$/CoOOH and Ir$_T$Ru$_V$/CoOOH were positioned at about 1040 cm$^{-1}$ and 1038 cm$^{-1}$, respectively (Supplementary Fig. 38a, b). The close characteristic peak wavenumber of *OOH

indicated the Ru single atoms at the V$_O$ sites did not influence the adsorption of *OOH intermediates on the Ir single atoms at the three-fold fcc hollow sites.

To further elucidate the reaction pathways, in-situ $^{18}$O isotope-labeling differential electrochemical mass spectrometry (DEMS) experiments were carried out (Supplementary Fig. 39). The mass signal for $^{36}$O$_2$ ($^{18}$O$^{18}$O) was barely discernible in all samples, precluding the oxygen-oxygen coupling mechanism (OPM) as reaction pathway for all samples (Supplementary Fig. 40a–d). Meanwhile, a trace amount of $^{34}$O$_2$ ($^{18}$O$^{16}$O) in the OER products was detected, which can be attributed to the natural abundance of $^{18}$O in deionized water within the experiments, suggesting the lattice oxygen-mediated mechanism (LOM) did not occur over those samples. Moreover, the predominant mass signal corresponded to $^{32}$O$_2$ ($^{16}$O$^{16}$O) was observed for all samples, indicating the Ru$_T$/CoOOH, Ir$_V$/CoOOH, Ir$_T$Ru$_V$/CoOOH, and Ru$_T$Ir$_V$/CoOOH follow the adsorbate evolution mechanism for oxygen evolution.

In-situ Raman measurements were further employed to probe the potential structure evolution of the support under realistic OER conditions. Two featured peaks at about 500 and 601 cm$^{-1}$ were observed with increasing applied potentials from OCP to 1.65 V, which were assigned to the $E_g$ and $A_{1g}$ vibration of the CoOOH phase, respectively (Fig. 4i)[55,56]. The settled positions of the featured peaks proved

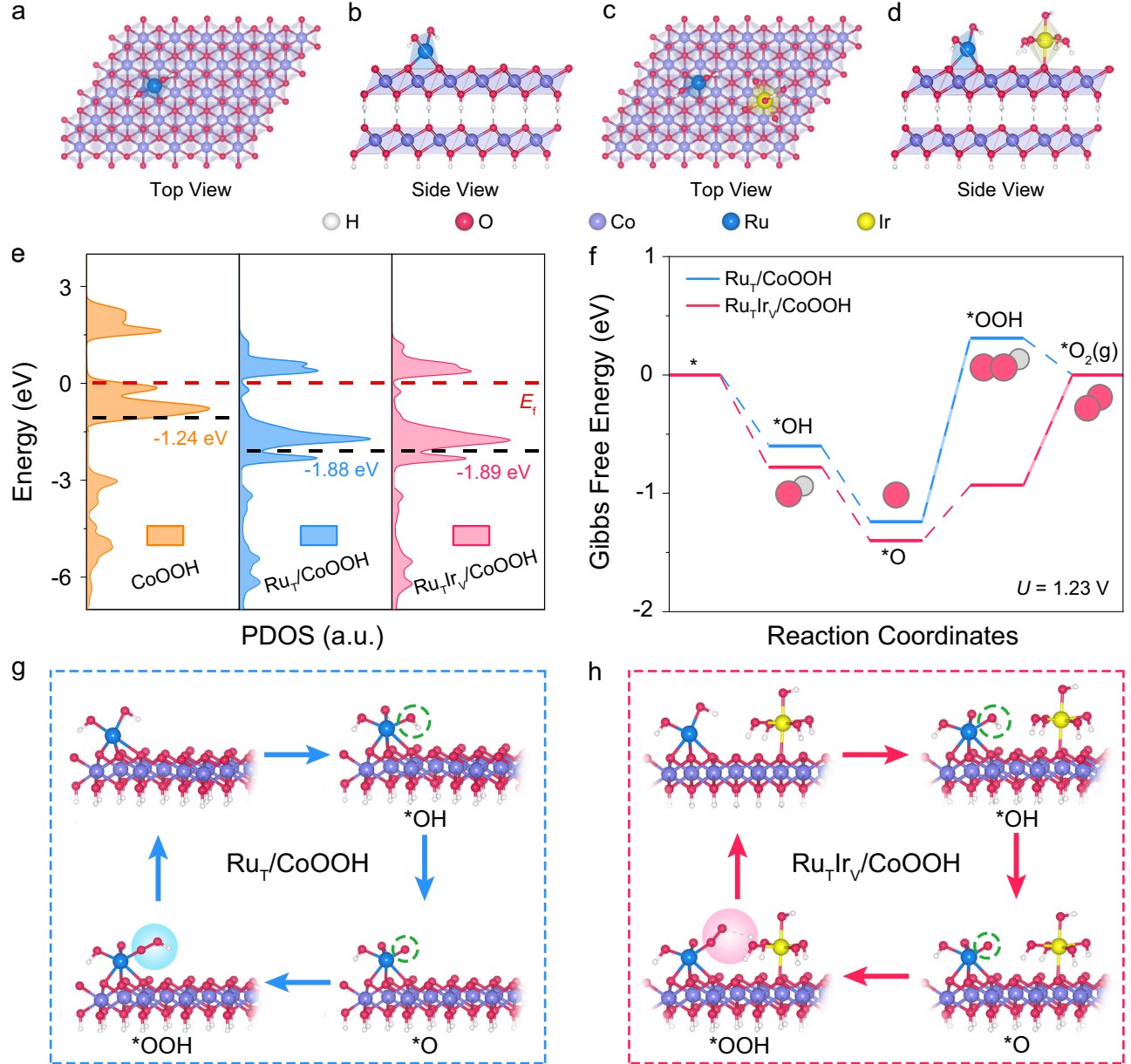

**Fig. 5 | Mechanistic studies.** Schematic structure model of $Ru_T/CoOOH$ from the top (**a**) and side (**b**) view. Schematic structure model of $Ru_TIr_V/CoOOH$ from the top (**c**) and side (**d**) view. **e** Co 3*d* PDOS of CoOOH, $Ru_T/CoOOH$, and $Ru_TIr_V/CoOOH$. **f** Free-energy diagrams of $Ru_T/CoOOH$ and $Ru_TIr_V/CoOOH$ toward OER. The schematic OER pathways of $Ru_T/CoOOH$ (**g**) and $Ru_TIr_V/CoOOH$ (**h**). The white, red, purple, blue, and yellow spheres represent H, O, Co, Ru, and Ir atoms, respectively. Source data are provided as a Source Data file.

negligible structure evolution of the support under oxygen evolution conditions. In-situ XAFS measurements were also conducted to investigate the potential evolution of Co species in $Ru_TIr_V/CoOOH$ under OER conditions (Supplementary Fig. 41a, b). The position of the absorption edge and the characteristic peaks in the EXAFS spectra exhibited negligible changes, indicating that the valence state and coordination environment of the Co species remained stable during OER (Supplementary Fig. 41c, d).

## Mechanistic studies

To provide an in-depth insight into the site-specific synergy in heterogeneous Ru and Ir single atoms for improving the oxygen evolution activity, systematic density functional theory (DFT) calculations were performed. Based on the Ru *K*-edge EXAFS results, Ru single atoms were stabilized by the three oxygen atoms at the three-fold fcc hollow sites, while the remaining two coordinated oxygen atoms were

suspended at its surface as dangling bonds (Fig. 5a, b). For Ir single atoms, $Ir(OH)_6^{2-}$ octahedra were fitted into the $V_O$ sites, and four side $OH^-$ of the octahedra formed hydrogen bonding with adjacent oxygen atoms on the CoOOH surface to stabilize the structure (Fig. 5c, d). The possible electronic interactions between Ru and Ir single atoms with support were elucidated by projected density of states (PDOS) calculations (Supplementary Data 1). Compared to the Co *d*-band center of original CoOOH located at −1.24 eV, the Co *d*-band center of $Ru_T/CoOOH$ was profoundly down-shifted to −1.88 eV, indicating a stronger interaction between Ru single atoms at the three-fold fcc hollow sites and the support (Fig. 5e). Notably, after anchored Ir single atoms onto the $V_O$ sites, the *d*-band center of Co species for $Ru_TIr_V/CoOOH$ was evaluated to be −1.89 eV, suggesting the interaction between Ir single atoms at the $V_O$ sites and the support was weaker. The site-specific interactions between single atoms and the support may originate from the distinct configurations of single atoms at different

sites. Specifically, the Ru single atoms at the three-fold fcc hollow sites interacted with Co atoms through three O atoms, while the Ir single atoms at the $V_O$ sites interacted with Co atoms via one O atom. The increased number of connected O atoms resulted in stronger interactions between Ru single atoms and Co species compared to those between Ir single atoms and Co species.

To elucidate the underlying OER mechanisms, a free-energy diagram toward OER was evaluated following the potential pathways. Ru single atoms were identified as active sites for the adsorption of reaction key intermediates according to the in-situ XAFS results. The adsorbate evolution mechanism with four concerted proton-electron transfer steps was considered the reaction mechanism based on the results of in-situ ATR-SEIRAS and $^{18}O$ isotope-labeling DEMS measurements. For $Ru_T/CoOOH$, the rate-determining step (RDS) was the formation of *OOH from *O. (Fig. 5f and Supplementary Table 9). Notably, the RDS for $Ru_TIr_V/CoOOH$ was shifted to the formation of $O_2(g)$ from *OOH intermediates. Moreover, the synergetic interaction in Ru and Ir single atoms resulted in a notable reduction of theoretical overpotential by 0.62 V, thereby enhancing the OER performance of $Ru_TIr_V/CoOOH$.

To further investigate the origins of the elevated activities induced by Ru and Ir single atoms, the reaction pathways towards OER were probed in detail. For $Ru_T/CoOOH$, the reaction started from the adsorption of $OH^-$ ions on the Ru site, followed by the sequential deprotonation to form *O, O-O bonding formation to generate OOH*, and desorption to produce $O_2(g)$. Remarkably, for $Ru_TIr_V/CoOOH$, as the reaction proceeds to the *OOH step, the distance between *OOH intermediates (H atom) on Ru single atom and coordinated hydroxide radicals of Ir single atom (O atom) was determined to be 2.37 Å, indicating the presence of hydrogen bonding interaction. The hydrogen bonding interactions stabilized the *OOH intermediates, resulting in a lower Gibbs free energy barrier and changing the RDS to the dehydrogenation step from *OOH to $O_2(g)$.

## Discussion

In conclusion, we revealed the site-specific synergy in heterogeneous Ru and Ir single atoms for oxygen evolution. The heterogeneous single-atom catalyst $Ru_TIr_V/CoOOH$ was fabricated by selectively anchoring Ru single atoms onto three-fold fcc hollow sites and Ir single atoms onto $V_O$ sites. In addition, $Ir_TRu_V/CoOOH$ was also prepared by switching the anchoring sites of Ru and Ir single atoms. Electrochemical measurements demonstrated that the $Ru_TIr_V/CoOOH$ exhibited enhanced OER performance compared to $Ir_TRu_V/CoOOH$. Specifically, the $Ru_TIr_V/CoOOH$ exhibited an overpotential of 180 mV at a current density of $10 mA cm^{-2}$ for OER, which impressively decreased compared with 270 mV of $Ir_TRu_V/CoOOH$. A series of in-situ spectroscopic characterizations and mechanistic studies indicated that the Ru single atoms at the three-fold fcc hollow sites acted as adsorption sites and Ir single atoms at $V_O$ sites stabilized the *OOH intermediates on the Ru single atoms via hydrogen bonding interactions. This work not only proposed a synthesis strategy for constructing heterogeneous single atoms but also disclosed the correlation between the synergy in heterogeneous single atoms and their anchoring sites.

## Methods

Chemicals. Cobalt (II) nitrate hexahydrate $(Co(NO_3)_2 \cdot 6H_2O$, 99.99% metals basis) and potassium hydroxide (KOH, 99.999%) were purchased from Macklin Co., China. Ethanol (EtOH, ≥99.7%), hydrochloric acid (HCl, 36.0–38.0%), and nitric acid ($HNO_3$, 65.0–68.0%) were purchased from Sinopharm Chemical Reagent Co., Ltd. Ruthenium chloride hydrate ($RuCl_3 \cdot xH_2O$, 99.95% metals basis) and iridium (IV) chloride hydrate ($IrCl_4 \cdot xH_2O$, 99.9% metals basis, Ir 56.0% min) were purchased from Aladdin. Nafion (5 wt% in lower aliphatic alcohols and water, contains 15–20% water) was purchased from Sigma-Aldrich.

Substrate Ni foam was described as thickness: 2.0 mm, regional density: $350 g m^{-2}$, Saibo. Ti felt was described as thickness: 0.25 mm, porosity: 50–60%, Sinero. Carbon paper (YLS-30T) was described as thickness: 0.235 mm, TORAY. 20% and 75% Pt/C were purchased from Johnson Matthey and Anhui Contango New Energy Technology Co., Ltd., respectively. All the other chemicals were of analytical grade and used as received without further purification. All aqueous solutions were prepared using deionized water with a resistivity of $18.2 MΩ cm^{-1}$.

Synthesis of CoOOH. The CoOOH was synthesized by electrochemical oxidation of the $Co(OH)_2$, which was synthesized via a modified electrochemical deposition method[24]. Electrochemical deposition was performed in a 120 ml electrolytic bath at room temperature. A $1 × 2 cm^2$ Ni foam, a carbon rod, and an Ag/AgCl electrode served as the working, counter, and reference electrodes, respectively. Before the electrochemical deposition, Ni foam was ultrasonically cleaned in 15 mL 3.0 M HCl solution at a 20 mL vial for 10 min (53 kHz, 75%). Then, transfer the Ni foam to a 20 mL vial containing 15 mL deionized water ultrasonically cleaned for 3 s (53 kHz, 75%). The process was repeated five times to remove the Ni ions from the surface of the Ni foam. The solution for electrochemical deposition was a 100 mL aqueous solution containing 1.455 g $Co(NO_3)_2 \cdot 6H_2O$. In a typical synthesis, the Ni foam was subjected to anodic treatment at a current density of $20 mA cm^{-2}$ for 600 s and then applied to a cathodic deposition at a current density of $-20 mA cm^{-2}$ for 600 s. The as-obtained $Co(OH)_2$ was subjected to electrochemical pretreatment under oxidative potentials (ranging from 1.10 to 1.80 V vs RHE) for ten cycles to obtain CoOOH. The mass loadings of CoOOH were $4.0 mg cm^{-2}$ measured by ICP-AES.

Synthesis of $Ru_T/CoOOH$. The $Ru_T/CoOOH$ was synthesized by a modified wet-chemical synthesis strategy[27]. The as-prepared CoOOH were placed in a beaker with 170 mL of deionized water. Then, 30 mL solution containing 16 mg $RuCl_3 \cdot xH_2O$ was injected into the beaker through a micro-injection pump (10 mL/h) under a continued stir (750 rpm). Then, the mixed solution was stirred for 10 h at room temperature. After isolation by centrifugation, the precipitates were washed three times with deionized water and ethanol, respectively. Finally, the product was freeze-dried in a vacuum freeze dryer for 8.0 h. The as-obtained $Ru_T/CoOOH$ was electrochemically pretreated under oxidative potentials before characterization. The mass loadings of $Ru_T/CoOOH$, Co species, and Ru species were about 4.2, 4.0, and $0.2 mg cm^{-2}$ measured by ICP-AES.

Synthesis of $Ru_TIr_V/CoOOH$. The $Ru_TIr_V/CoOOH$ were synthesized by an electrochemical deposition method[28]. The electrochemical deposition was conducted in a standard three-electrode system (CHI 660E, Shanghai CH Instruments), where the as-prepared Ni foam loaded $Ru_T/CoOOH$, a carbon rod, and an Ag/AgCl electrode was used as the working, counter, and reference electrodes, respectively. The working electrode was pretreated using a linear sweep method under a potential ranging from 1.10 V to 1.80 V for five cycles. Then 0.25 mL 100 μM $IrCl_4 \cdot xH_2O$ was added to the electrolyte as the Ir precursor. The mixture was fully mixed under magnetic stirring for 10 min. Then electrochemical deposition was carried out using the linear sweep method from 1.10 V to 1.80 V with a sweep rate of $5 mV s^{-1}$ for one cycle. After the electrochemical deposition, the obtained samples were collected by ultrasonication and washed with ethanol for later electrochemical measurements and characterization. The mass loadings of $Ru_TIr_V/CoOOH$, Co species, Ru species, and Ir species were about 4.4, 4.0, 0.2, and $0.2 mg cm^{-2}$ measured by ICP-AES.

Synthesis of $Ir_V/CoOOH$. $Ir_V/CoOOH$ was synthesized via similar procedures as synthesizing $Ru_TIr_V/CoOOH$ except for changing the support from $Ru_T/CoOOH$ to CoOOH. The mass loadings of $Ir_V/CoOOH$, Co species, and Ir species were about 4.2, 4.0, and $0.2 mg cm^{-2}$ measured by ICP-AES.

Synthesis of $Ir_T/CoOOH$. The synthesis of $Ir_T/CoOOH$ was similar to that of $Ru_T/CoOOH$ except that the single-atom precursor was a

30 mL solution containing 8 mg $IrCl_4 \cdot xH_2O$. The mass loadings of $Ir_T$/CoOOH, Co species, and Ir species were about 4.9, 4.7, and 0.2 mg cm$^{-2}$ measured by ICP-AES.

Synthesis of $Ir_TRu_V$/CoOOH. The synthesis process of $Ir_TRu_V$/CoOOH was similar to that of $Ru_TIr_V$/CoOOH except for changing the support from $Ru_T$/CoOOH to $Ir_T$/CoOOH. In addition, 0.50 mL of 100 μM $RuCl_3 \cdot xH_2O$ was used as the Ru single-atom precursor. The mass loadings of $Ir_TRu_V$/CoOOH, Co species, Ir species, and Ru species were about 5.1, 4.7, 0.2, and 0.2 mg cm$^{-2}$ measured by ICP-AES.

XAFS measurements. XAFS spectra at Ru K-edge and Ir L$_3$-edge were obtained at the beamline 1W1B of Beijing Synchrotron Radiation Facility (BSRF, Beijing) under fluorescence mode. XAFS spectra at Co K-edge were obtained at the beamline BL11B of the Shanghai Synchrotron Radiation Facility (SSRF, Shanghai) under transmission mode. The energies of Ru, Ir, and Co were calibrated according to the absorption edge of pure Ru foil, Ir foil, and Co foil, respectively. Athena and Artemis codes were used to extract the data and fit the profiles. For the XANES spectra, the experimental absorption coefficients as a function of energies $\mu(E)$ were processed by background subtraction and normalization procedures. We refer to this process as 'normalized absorption'. For the Ru K-edge EXAFS, the Fourier-transformed data in R space were analyzed by applying the first-shell approximation to the Ru-O shell and the metallic Ru model to the Ru-Ru shell. For the Ir L$_3$-edge EXAFS, the Fourier-transformed data in R space were analyzed by applying the first-shell approximation to the Ir-O shell and the metallic Ir model to the Ir-Ir shell. The determined factors were fixed for further analysis of the measured samples. Other parameters such as CNs and bond distance around the absorbed atoms were allowed to vary during the fitting process. XAS spectra at Co L-edge were measured at the beamline BL12B (Soochow Beamline for Energy Materials) of the National Synchrotron Radiation Laboratory (NSRL, Hefei).

Electrochemical measurements. An electrochemical workstation (CHI 660E, Shanghai CH Instruments) equipped with a current amplifier (CHI 680D, Shanghai CH Instruments) was used to evaluate the electrocatalytic properties of the catalysts. The 1.0 M KOH electrolyte used for electrochemical measurements was prepared before the experiments. Specifically, 56 g of pure KOH was transferred into a beaker. Then, adding 500 mL of deionized water to the beaker and stirred until KOH is completely dissolved. Subsequently, transfer the KOH solution to a volumetric flask. Bring the solution to the 1.0 L mark with deionized water and shake the flask to ensure adequate mixing. Electrochemical measurements can be performed after the electrolyte cooled down. The pH of the KOH electrolyte was measured using a pH meter, and three independent measurements provided a consistent pH value of 14.0. The electrocatalytic measurements were conducted in a standard three-electrode system at room temperature. The Ni foam loaded with the as-obtained catalysts (1 × 1 cm$^2$) was used as the working electrode. A carbon rod was used as the counter electrode. An Ag/AgCl electrode was used as the reference electrode. The rotation rate during the measurements was 1800 rpm. Potentials were measured against the Ag/AgCl electrode and converted to reversible hydrogen electrode (RHE) scale by $E$ (V vs RHE) = $E$ (V vs Ag/AgCl) + 0.197 V + 0.0591 pH V. In the given equation, 0.197 V was obtained by calibration with respect to the reversible hydrogen electrode (RHE). The calibration was carried out in a three-electrode system using a high-purity hydrogen-saturated 0.5 M $H_2SO_4$ electrolyte, where a Pt wire, another Pt wire, and an Ag/AgCl electrode were used as the working, counter, and reference electrodes, respectively. The calibration was conducted using a cyclic voltammetry method with a sweep rate of 1 mV s$^{-1}$. The average of the two potentials at which the current crossed zero was taken to be the thermodynamic potential for the hydrogen electrode reactions. For electrochemical evaluation using Hg/HgO as the reference electrode, potentials were measured against the Hg/HgO electrode and converted to reversible hydrogen electrode (RHE) scale by $E$ (V vs RHE) = $E$ (V vs Hg/HgO) + 0.098 V + 0.0591 pH V.

The polarization curves of OER were obtained using a linear sweep voltammetry method with a sweep rate of 5 mV s$^{-1}$ in oxygen-saturated 1.0 M KOH electrolyte. The potentials were corrected to compensate for the effect of solution resistance, which were calculated by the following equation: $E_{iR\text{-corrected}} = E$ (V vs RHE) − $iR$, where $i$ is current, and $R$ is the uncompensated ohmic electrolyte resistance. In oxygen-saturated 1.0 M KOH, $R$ is measured as 0.75 Ω via high-frequency alternating current impedance.

The mass activities were obtained by normalizing the current against the mass loadings of Ru and Ir single atoms on the samples. To prepare the working electrode for the ECSAs measurements, 5 mg of the catalysts, 0.8 mL of $H_2O$, 0.4 mL of ethanol, and 80 μL of Nafion were uniformly mixed under ultrasonication. Then 5 μL of the above mixture was cast on the glassy carbon electrode as the working electrode. A carbon rod and an Ag/AgCl electrode were used as the counter and reference electrodes. ECSAs were acquired according to the equation: ECSAs = $R_f$ * $S$, where $R_f$ is the roughness factor; $S$ is the geometric area of the glassy carbon electrode, which is 0.07 cm$^{-2}$ in this work. $R_f$ was determined by $R_f$ = $C_{dl}$ / 60 μF cm$^{-2}$ based on the double-layer capacitance ($C_{dl}$) of a smooth oxide surface. $C_{dl}$ was estimated by plotting the $\Delta j$ ($j_a - j_c$) at 0.48 V vs RHE against scan rates of 20, 40, 60, 80, and 100 mV s$^{-1}$. $\Delta j$ was acquired by cyclic voltammetry (CV) measurement under potential windows of 0.42 ~ 0.54 V vs RHE. $j_a$ and $j_c$ respond to the highest and lowest current density values at 0.48 V, respectively. The potentials of CV curves were provided with $iR$-compensation, $R$ was measured to be 9.0 Ω via high-frequency alternating current impedance. The specific activities were obtained by normalizing the current densities against ECSAs. Tafel slope ($b$) was determined by fitting polarization curves data to the Tafel equation: $\eta = a + b \log |j|$, where $\eta$ is the overpotential for the OER and $j$ is the current density at the given overpotential. For the EIS tests, the working electrode was prepared via a similar procedure as for the ECSAs measurements. A carbon rod and an Ag/AgCl electrode were used as the counter and reference electrodes. The EIS tests were conducted at 1.63 V. The amplitude of the sinusoidal wave was 5 mV and the frequency scan range was 100 kHz-0.01 Hz.

For the stability test at a current density of 10 and 100 mA cm$^{-2}$, Ni foam loaded with the as-obtained $Ru_TIr_V$/CoOOH was used as the working electrode, a carbon rod was used as the counter electrode, and a Hg/HgO electrode was used as the reference electrode. All stability tests were conducted in 1.0 M KOH at room temperature. The dissolved Ru and Ir species were quantified using ICP-MS. We measured the mass loading of Ru and Ir species in the pristine $Ru_TIr_V$/CoOOH and conducted stability test for this sample in 100 mL 1.0 M KOH electrolyte at a current density of 10 mA cm$^{-2}$. During this period, 5 mL of the electrolyte was collected every 50 h for ICP-MS analysis to quantify the dissolved Ru and Ir species. After each collection, 5 mL of 1.0 M KOH was added to replenish the electrolyte volume. The dissolution fraction of Ru and Ir species was obtained by comparing the mass of the dissolved Ru and Ir species with the initial mass of Ru and Ir species in pristine catalysts.

AEM water electrolyzer and MEA water electrolyzer tests. For the AEM water electrolyzer measurements, the anode electrode was prepared by spraying $Ru_TIr_V$/CoOOH inks on a 1.0 × 1.0 cm$^2$ Ti felt, the spraying was controlled to achieve a mass loading of 3.0 mg cm$^{-2}$. For the cathode electrode, 20% Pt/C was sprayed on a 1.0 × 1.0 cm$^2$ Ti felt with a mass loading of 3.0 mg cm$^{-2}$. The Ti felt was pretreated in 3.0 M HCl solution and deionized water before use. An anion exchange membrane (Sustanion® x37-50-grade 60) with area of 2.0 × 2.0 cm$^2$ and thickness of 50 μm was used to separate the anode and cathode compartments of the AEM electrolyzer. Before the AEM water electrolyzer measurements, the anion exchange membrane was treated in oxygen-saturated 1.0 M KOH electrolyte at a temperature of 80 °C for one day, then assembled it into the AEM water electrolyzer. The measurements were conducted in an oxygen-saturated 1.0 M KOH

electrolyte at a temperature of 80 °C. The cell voltage of the AEM water electrolyzer measurements was recorded without *iR*-compensation. For the MEA water electrolyzer test, $Ru_TIr_V/CoOOH$ served as the anode catalyst, 75% Pt/C was employed as the cathode catalyst, and MTCP-50 was utilized as the anion exchange membrane[57]. The thickness of the MTCP-50 was 40 μm. To prepare the anode and cathode inks, catalysts were dispersed to a mixture of isopropanol, deionized water, and Nafion. After ultrasonicated for 30 min, a uniform catalyst ink was obtained. Then, the anode and cathode catalysts were directly air sprayed on the two sides of the MTCP-50 with a geometric area of $2.0 \times 2.0$ cm² in the ultrasonic spray coating system. The anode and cathode catalysts loading were controlled to be 2.0 mg cm⁻² and 0.5 mg cm⁻², respectively. Finally, the $Ru_TIr_V/CoOOH$-coated membranes were hot pressed at 500 kPa for 3 min at a temperature of 80 °C. Before the test, 80 °C KOH was cycled in the MEA water electrolyzer tests for 12 h to activate MTCP-50. The MEA water electrolyzer test was conducted in an oxygen-saturated 1.0 M KOH electrolyte at a temperature of 80 °C. The cell voltage of the MEA water electrolyzer test was recorded without *iR*-compensation.

In-situ XAFS measurements. In-situ XAFS spectra at Ru *K*-edge and Ir *L₃*-edge were obtained at the beamline BL14W1 of the SSRF. We performed the experiments in a specialized in-situ XAFS electrolytic cell by using a three-electrode standard electrochemical workstation. 20 mg of the $Ru_TIr_V/CoOOH$, 2 mL of ethanol, and 40 μL of Nafion were uniformly mixed under ultrasonication. Then the above mixture was sprayed on the carbon paper ($2.0 \times 2.0$ cm²) as the working electrode and then sealed in the cell by Kapton film. Pt wire and Ag/AgCl electrodes were used as the counter and reference electrodes, respectively. Before the data collection, a series of potentials were applied to the electrode for 5 min, respectively. All XAFS data were collected during one period of beam time and each spectroscopy was recorded for 12 min. In-situ Co *K*-edge XAFS experiments were conducted on a TableXAFS-500A from Anhui Chuangpu Instrument Technology Co., LTD. The working electrode was prepared via a similar procedure as for the in-situ XAFS spectra at Ru *K*-edge and Ir *L₃*-edge, except that 30 mg of the $Ru_TIr_V/CoOOH$ was sprayed on the carbon paper ($2.0 \times 2.0$ cm²). The monochromatized X-ray beam was provided by an X-ray tube and a spherically bent crystal assembled on the R250 mm Rowland circle. All the spectra were recorded in transmission mode.

In-situ ATR-SEIRAS spectroscopy. In-situ ATR-SEIRAS spectra were measured on a Fourier transform infrared spectrometer (Thermo Fisher IS50) with a Si crystal as the infrared transmission window in a specialized ATR unit. 10 mg of the samples and 20 μL of Nafion were dispersed in 2 mL of ethanol under ultrasonication for 1 h. Then the mixture was sprayed onto the Au-coated Si crystal to completely cover the Au film. The prepared prism was used as the working electrode after being dried naturally. A Pt wire and an Ag/AgCl electrode were used as counter and reference electrodes, respectively. All electrochemical tests were measured in 1.0 M KOH electrolyte and controlled by a CHI660E electrochemical workstation at room temperature. The background spectra of the working electrode were obtained at OCP before the tests with a resolution of 8 cm⁻¹ for 64 scans in the ATR unit at room temperature.

In-situ ¹⁸O isotope-labeling DEMS measurements. The experiments were conducted using an in-situ differential electrochemical mass spectrometer provided by Hiden Analytical. The catalyst ink was directly dropped onto a carbon paper with a mass loading of 2 mg cm⁻². Two steps of DEMS experiments using $H_2^{18}O$ and $H_2^{16}O$ as the supporting electrolyte were designed (1.0 M KOH). The $Ru_T/CoOOH$, $Ir_V/CoOOH$, $Ir_TRu_V/CoOOH$, and $Ru_TIr_V/CoOOH$ were labeled with ¹⁸O isotopes by conducting cyclic voltammetry in the $H_2^{18}O$ electrolyte within the potential range of 1.0 V ~ 2.0 V vs RHE at a scan rate of 10 mV s⁻¹. Subsequently, the samples were rinsed with abundant water and then operated in the $H_2^{16}O$ electrolyte. Cyclic voltammetry cycles within the potential range of 1.0 V ~ 2.0 V vs RHE at a scan rate of

10 mV s⁻¹ were carried out with the simultaneous detection of signals for ³⁶O₂ (¹⁸O¹⁸O), ³⁴O₂ (¹⁸O¹⁶O), and ³²O₂ (¹⁶O¹⁶O), respectively.

In-situ Raman spectroscopy. The Raman spectra were carried out on a confocal microscope Raman system (Horiba LabRAM HR Evolution). The excitation wavelength was 532.1 nm. A specialized Raman cell was used for the in-situ electrochemical Raman measurements. Carbon paper ($2.0 \times 2.0$ cm²) loaded $Ru_TIr_V/CoOOH$, a Pt wire, and an Ag/AgCl electrode were used as the working, counter, and reference electrodes, respectively. The working electrode was prepared using a procedure similar to the in-situ XAFS measurements. Before the data collection, a series of potentials (OCP ~ 1.65 V) were applied to the electrode for 5 min, respectively.

DFT calculations. The density-functional theory (DFT) calculations were performed by utilizing the Cambridge Sequential Total Energy Package (CASTEP) code based on the plane-wave pseudopotential method[58,59]. Nonlocal exchange and correlation energies were treated with the Perdew-Burke-Ernzerhof functional, which was based on the generalized gradient approximation (GGA)[60]. To separate the self-interaction effects, a vacuum space of 15 Å along the z direction was applied. The Brillouin-zone integration is sampled by a $3 \times 2 \times 1$ k-mesh. The long-range van der Waals interaction is described by the DFT-D2 approach. A cutoff energy of 490 eV was applied for plane-wave expansion. The convergence thresholds for atomic forces and energy were set to 0.05 eV/Å and $2 \times 10^{-5}$ eV, respectively. According to previous studies, $\Delta G$ was computed using a computational hydrogen electrode model[61,62]. For each step of the OER, $\Delta G$ was acquired via the formula $\Delta G = \Delta E + \Delta E_{zpe} - T\Delta S$, where $\Delta E$, $\Delta E_{zpe}$, and $\Delta S$ are the changes in DFT energy, zero-point energy, and entropy, respectively. $T$ was set at 298 K in this work.

Theoretical evaluation of activity. It was assumed that the theoretical overpotentials of $Ru_T/CoOOH$ and $Ru_TIr_V/CoOOH$ followed the conventional OER mechanism. Referring to previous studies, the computational hydrogen electrode model was used to express the chemical potentials of protons and electrons at a given pH and applied potential[63,64]. Under alkaline conditions, the elementary steps during the OER process involve the formation of adsorbed OH, O, and OOH species on the surface (*) according to the following steps:

$$OH^- + * \rightarrow *OH + e^- \tag{1}$$

$$*OH + OH^- \rightarrow *O + H_2O + e^- \tag{2}$$

$$*O + OH^- \rightarrow *OOH + e^- \tag{3}$$

$$*OOH + OH^- \rightarrow * + O_2 + H_2O + e^- \tag{4}$$

Due to the thermodynamic equivalence of the OER process under alkaline and acidic conditions, we modeled the thermochemistry of OER under acidic conditions[63]. Then, steps 1-4 were modified as:

$$H_2O + * \rightarrow *OH + H^+ + e^- \tag{5}$$

$$*OH + H_2O \rightarrow *O + H_2O + H^+ + e^- \tag{6}$$

$$*O + H_2O \rightarrow *OOH + H^+ + e^- \tag{7}$$

$$*OOH + H_2O \rightarrow * + O_2 + H_2O + H^+ + e^- \tag{8}$$

Thus, the Gibbs free energy change for steps 5-8 can be expressed as:

$$\Delta G_1 = \Delta G_{*OH} - eU + \Delta G_{H+}(pH) \tag{9}$$

$$\Delta G_2 = \Delta G_{*O} - \Delta G_{*OH} - eU + \Delta G_{H+}(pH) \tag{10}$$

$$\Delta G_3 = \Delta G_{*OOH} - \Delta G_{*O} - eU + \Delta G_{H+}(pH) \tag{11}$$

$$\Delta G_4 = 4.92[eV] - \Delta G_{*OOH} - eU + \Delta G_{H+}(pH) \tag{12}$$

where $U$ represents for applied external bias $U$; $\Delta G_{H+}(pH) = -k_B T \ln(10) \times pH$ is the free energy change at a nonzero pH value. Because the $O_2$ bond energy is difficult to determine by DFT calculations, the sum of $-\Delta G_{1\text{-}4}$ was fixed to the experimental Gibbs free energy of $-4.92$ eV for forming two water molecules. The Gibbs free energy corrections of *OH, *O, and *OOH intermediates include zero-point energy (ZPE) and entropy corrections according to $\Delta G = \Delta E + ZPE - T\Delta S$. The entropy corrections of *OH, *O, and *OOH were set as 0.35, 0.05, and 0.40, respectively, using the harmonic approximation[65]. The theoretical overpotential ($\eta$) was then defined as:

$$\eta = \max(\Delta G_1, \Delta G_2, \Delta G_3, \Delta G_4)/e - 1.23\,[V] \tag{13}$$

**Instrumentations.** XRD patterns were recorded using a Philips X'Pert Pro Super diffractometer with Cu-Kα radiation ($\lambda = 1.54178$ Å). HAADF-STEM images were carried out on a Thermo Fisher Scientific Themis Z transmission electron microscope using Mo-based TEM grids. EDX elemental mapping images were carried out on an FEI Talos F200X field-emission transmission electron microscope operating at an accelerating voltage of 200 kV using Mo-based TEM grids. TEM images were carried out in a JEOL 7700 field-emission electron microscope using Mo-based TEM grids. XPS measurements were performed on a Kratos AXIS SUPRA+ with Al Kα = 1486.6 eV as the exciting source. ICP-AES analyses were used to determine the mass loadings of metal species on an Atomscan Advantage, Thermo Jarrell Ash, USA. In-situ DEMS experiments were performed using an in-situ differential electrochemical mass spectrometer supplied by Hiden Analytical.

## Data availability
The data that support the findings of this work are available within the manuscript, Supplementary Information files, and Source Data File. Source data are provided with this paper.

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

## Acknowledgements

This work was supported by National Natural Science Foundation of China (12241502) [J.B.], (22202188) [J.X.], and (22202192) [Z.Z.]); Fundamental Research Funds for the Central Universities (20720220010) [J.B.]; National Key Research and Development Program of China (2019YFA0405602) [J.B.]; USTC Research Funds of the Double First-Class Initiative (YD9990002016) [Z.Z.]; Anhui Natural Science Foundation for Young Scholars (2208085QB52) [Z.Z.]; Fellowship of China Postdoctoral Science Foundation (2023M743372) [P.M.] and (2024M753119) [J.L.]; Postdoctoral Fellowship Program of CPSF (GZC20232537) [P.M.]. The authors express their gratitude to Prof. Tongwen Xu for supplying the anion exchange membrane MTCP-50 utilized in the MEA water electrolyzer tests. This work was partially carried out at the Instruments Center for Physical Science, University of Science and Technology of China.

## Author contributions

Z.Z. and J.B. designed the study. P.M. and H.C. conducted the experiments. J.L. and R.W. processed the XAFS results. M.Z. conducted HAADF-STEM analysis. J.X. carried out DFT calculations. P.M., Z.Z. and J.B. wrote the paper. All authors discussed the results and contributed to the manuscript.

## Competing interests

The authors declare no competing interests.

## Additional information

Zhirong Zhang or Jun Bao.

**Peer review information** *Nature Communications* thanks M. Gu and the
other anonymous reviewer(s) for their contribution to the peer review of
this work. A peer review file is available.

