## [Transparent Peer Review file · Nature Communications]

Site-specific synergy in heterogeneous single atoms for efficient oxygen evolution

Corresponding Author: Professor Jun Bao

Version 0:

Reviewer comments:

Reviewer #1

(Remarks to the Author)

Ma et al. fabricated heterogeneous single-atom catalyst RuTlIrV/CoOOH by selectively anchoring Ru single atoms onto three-fold fcc hollow sites and Ir single atoms onto VO sites. A series of in-situ spectroscopic characterizations and mechanistic studies indicated that the Ru single atoms at the three-fold fcc hollow sites acted as adsorption sites and Ir single atoms at VO sites stabilized the *OOH intermediates on the Ru single atoms via hydrogen bonding interactions. The performance appears to be OK, however, there are issues with the characterization and discussion on the active species looks unclear. Therefore, before the reviewer can recommend the work for publication, the following some concerns need to be addressed.

1 According to experimental procedure, the authors used Ni foam as the framework, then why is CoOOH formed instead of CoNiOOH? The author should provide the LSV curve of pure Ni foil in the electrolyte as a reference. NiOOH itself is an active catalyst.

2 Is there a specific synthesis condition that promoted the formation of CoOOH nanosheet morphology?

3 The authors emphasized the importance of the site-specific synergy in heterogeneous single atoms, however, the synergistic effect between Ru, Ir and CoOOH is not well demonstrated, and the discussion is not sufficient.

4 As demonstrated in Figure 1f and g, there seemed to be more single atoms of Ir than Ru in CoOOH nanosheets, how did the authors explain this?

5 The simulation part of the manuscript remains a major concern. Only the atomic structural and electrochemical properties was discussed in detail for RuTlIrV/CoOOH, RuT/CoOOH, IrV/CoOOH. The atomic structural of IrTRuV/CoOOH should be analyzed for comparison.

6 The ratio of three-fold fcc hollow sites (Ru SA) and oxygen vacancy (Ir SA) should be quantified and systematically studied. How can this ratio be affected by reaction condition? And how can this ratio affect the ultimate electro-catalytic performance?

7 It is well acknowledged that the Ag/AgCl electrode (and every KCl-based reference electrode) is not appropriate for electrochemical experiments in alkaline media, as the electrolyte OH⁻ ions diffuse into the reference electrode changing its composition during measurements. Thus, its potential does not remain constant, while the leached Cl⁻ can contaminate the electrode surface. You should have either used Hg/HgO reference electrode or immersed the SCE in a Luggin capillary for protection. Was the OCV stable, and were the measurements reproducible during your experiments?

8 The LSVs should be corrected for the uncompensated ohmic resistance for a reliable kinetics analysis. Furthermore, you can propose possible reaction pathways based on the extracted Tafel slopes.

9 The dissolution of Ru is a common problem in Ru based electrocatalysts especially during OER process. but the corresponding content is not discussed in this manuscript.

Reviewer #2

(Remarks to the Author)

I have carefully reviewed the manuscript titled "Site-specific synergy in heterogeneous single atoms for efficient oxygen evolution" submitted to Nature Communications. While the authors present an interesting study on heterogeneous single-atom catalysts for the oxygen evolution reaction, I have several major concerns that preclude publication of this work in its current form. My recommendation is to reject this manuscript for the following reasons:

1. Lack of direct structural evidence: The entire premise of this work relies on the authors' claim that in the RuTlIrV/CoOOH

sample, "Ru single atoms are onto three-fold fcc hollow sites and Ir single atoms are onto oxygen vacancy (VO) sites". However, no direct visual evidence, such as atomic-resolution HAADF-STEM images, is provided to unambiguously confirm this specific atomic arrangement. The authors rely heavily on EXAFS data interpretation, which can be subjective and model-dependent. In addition, the manuscript fails to clearly explain how the EXAFS results definitively prove the claimed atomic configuration.

2. Ambiguous in-situ spectroscopic analysis: In the in-situ XAFS measurements, the authors observe an increase in Ru-O coordination numbers from 5.0 to 5.4 under applied potential. While they interpret this as evidence for adsorption of oxygenated reaction intermediates, alternative explanations are not adequately addressed or ruled out. This change could potentially arise from other structural modifications or experimental artifacts. A more rigorous analysis considering multiple scenarios is necessary.

3. Inconsistencies in structural characterization: There are discrepancies between the microscopy and diffraction data presented. TEM images (e.g., Fig. 1f, 1g, S6, S11) suggest a relatively crystalline substrate, yet the corresponding XRD patterns show weak, broad peaks. This inconsistency is not addressed in the manuscript. Furthermore, some XRD patterns (particularly the (002) peak in Fig. S3) do not align well with the reference CoOOH pattern (PDF #26-1107). These discrepancies raise questions about the true nature and uniformity of the support material.

4. Overinterpretation of DFT results: The computational studies appear to be based on idealized models that may not accurately represent the experimental system, given the uncertainties in the actual atomic structure. Without a more robust structural characterization, the relevance and validity of the DFT calculations are questionable.

5. Limited comparison with state-of-the-art catalysts: While the authors claim superior performance, a comprehensive comparison with other leading OER catalysts in the field is lacking. This makes it difficult to assess the true significance of the reported activity improvements.

6. Insufficient mechanistic insights: The proposed synergistic effects between Ru and Ir atoms are not thoroughly elucidated. The manuscript lacks a clear, molecularly-detailed mechanism explaining how the specific atomic arrangements lead to enhanced catalytic activity.

7. Concerns about reproducibility: Given the complexity of the synthesis process and the sensitivity of single-atom catalysts to preparation conditions, more detailed information on the reproducibility of the catalyst synthesis and performance is needed.

In light of these significant concerns, I recommend that the manuscript be rejected in its current form. Substantial additional experimental evidence, particularly direct atomic-scale imaging and more rigorous spectroscopic analyses, would be required to support the authors' claims. A major revision addressing these points might be considered for resubmission, but the current work falls short of the standards expected for publication in Nature Communications.

Reviewer #3

(Remarks to the Author)

The work by P. Ma et al. reports a relatively novel method to construct heterogeneous single-atom systems and reveals the site-specific synergy in heterogeneous single atoms for oxygen evolution. The as-formed catalysts show decent OER activity and long durability. However, I have concerns regarding numerous explanations throughout the paper that appear to be illogical or not sufficiently justified. Considering rather unclear expression of reaction mechanism design and performance, this manuscript cannot be accepted for publication before the following issues have been addressed. Please find the specific comments below:

1. The paper asserts the site-specific synergy and claims that RuTIrV/CoOOH shows the better OER performance than IrTRuV/CoOOH. However, the studies about IrTRuV/CoOOH are relatively deficient. Whether it is more suitable to compare the difference between RuTIrV/CoOOH and IrTRuV/CoOOH for the topic of "site-specific", rather than do excessive efforts on RuT/CoOOH and IrV/CoOOH samples.

2. For EXAFS fitting, usually, for the fitting of transition metals in the first shell, the inner potential correction is generally $-10 \text{ eV} \leq \Delta E_0 \leq 10 \text{ eV}$; the disorder degree (σ^2) is generally considered to be acceptable at 0.003-0.007; also, the amplitude reduction factor (S_0^2) should be firstly provided. Therefore, it is recommended that the authors check the data in Supplementary Tables and refit EXAFS more carefully.

3. Please add some values in Fig. 3e for intuitively evaluating the OER performance. In addition, please consider ECSA- or BET- corrected performance for the further comparisons.

4. There are no obvious potential decays can be observed in Figs. 3f and 3i, please extend the operating times.

5. Please provide the table of catalytic activity of reported electrocatalysts for AEM water electrolyzer to compare with other electrocatalysts. Moreover, we suggest that the author adopt the method of catalyst coated membrane (CCM) to fabricate membrane electrode assembly (MEA), which is expected to reduce the internal resistance of proton coupled electron transfer at the catalyst-membrane interface, improving reactor performance.

6. For in-situ EXAFS spectra of Ru, the coordination numbers of Ru-O increased from 5.0 to 5.4 as the applied voltage increased from OCP to 1.65 V, but the average distance between Ru and O (R) keep unchanged, why? In my opinion, under the OER conditions, the average R is comprised of two different types, which is related with Ru atoms connected with the lattice oxygen and Ru atoms connected with intermediates.

7. It is not sufficient to determine the AEM pathway by the evolution of *OOH species from the in-situ ATR-SEIRAS, LOM pathway can co-exist in alkaline water splitting, especially for these kinds of substrates [(MOOH or M(OH)_x]. DEMS is

recommended.

8. The section about mechanistic studies is insufficient. It is recommended that the authors disclose the effects of site-specific on the OER performance more deeply. For example, the difference of the distortions by the heterogeneous atoms' assembly, which is directly influence the efficiency of charge transfer between active sites and reactants. Please pay more attention to the improvements of intrinsic activity induced by the specific atom assemble.

9. Ru single atoms at the three-fold fcc hollow sites serve as adsorption sites for key reaction intermediates. Meanwhile, Ir single atoms at VO sites stabilized the *OOH intermediates on the Ru single atoms via hydrogen bonding interactions. But, the current experiments cannot support this point. What is hydrogen bonding and how to confirm this point? What is the effective distance of hydrogen bonding interaction?

Reviewer #4

(Remarks to the Author)

In this article, the authors introduced a precision synthesis strategy for constructing heterogeneous single-atom catalysts. They proposed that the synergy effect in heterogeneous single atoms was strongly related to the single-atom anchoring sites. The systematic electrochemical evaluation demonstrated that the RuTlrV/CoOOH with Ru single atoms at three-fold fcc hollow sites and Ir single atoms at oxygen vacancies exhibited superior OER performance compared to IrTRuV/CoOOH. Overall experiments including a series of in-situ spectroscopic evidence and mechanistic studies further proved the intrinsic mechanism of the site-specific synergy in heterogeneous single atoms. In our personally opinion, the work is interesting and proposes a new insight into the rational design of highly efficient catalysts applied in OER. We think this manuscript is very worthy of publication in Nature Communications after some revisions. The following are our questions

1. The authors proposed that the Ru and Ir single atoms were selectively anchored at the three-fold fcc hollow sites and oxygen vacancies, respectively. Please discuss the intrinsic mechanism of the selective combination between single atoms and anchoring sites.

2. Multiple characterizations including XAFS, XPS, and XAS demonstrated that the Ru single atoms at three-fold fcc hollow sites exhibited stronger interactions with Co species than Ir single atoms at oxygen vacancies. Please clearly origins of the site-specific interactions.

3. Through a series of in-situ experiments and theoretical calculations, the authors have well explained the intrinsic mechanism for the improved performance of RuTlrV/CoOOH compared to RuT/CoOOH. However, the reason for the weakened synergy in Ir single atoms at the three-fold fcc hollow sites and Ru single atoms at the oxygen vacancies was less clearly explained. It is recommended to discuss the attenuated synergy in detail.

4. The working electrode is not represented in the optical image of the in-situ ATR-SEIRAS experiment (Supplementary Fig. 22b). This optical image should be re-shooting.

5. In-situ Raman spectra of RuTlrV/CoOOH were performed to explore the potential structure evolution of the CoOOH under oxygen evolution conditions. It is recommended the author conduct characterizations such as in-situ XAFS to discover the potential evolution in the valence state of Co species.

6. Whether Ru and Ir single atoms on the RuTlrV/CoOOH surface dissolved into the electrolyte during the OER process.

7. The structure characterizations of RuTlrV/CoOOH after OER should be also added to the manuscript, such as XRD pattern, TEM image, and HAADF-STEM image.

8. The oxygen evolution performance of IrT/CoOOH should be evaluated and added to the manuscript. In addition, the Tafel slopes of IrT/CoOOH and IrTRuV/CoOOH should be compared with RuTlrV/CoOOH.

Version 1:

Reviewer comments:

Reviewer #1

(Remarks to the Author)

I recommend accepting the manuscript.

Reviewer #2

(Remarks to the Author)

The authors have addressed the comments of reviewers properly. It is recommended for publication now.

Reviewer #3

(Remarks to the Author)

It can be accepted as it is

Reviewer #4

(Remarks to the Author)

My concern is well addressed in the revised version. Personally, I think it could be published at current state.

Point-by-point response to reviewer comments

Manuscript Number: NCOMMS-24-57803

Manuscript Type: Article

Title: “Site-specific synergy in heterogeneous single atoms for efficient oxygen evolution”

Authors(s): Peiyu Ma, Jiawei Xue, Ji Li, Heng Cao, Ruyang Wang, Ming Zuo, Zhirong Zhang, Jun Bao

Corresponding authors: Zhirong Zhang, Jun Bao

Reviewer #1 (Remarks to the Author):

Ma et al. fabricated heterogeneous single-atom catalyst Ru_TIr_V/CoOOH by selectively anchoring Ru single atoms onto three-fold fcc hollow sites and Ir single atoms onto V_O sites. A series of *in-situ* spectroscopic characterizations and mechanistic studies indicated that the Ru single atoms at the three-fold fcc hollow sites acted as adsorption sites and Ir single atoms at V_O sites stabilized the *OOH intermediates on the Ru single atoms via hydrogen bonding interactions. The performance appears to be OK, however, there are issues with the characterization and discussion on the active species looks unclear. Therefore, before the reviewer can recommend the work for publication, the following some concerns need to be addressed.

Response:

We sincerely thank the reviewer for the positive comments and valuable suggestions on the manuscript, which certainly helped to improve its quality. We have carefully considered the comments and suggestions to revise the manuscript accordingly. Firstly, the content of Ni species in CoOOH was quantified by inductively coupled plasma-atomic emission spectrometry (ICP-AES) to clarify the components of synthesized samples. The OER activity of Ni foam was also evaluated. In addition, we investigated the effects of deposition solution concentration, current density, and time on CoOOH morphology by modulating these conditions, respectively. The synergistic effect between Ru, Ir and CoOOH was also elucidated. Additionally, we investigated the effect of the ratio of Ru and Ir single atoms on the OER performance of the Ru_TIr_V/CoOOH. The atomic structure of Ir_TRu_V/CoOOH was also discussed in detail. Subsequently, we illustrated the OER mechanism of the catalyst using *in-situ* ¹⁸O isotope-labeling differential electrochemical mass spectrometry (DEMS) and probed the dissolution of Ru species during OER. We believe that the improvements have strengthened the manuscript and further highlighted the significance of this work.

1. According to experimental procedure, the authors used Ni foam as the framework, then why is CoOOH formed instead of CoNiOOH? The author should provide the LSV curve of pure Ni foil in the electrolyte as a reference. NiOOH itself is an active catalyst.

Response:

We sincerely thank the reviewer for the insightful questions. In this manuscript, CoOOH was

synthesized by electrochemical oxidation of the Co(OH)_2 . The Co(OH)_2 was synthesized on Ni foam using a modified electrochemical deposition method. Before the electrochemical deposition procedure, the Ni foam was ultrasonically cleaned for 10 min in 3.0 M HCl solution to remove the surface NiO. Then, the Ni foam was subjected to a cleaning procedure in deionized water by ultrasonication. This process was repeated five times to nearly completely remove the Ni ions from the surface of the Ni foam. To measure the Ni species content in CoOOH, we conducted ICP-AES. The result showed that the Ni species content in CoOOH was about 0.07 at%, indicating trace amounts of Ni species present in the CoOOH. To simplify, the sample was labeled as CoOOH.

The LSV curve of pure Ni foam was also measured and added to the manuscript as a reference. The polarization curves were recorded in a standard three-electrode system using a 1.0 M KOH electrolyte. As shown in **Fig. R1a**, the Ni foam exhibited poor electrocatalytic activity towards OER. Specifically, the overpotential required to reach a current density of 10 mA cm^{-2} was 370 mV for Ni foam, which was 70 mV, 150 mV, 160 mV, and 190 mV higher than that of CoOOH, Ru_T/CoOOH , Ir_V/CoOOH , and $\text{Ru}_T\text{Ir}_V/\text{CoOOH}$, respectively (**Fig. R1b**). The above results elucidated that the main contributor to catalyst performance was the CoOOH-supported single-atom catalysts. The correlated data and discussions have been added to the revised manuscript.

Fig. R1 | Electrocatalytic performance towards oxygen evolution. a. Polarization curves of samples towards oxygen evolution in 1.0 M KOH electrolyte. Ni foam served as a reference. **b.** Overpotentials of Ni foam, CoOOH, Ru_T/CoOOH , Ir_V/CoOOH , and $\text{Ru}_T\text{Ir}_V/\text{CoOOH}$ at a current density of 10 mA cm^{-2} .

2. Is there a specific synthesis condition that promoted the formation of CoOOH nanosheet morphology?

Response:

We genuinely thank the reviewer for the valuable comment. In this manuscript, the CoOOH was synthesized by electrochemical oxidation of the Co(OH)_2 . Therefore, the morphology of CoOOH was influenced by the Co(OH)_2 preparation conditions. The Co(OH)_2 was prepared using a modified electrochemical deposition method. The solution for the electrochemical deposition was 0.05 M (1.455 g) $\text{Co(NO}_3)_2 \cdot 6\text{H}_2\text{O}$. In a typical synthesis, the substrate Ni foam was first

subjected to anodic treatment at a current density of 20 mA cm^{-2} for 600 s and then applied to a cathodic deposition at a current density of -20 mA cm^{-2} for 600 s. Subsequently, the as-obtained sample was electrochemically pretreated under oxidative potentials to obtain CoOOH. Consequently, the main factors affecting the morphology of CoOOH were the solution concentration during electrochemical deposition, deposition current density, and deposition time. To investigate the effect of these factors on the morphology of CoOOH, we modulated these factors, respectively.

Firstly, the concentration of deposition solution was adjusted to 0.025 M (0.7275 g) and 0.1 M (2.910 g) of $\text{Co}(\text{NO}_3)_2 \cdot 6\text{H}_2\text{O}$, respectively, while maintaining constant deposition current density and deposition time. As shown in the transmission electron microscopy (TEM) images, the CoOOH prepared in 0.025 and 0.05 M $\text{Co}(\text{NO}_3)_2 \cdot 6\text{H}_2\text{O}$ solution showed nanosheet morphology (**Fig. R2a** and **2b**). Nevertheless, nanoplates formed for CoOOH prepared in 0.1 M $\text{Co}(\text{NO}_3)_2 \cdot 6\text{H}_2\text{O}$ solution (**Fig. R2c**). In addition, the mass loadings of CoOOH on Ni foam were measured to be 1.5, 4.0, and 5.1 mg cm^{-2} for CoOOH synthesized in 0.025, 0.05, and 0.1 M $\text{Co}(\text{NO}_3)_2 \cdot 6\text{H}_2\text{O}$ solution, respectively.

Fig. R2 | Morphology characterization of CoOOH synthesized using different deposition solution concentrations. a. b. c. TEM images of CoOOH synthesized in solutions with $\text{Co}(\text{NO}_3)_2 \cdot 6\text{H}_2\text{O}$ concentrations of 0.025 M (**a**), 0.05 M (**b**), and 0.1 M (**c**).

Subsequently, the anodic and cathodic deposition current density was regulated to 10 and 40 mA cm^{-2} , respectively, while maintaining constant deposition solution concentration and deposition time. The CoOOH prepared at the deposition current density of 10 and 20 mA cm^{-2} showed nanosheet morphology (**Fig. R3a** and **3b**). When the deposition current density was increased to 40 mA cm^{-2} , a nanoplate morphology appeared in CoOOH (**Fig. R3c**). The mass loadings of CoOOH on Ni foam were measured to be 1.4, 4.0, and 5.3 mg cm^{-2} for CoOOH synthesized under current density at 10, 20, and 40 mA cm^{-2} , respectively.

Fig. R3 | Morphology characterization of CoOOH synthesized under different current density. a. b. c. TEM images of CoOOH synthesized at the anodic and cathodic deposition current density of 10 mA cm^{-2} (a), 20 mA cm^{-2} (b), and 40 mA cm^{-2} (c).

Afterward, the anodic and cathodic deposition time was modulated to 300 s and 1200 s, respectively, while maintaining constant deposition solution concentration and current density. As shown in the TEM images, the CoOOH prepared in anodic and cathodic deposition times of 300 s and 600 s showed nanosheet morphology (Fig. R4a and 4b). Nevertheless, nanoplate morphology emerged for CoOOH at the deposition time of 1200 s (Fig. R4c). The mass loadings of CoOOH on Ni foam were measured to be 1.3, 4.0, and 5.4 mg cm^{-2} for CoOOH synthesized with the deposition time of 300, 600, and 1200 s, respectively.

Fig. R4 | Morphology characterization of CoOOH synthesized at different deposition time. a. b. c. TEM image of CoOOH synthesized at the anodic and cathodic deposition time of 300 s (a), 600s (b), and 1200 s (c).

The experimental results elucidated the morphology of CoOOH was influenced by the concentration of the electrochemical deposition solution, the current density during deposition, and the deposition time. At lower deposition solution concentration, current density, and shorter deposition time, CoOOH predominantly exhibited a nanosheet morphology. However, under these conditions, the relatively low mass loading of CoOOH was detrimental to the OER performance. When the deposition solution concentration, current density, and deposition time were increased, nanoplates formed in the synthesized CoOOH. To ensure the homogeneity of the sample's morphology and maintain an optimal mass loading, the synthesis parameters for the CoOOH in this manuscript were described as follows: concentration of the electrochemical

deposition solution was 0.05 M (1.455 g) $\text{Co}(\text{NO}_3)_2 \cdot 6\text{H}_2\text{O}$, the deposition current density was 20 mA cm^{-2} and the deposition time was 600 s for both anodic and cathodic deposition, respectively.

3. The authors emphasized the importance of the site-specific synergy in heterogeneous single atoms, however, the synergistic effect between Ru, Ir and CoOOH is not well demonstrated, and the discussion is not sufficient.

Response:

We sincerely thank the reviewer for the valuable comment. In this work, the Ru single atoms were anchored at the three-fold fcc hollow sites on Ru_T/CoOOH . The results of XAFS, Co *L*-edge spectra, and XPS revealed electron transfer from Ru single atoms to CoOOH, which could modulate the adsorption behavior of reaction key intermediates (**Fig. 2i, Supplementary Fig. 15, 16**). Remarkably, according to the results of *in-situ* XAFS spectra, the Ru single atoms served as the active sites for the adsorption of reaction key intermediates. Theoretical calculations indicated no significant interaction between the reaction key intermediates on the Ru single atoms and CoOOH (**Fig. 5g**). The above results suggested the synergy between Ru single atoms and CoOOH modulates the adsorption behavior of reaction key intermediates via electronic interactions, rather than altering the configuration of the intermediates.

In the case of Ir_V/CoOOH , the Ir single atoms were anchored at the V_O sites. Based on the results of XAFS spectra, Co *L*-edge XAS spectra, and XPS, the Ir single atoms did not modulate the electronic structure of Co species (**Fig. 2i, Supplementary Fig. 15, 16**). Therefore, the enhanced OER performance of Ir_V/CoOOH cannot be ascribed to electronic interactions between Ir single atoms and Co species. Further theoretical calculations revealed a hydrogen bonding formed between Ir single atoms and the $^*\text{OOH}$ intermediates on the Co sites, which contributed to the enhanced OER performance of CoOOH (**Fig. R5a and 5b**). In the manuscript, the Ir single atoms on $\text{Ru}_T\text{Ir}_V/\text{CoOOH}$ also formed hydrogen bonding with $^*\text{OOH}$ intermediates on the Ru single atoms.

Fig. R5 | Mechanistic studies of Ir_V/CoOOH towards OER. a. The schematic OER pathways of Ir_V/CoOOH . The white, red, purple, and yellow spheres represent H, O, Co, and Ir atoms, respectively. The active sites are highlighted by green circles. The red dashed lines indicated hydrogen bonding. **b.** Free-energy diagrams of Ir_V/CoOOH towards OER.

To assess the influence of the two kind of hydrogen bonding interactions on the OER performance, we evaluated the reaction kinetics of the samples. The Ir_v/CoOOH showed a Tafel slope value of 57 mV dec⁻¹, which was 19 mV dec⁻¹ lower than that of CoOOH (76 mV dec⁻¹). Notably, the Tafel slope value of Ru_TIr_v/CoOOH was 41 mV dec⁻¹, 27 mV dec⁻¹ lower than that of Ru_T/CoOOH (68 mV dec⁻¹). The more significant decrease in the Tafel slope indicated that the hydrogen bonding interactions between Ir single atoms and the *OOH intermediates on the Ru single atoms were more favorable for promoting OER than those formed between Ir single atoms and the *OOH intermediates on the CoOOH surface. The correlated data and discussions have been added to the revised manuscript.

4. As demonstrated in Figure 1f and g, there seemed to be more single atoms of Ir than Ru in CoOOH nanosheets, how did the authors explain this?

Response:

We genuinely thank the reviewer for the insightful comment. In high-angle annular dark-field scanning transmission electron microscopy (HAADF-STEM) images, the brightness of specific atoms is proportional to the square of their atomic number. For Ru_T/CoOOH, the atomic number of Ru atoms (44) was higher than Co atoms (27), thus the spots with higher brightness were Ru atoms and lower brightness were Co atoms. In the case of the Ru_TIr_v/CoOOH, the atomic number of Ir atoms (77) significantly exceeds that of Ru atoms. The substantial variations in atomic number result in significant differences in the relative brightness of Ru and Ir single atoms in HAADF-STEM images. Consequently, when the relatively bright Ir single atoms were clearly visible, the relatively dark Ru single atoms became more difficult to identify. To determine the contents of Ru and Ir species in the samples accurately, we performed ICP-AES measurements on Ru_T/CoOOH and Ru_TIr_v/CoOOH. The results demonstrated the contents of Ru elements was about 4.6 wt% for Ru_T/CoOOH (4.2 at%). In addition, the Ru and Ir elements were measured to be about 4.4 wt% (4.1 at%) and 4.9 wt% (2.4 at%) for Ru_TIr_v/CoOOH, respectively.

5. The simulation part of the manuscript remains a major concern. Only the atomic structural and electrochemical properties was discussed in detail for Ru_TIr_v/CoOOH, Ru_T/CoOOH, Ir_v/CoOOH. The atomic structural of Ir_TRu_v/CoOOH should be analyzed for comparison.

Response:

We honestly thank the reviewer for the valuable comment. We agree that the atomic structure of Ir_TRu_v/CoOOH should be analyzed in detail for comparison. The Ir_TRu_v/CoOOH was prepared by anchoring Ir single atoms onto the three-fold fcc hollow sites and Ru single atoms onto the V_o sites, respectively. TEM image showed the as-prepared sample presented nanosheet morphologies. Furthermore, all characteristic peaks in the X-ray diffraction (XRD) patterns were attributed to CoOOH (PDF #26-1107), indicating the absence of Ir- and Ru-based metals or metal oxides. (**Fig. R6a** and **6b**). HAADF-STEM image of Ir_TRu_v/CoOOH identified isolated bright spots relative to the support, corresponding to Ir or Ru atoms (**Fig. R6c**). Energy-dispersive X-ray (EDX) elemental mapping images showed the uniform distribution of Ir

and Ru elements across the Ir₇Ru_v/CoOOH (**Fig. R6d**). The quantitative analysis by ICP-AES demonstrated that the contents of Ir and Ru elements were 3.9 wt% and 4.0 wt%, respectively.

Fig. R6 | Morphology and structure characterizations of Ir₇Ru_v/CoOOH. a. TEM image. b. XRD pattern. c. HAADF-STEM image. d. EDX elemental mapping.

The detailed electronic structures and coordination environments of Ir and Ru single atoms were investigated by X-ray absorption near-edge spectroscopy (XANES) and extended X-ray absorption fine structure (EXAFS). In the Ir L_3 -edge XANES spectra, the white line intensity of Ir₇Ru_v/CoOOH was higher than that of Ru₇Ir_v/CoOOH, indicating that the valence state of Ir single atoms anchored at three-fold fcc hollow sites was higher than that of anchored at V_O sites (**Fig. R7a**). In the Ir L_3 -edge EXAFS spectra, the Ir₇Ru_v/CoOOH exhibited only one characteristic peak at about 1.6 Å assigned to Ir-O bonding, similar to that of Ru₇Ir_v/CoOOH, which substantiated the isolated dispersion of Ir atoms. EXAFS fitting results demonstrated that the coordination numbers (CNs) of Ir-O for Ir₇Ru_v/CoOOH and Ru₇Ir_v/CoOOH were about 6.0 (**Fig. R7b** and **Table R1**). Therefore, the Ir single atoms at three-fold fcc hollow sites were stabilized by three oxygen atoms of the sites, while the remaining three coordinated oxygen atoms were suspended at its surface as dangling bonds (**Fig. R8a**). By comparison, for Ir single atoms at the V_O sites on Ru₇Ir_v/CoOOH, one apex oxygen of the IrO₆ octahedral structure was inserted into the V_O sites, while four side OH⁻ of the octahedra formed hydrogen bonding with adjacent oxygen atoms on the CoOOH surface to stabilize the structure (**Fig. R8b**).

Fig. R7 | Atomic structural analysis of Ir_TRu_V/CoOOH and Ru_TIr_V/CoOOH. a. XANES spectra at the Ir L_3 -edge of Ir_TRu_V/CoOOH and Ru_TIr_V/CoOOH. **b.** Experimental and fitting EXAFS spectra at the Ir L_3 -edge of Ir_TRu_V/CoOOH and Ru_TIr_V/CoOOH. The inset atomic models are the first-shell coordination of Ir single atoms at three-fold fcc hollow sites. The red and yellow spheres represent O and Ir atoms, respectively. Ir foil and IrO₂ were used as references.

Samples	Path	R (Å)	CNs	σ^2 (10^{-3})	ΔE_0 (eV)	R -factor
Ir _T Ru _V /CoOOH	Ir-O	2.00 ± 0.02	6.0 ± 1.1	6.5	9.8	0.01
Ru _T Ir _V /CoOOH	Ir-O	2.00 ± 0.01	6.0 ± 1.1	7.9	10.0	0.02

Table R1 | Fitting results of Ir L_3 -edge EXAFS spectra for Ir_TRu_V/CoOOH and Ru_TIr_V/CoOOH. R , the distance between the absorber and backscatter atoms. The R value is phase corrected during fitting process; CNs , coordination numbers; σ^2 , Debye-Waller factors; ΔE_0 , the inner potential correction that accounts for the difference in the inner potential between the sample and the references; S_0^2 , the amplitude reduction factor, the S_0^2 for the Ir L_3 -edge EXAFS spectra fitting was determined to be 1.05; R -factor, the goodness of fit.

Fig. R8 | Atomic structural analysis of the Ir single atoms at diverse sites. a. b. Structural models of Ir single atoms at the three-fold fcc hollow sites (a) and V_O sites (b).

Subsequently, we investigated the electronic structures and coordination environments of Ru

single atoms at the V_O sites of $\text{Ir}_T\text{Ru}_V/\text{CoOOH}$. In the Ru K -edge XANES spectra, the absorption edge of $\text{Ir}_T\text{Ru}_V/\text{CoOOH}$ shifted to a higher energy than that of $\text{Ru}_T\text{Ir}_V/\text{CoOOH}$, indicating an elevated valence state of Ru single atoms (**Fig. R9a**). In the Ru K -edge EXAFS spectra, only one prominent peak was exhibited at about 1.5 Å for both $\text{Ir}_T\text{Ru}_V/\text{CoOOH}$ and $\text{Ru}_T\text{Ir}_V/\text{CoOOH}$, confirming the atomic dispersion of individual Ru atoms (**Fig. R9b**). By fitting the EXAFS spectra, the first-shell coordination of $\text{Ir}_T\text{Ru}_V/\text{CoOOH}$ was determined to be Ru-O with CNs of about 6.0, which was different from the Ru single atoms on $\text{Ru}_T\text{Ir}_V/\text{CoOOH}$ that coordinated with five oxygen atoms (**Table R2**). Accordingly, for Ru single atoms at V_O sites, one apex oxygen of the RuO_6 octahedral structure was inserted into the V_O sites, while four side OH^- of the octahedra formed hydrogen bonding with adjacent oxygen atoms on the CoOOH surface to stabilize the structure (**Fig. R10a**). In contrast, Ru single atoms at three-fold fcc hollow sites on $\text{Ru}_T\text{Ir}_V/\text{CoOOH}$ were stabilized by the three oxygen atoms of the sites, while the remaining two coordinated oxygen atoms were suspended at its surface as dangling bonds (**Fig. R10b**). The above results demonstrated that differences in the anchoring sites result in diverse electronic structures and configurations of Ir and Ru single atoms on $\text{Ru}_T\text{Ir}_V/\text{CoOOH}$ and $\text{Ir}_T\text{Ru}_V/\text{CoOOH}$ (**Fig. R11**).

Fig. R9 | Atomic structural analysis of $\text{Ir}_T\text{Ru}_V/\text{CoOOH}$ and $\text{Ru}_T\text{Ir}_V/\text{CoOOH}$. **a.** XANES spectra at the Ru K -edge of $\text{Ir}_T\text{Ru}_V/\text{CoOOH}$ and $\text{Ru}_T\text{Ir}_V/\text{CoOOH}$. **b.** Experimental and fitting EXAFS spectra at the Ru K -edge of $\text{Ir}_T\text{Ru}_V/\text{CoOOH}$ and $\text{Ru}_T\text{Ir}_V/\text{CoOOH}$. The inset atomic models are the first-shell coordination of Ru single atoms at V_O sites. The red and blue spheres represent O and Ru atoms, respectively. Ru foil and RuO_2 were used as references.

Samples	Path	R (Å)	CNs	σ^2 (10^{-3})	ΔE_0 (eV)	R -factor
$\text{Ir}_T\text{Ru}_V/\text{CoOOH}$	Ru-O	2.00 ± 0.02	6.1 ± 1.3	3.8	-4.8	0.02
$\text{Ru}_T\text{Ir}_V/\text{CoOOH}$	Ru-O	2.00 ± 0.02	5.0 ± 0.8	5.0	-0.4	0.01

Table R2 | Fitting results of Ru K -edge EXAFS spectra for $\text{Ir}_T\text{Ru}_V/\text{CoOOH}$ and $\text{Ru}_T\text{Ir}_V/\text{CoOOH}$. R , the distance between the absorber and backscatter atoms. The R value was phase corrected during fitting process; CNs, coordination numbers; σ^2 , Debye-Waller factors; ΔE_0 , the inner potential correction that accounts for the difference in the inner potential between

the sample and the references; S_0^2 , the amplitude reduction factor, the S_0^2 for the Ru K -edge EXAFS spectra fitting was determined to be 0.74; R -factor, the goodness of fit.

Fig. R10 | Atomic structural analysis of the Ru single atoms at diverse sites. a. b. Structural models of Ru single atoms at the V_O sites (a) and three-fold fcc hollow sites (b).

Fig. R11 | Schematic structural model of the site-specific single-atom catalysts. a. b. Structural models of $Ir_T Ru_V / CoOOH$ (a) and $Ru_T Ir_V / CoOOH$ (b).

6. The ratio of three-fold fcc hollow sites (Ru SA) and oxygen vacancy (Ir SA) should be quantified and systematically studied. How can this ratio be affected by reaction condition? And how can this ratio affect the ultimate electro-catalytic performance?

Response:

We thank the reviewer for the insightful comment and agree that the ratio of Ru and Ir species can influence the electrocatalytic performance. To determine the ratio of Ru and Ir species in $Ru_T Ir_V / CoOOH$, we performed ICP-AES measurements. The results demonstrated that the Ru and Ir species were about 4.1 at% and 2.4 at% for $Ru_T Ir_V / CoOOH$, respectively. Thus, the ratio of Ru and Ir species was about 1:0.6 (at%). To investigate the influence of the ratio on electrocatalytic performance, we fixed the mass loading of Ru single atoms while modifying the mass loading of Ir species. Different mass loadings of Ir species were achieved by adjusting the volume of single-atom precursors added into the electrolyte (0.10, 0.25, and 0.40 mL 100 μ M $IrCl_4$). ICP-AES measurements demonstrated that the ratio of Ru and Ir species were 1:0.3, 1:0.6, and 1:0.8 (at%), respectively. HAADF-STEM images showed that, for $Ru_T Ir_V / CoOOH$ (1:0.3)

and Ru_TIr_V/CoOOH (1:0.6), the numbers of Ir single atoms increased with the IrCl₄ concentration (**Fig. R12a** and **12b**). However, in the case of Ru_TIr_V/CoOOH (1:0.8), Ir clusters were emerged in the HAADF-STEM image (**Fig. R12c**).

Furthermore, we evaluated the OER performance of Ru_TIr_V/CoOOH with different ratios of Ru and Ir species. The polarization curves demonstrated that as the ratio of Ru and Ir species was regulated from 1:0.3 to 1:0.6, the overpotentials required at a current density of 500 mA cm⁻² for Ru_TIr_V/CoOOH were reduced from 290 to 260 mV, indicating a significant performance improvement (**Fig. R12d**). Nevertheless, when the ratio of Ru and Ir species was further adjusted from 1:0.6 to 1:0.8, the overpotentials required at a current density of 500 mA cm⁻² were slightly reduced from 260 to 250 mV for Ru_TIr_V/CoOOH, reflecting a minimal improvement in OER performance. The limited enhancement in performance may be attributed to the insufficient synergy between the Ru single atoms and Ir clusters. Above all, when the ratio of Ru and Ir single atoms exceeded a suitable value, some Ru single atoms failed to synergize with Ir single atoms, limiting the OER performance of the sample. In contrast, an excessively low ratio resulted in the formation of Ir clusters, which restricted further performance improvement. These results indicated that efficient synergy interaction between heterogeneous single atoms requires the ratio of Ru and Ir single atoms in a suitable range.

Fig. R12 | Structure characterizations and electrocatalytic performance of Ru_TIr_V/CoOOH with different ratios of Ru and Ir species. a. b. c. HAADF-STEM image of Ru_TIr_V/CoOOH with Ru and Ir species ratios of 1:0.3 (a), 1:0.6 (b), and 1:0.8 (c). d. Polarization curves of Ru_TIr_V/CoOOH with different ratios of Ru and Ir species for oxygen evolution in 1.0 M KOH electrolyte.

7. It is well acknowledged that the Ag/AgCl electrode (and every KCl-based reference electrode)

is not appropriate for electrochemical experiments in alkaline media, as the electrolyte OH^- ions diffuse into the reference electrode changing its composition during measurements. Thus, its potential does not remain constant, while the leached Cl^- can contaminate the electrode surface. You should have either used Hg/HgO reference electrode or immersed the SCE in a Luggin capillary for protection. Was the OCV stable, and were the measurements reproducible during your experiments?

Response:

We genuinely thank the reviewer for the valuable concern. We are sorry for the misleading that we skipped some details. In this work, before the electrochemical measurements, the Ag/AgCl electrode underwent a calibration process to ensure the potential accuracy. The calibration process was carried out in a three-electrode system using a high-purity hydrogen-saturated 0.5 M H_2SO_4 electrolyte, where a Pt wire, another Pt wire, and an Ag/AgCl electrode were used as the working, counter, and reference electrodes, respectively (**Fig. R13a**). Cyclic voltammetry was employed for the calibration with a scan rate of 1 mV s^{-1} . The average of the two potentials at which the current crossed zero was considered the thermodynamic potential for the hydrogen electrode reactions. The measured potential was determined to be 0.1965 V, which was close to the theoretical value of 0.197 V for the Ag/AgCl electrode (**Fig. R13b**). The calibration details have been added to the electrochemical measurements section in the revised manuscript.

Fig. R13 | Electrode calibration. a. Optical image of the Ag/AgCl electrode calibration process. b. Cyclic voltammety curve of Ag/AgCl electrode calibration in hydrogen-saturated 0.5 M H_2SO_4 solution at a scan rate of 1 mV s^{-1} .

To exclude the influence of electrode differences on electrochemical experiments, the performance of the samples was also evaluated using Hg/HgO as the reference electrode in 1.0 M KOH electrolyte. The polarization curves showed that the activities of the samples measured with the Hg/HgO electrode closely overlapped with those measured using the Ag/AgCl electrode in the manuscript (**Fig. R14a-d**). The results demonstrated that the performance evaluation presented in the manuscript are reliable.

Fig. R14 | Electrochemical performance of the samples towards oxygen evolution employing different reference electrode. a. b. Polarization curves of samples towards oxygen evolution in 1.0 M KOH electrolyte using Ag/AgCl and Hg/HgO as the reference electrode, respectively. **c. d.** Overpotentials required at a current density of 10 mA cm^{-2} for samples using Ag/AgCl and Hg/HgO as reference electrodes.

We also measured the open-circuit voltage (OCV) for $\text{Ru}_7\text{Ir}_V/\text{CoOOH}$ using Hg/HgO as a reference electrode. The results showed the OCV of the three independent measurements were 0.416, 0.416, and 0.415 V, respectively (**Fig. R15**). The consistent values confirm the stable OCV of $\text{Ru}_7\text{Ir}_V/\text{CoOOH}$.

Fig. R15 | OCV measurements for $\text{Ru}_7\text{Ir}_V/\text{CoOOH}$. The measurements were conducted three times.

In addition, to ensure the reproducibility of the electrocatalytic performance, the polarization curves were recorded three times for those samples with Hg/HgO as the reference electrode. The results exhibited inconspicuous differences in the polarization curves across the three independent tests for each sample (Fig. R16a and 16b). Furthermore, the differences in overpotentials required at a current density of 10 mA cm^{-2} were minimal across the three measurements (Fig. R16c and 16d). The minimal variations in the performance indicated excellent experiment reproducibility.

Fig. R16 | Electrocatalytic performance of the samples towards oxygen evolution in three independent experiments. a. b. Polarization curves of samples towards oxygen evolution in three independent experiments. **c. d.** Overpotentials of samples in three independent experiments at a current density of 10 mA cm^{-2} with error bars.

8. The LSVs should be corrected for the uncompensated ohmic resistance for a reliable kinetics analysis. Furthermore, you can propose possible reaction pathways based on the extracted Tafel slopes.

Response:

We sincerely thank this reviewer for the insightful questions. The LSV potentials in this manuscript were corrected to compensate for the solution resistance, which was calculated by the following equation: $E_{iR\text{-corrected}} = E \text{ (V vs RHE)} - iR$, where i is current, and R is the uncompensated ohmic electrolyte resistance. In oxygen-saturated 1.0 M KOH, R is measured as 0.75Ω via high-frequency alternating current impedance. The related information has been added to the electrochemical measurements section in the revised manuscript.

We agree with the reviewer’s viewpoint that the possible reaction pathways could be speculated

on the extracted Tafel slopes. Nevertheless, the Tafel slopes were affected by a variety of factors, including the surface intermediates coverage and the selected voltage position (*Nature* **587**, 408-413 (2020); *Science* **334**, 6061 (2021)). These factors made it difficult to accurately evaluate the Tafel slopes.

To propose the reaction pathways more precisely, *in-situ* ^{18}O isotope-labeling DEMS experiments were carried out. Two steps of DEMS experiments were designed using H_2^{18}O and H_2^{16}O as the supporting electrolyte (1.0 M KOH). At the first step, the $\text{Ru}_\text{T}/\text{CoOOH}$, $\text{Ir}_\text{V}/\text{CoOOH}$, $\text{Ir}_\text{T}\text{Ru}_\text{V}/\text{CoOOH}$, and $\text{Ru}_\text{T}\text{Ir}_\text{V}/\text{CoOOH}$ were labeled with ^{18}O isotopes by conducting cyclic voltammetry in the H_2^{18}O electrolyte. At the second step, the samples were rinsed with abundant water and then operated in the H_2^{16}O electrolyte. Cyclic voltammetry cycles were carried out with the simultaneous detection of signals for $^{36}\text{O}_2$ ($^{18}\text{O}^{18}\text{O}$), $^{34}\text{O}_2$ ($^{18}\text{O}^{16}\text{O}$), and $^{32}\text{O}_2$ ($^{16}\text{O}^{16}\text{O}$), respectively. The mass signal for $^{36}\text{O}_2$ ($^{18}\text{O}^{18}\text{O}$) was barely discernible in all samples, precluding the oxygen-oxygen coupling mechanism (OPM) as reaction pathway for all samples (**Fig. R17a-d**). Meanwhile, trace amount of $^{34}\text{O}_2$ ($^{18}\text{O}^{16}\text{O}$) in the OER products was detected, which can be attributed to the natural abundance of ^{18}O in deionized water within the experiments, suggesting the lattice oxygen-mediated mechanism (LOM) did not occur over those samples. Moreover, the predominant mass signal corresponded to $^{32}\text{O}_2$ ($^{16}\text{O}^{16}\text{O}$) was observed for all samples, indicating the adsorbate evolution mechanism for oxygen evolution. The correlated data and discussions have been added to the revised manuscript.

Fig. R17 | *In-situ* ^{18}O isotope-labeling DEMS measurements. **a-d.** *In-situ* DEMS signals of $^{36}\text{O}_2$, $^{34}\text{O}_2$, and $^{32}\text{O}_2$ for $\text{Ru}_\text{T}/\text{CoOOH}$ (**a**), $\text{Ir}_\text{V}/\text{CoOOH}$ (**b**), $\text{Ir}_\text{T}\text{Ru}_\text{V}/\text{CoOOH}$ (**c**), and $\text{Ru}_\text{T}\text{Ir}_\text{V}/\text{CoOOH}$ (**d**).

9. The dissolution of Ru is a common problem in Ru based electrocatalysts especially during OER process. but the corresponding content is not discussed in this manuscript.

Response:

We genuinely thank the reviewer for the discerning comment. We have quantified the content of dissolved Ru species in the electrolyte using ICP-MS during the stability test. As shown in **Fig. R18**, only 5.9 wt% of Ru species were dissolved after 500 h stability test at a current density of 10 mA cm^{-2} . The insignificant dissolution of Ru species may be attributed to the strong interactions between Ru single atoms and O atoms of the support. The corresponding discussions have been added to the revised manuscript.

Fig. R18 | The dissolved percentage of Ru species during the stability test. The dissolved Ru species were detected by ICP-MS during the stability test.

Reviewer #2 (Remarks to the Author):

I have carefully reviewed the manuscript titled "Site-specific synergy in heterogeneous single atoms for efficient oxygen evolution" submitted to Nature Communications. While the authors present an interesting study on heterogeneous single-atom catalysts for the oxygen evolution reaction, I have several major concerns that preclude publication of this work in its current form. My recommendation is to reject this manuscript for the following reasons:

Response:

We sincerely thank the reviewer for the valuable comments and suggestions on the manuscript. The simulated and experimental HAADF-STEM images were analyzed to provide direct evidence of the single atoms' anchoring sites. Besides, we recorded the *in-situ* XAFS spectrum at the Ru *K*-edge of Ru_TIr_V/CoOOH back to open circle potential (OCP) to prove that the increase in Ru-O coordination numbers during OER arose from oxygenated intermediates adsorption. Additionally, the XRD patterns of the samples were re-recorded at a scan rate of $2^\circ/\text{min}$ to accurately determine their crystal structures. A comprehensive comparison between Ru_TIr_V/CoOOH and state-of-the-art catalysts was also added to the revised manuscript. Moreover,

the samples were all re-prepared and their OER performance was also evaluated to ensure the reproducibility of the catalytic synthesis and performance. Hopefully, the revised manuscript will offer a clearer presentation of our conclusions.

1. Lack of direct structural evidence: The entire premise of this work relies on the authors' claim that in the Ru_TIr_V/CoOOH sample, "Ru single atoms are onto three-fold fcc hollow sites and Ir single atoms are onto oxygen vacancy (V_O) sites". However, no direct visual evidence, such as atomic-resolution HAADF-STEM images, is provided to unambiguously confirm this specific atomic arrangement. The authors rely heavily on EXAFS data interpretation, which can be subjective and model-dependent. In addition, the manuscript fails to clearly explain how the EXAFS results definitively prove the claimed atomic configuration.

Response:

We honestly thank this reviewer for the insightful questions. Firstly, the synthesis mechanisms were investigated to explore the anchoring sites of Ru and Ir single atoms. Specifically, the Ru single atoms were fabricated by the wet-chemical synthesis strategy. During synthesis process, the positively charged Ru³⁺ ions in the solution were selectively anchored onto negatively charged three-fold fcc hollow sites via electrostatic adsorption. Conversely, the Ir single atoms were constructed by an electrochemical deposition strategy. The negatively charged Ir(OH)₆²⁻ ions in the electrolyte were selectively anchored onto the positively charged V_O sites by electrostatic adsorption. Thus, the Ru and Ir single atoms should be anchored at three-fold fcc hollow sites and V_O sites, respectively.

To elucidate the anchoring sites of Ru and Ir single atoms, we analyzed the simulated HAADF-STEM images of Ru_T/CoOOH and Ir_V/CoOOH. In the simulated HAADF-STEM image of Ru_T/CoOOH from [-111] projection, Ru atoms at three-fold fcc hollow sites almost overlap with the Co column (**Fig. R19a**). Conversely, the Ir atoms at V_O sites were located at the interstice of three triangular Co columns in the simulated HAADF-STEM image of Ir_V/CoOOH from [-111] projection (**Fig. R19b**). Therefore, anchoring sites of Ru and Ir single atoms can be precisely identified according to the HAADF-STEM images. In the experimental HAADF-STEM image of Ru_T/CoOOH, Ru single atoms almost overlap with the Co column, consistent with the simulated HAADF-STEM image (**Fig. R19c**). The result suggested the Ru single atoms were anchored at the three-fold fcc hollow sites. In contrast, in the experimental HAADF-STEM image of Ir_V/CoOOH, bright spots can be discerned in the interstice of three triangular lattice sites, which were attributed to Ir single atoms at the V_O sites (**Fig. R19d**). The above results provided direct evidence that the Ru and Ir single atoms were anchored at three-fold fcc hollow sites and V_O sites, respectively.

Fig. R19 | Imaging of the anchoring sites of Ru and Ir single atoms on CoOOH. **a. b.** Simulated HAADF-STEM images of Ru and Ir single atoms at the three-fold fcc hollow site (**a**) and V_o site (**b**) of CoOOH from [-111] projection. **c. d.** Experimental HAADF-STEM images of $Ru_T/CoOOH$ (**c**) and Ir_VCoOOH (**d**). Ru and Ir single atoms were indicated by blue and yellow circles, respectively.

Subsequently, the EXAFS results and the structure of anchoring sites were analyzed to identify the configurations of Ru and Ir single atoms, respectively. For Ru single atoms on the $Ru_T/CoOOH$ and $Ru_TIr_V/CoOOH$, the Ru K -edge EXAFS fitting results demonstrated the Ru single atoms were coordinated with five oxygen atoms. Based on the coordination numbers of Ru single atoms and the structure of three-fold fcc hollow sites, the configuration of the Ru single atoms can be described as follows: Ru single atoms were stabilized by the three oxygen atoms at the three-fold fcc hollow sites, while the remaining two coordinated oxygen atoms were suspended at its surface as dangling bonds (**Fig. R20a**). For Ir single atoms on the $Ir_V/CoOOH$ and $Ru_TIr_V/CoOOH$, the Ir L_3 -edge EXAFS fitting results indicated the Ir single atoms were coordinated with six oxygen atoms. Consequently, the configuration of the Ir single atoms can be deduced as follows: one apex oxygen of the IrO_6 octahedral structure was inserted into the V_o sites, while four side OH of the octahedra formed hydrogen bonding with adjacent oxygen atoms on the CoOOH surface to stabilize the structure (**Fig. R20b**). The correlated data and discussions have been added to the revised manuscript.

Fig. R20 | Atomic structural analysis of the Ru and Ir single atoms at diverse sites. a. b. Structural models of Ru single atoms at the three-fold fcc hollow sites (a) and Ir single atoms at the V_o sites (b).

2. Ambiguous *in-situ* spectroscopic analysis: In the *in-situ* XAFS measurements, the authors observe an increase in Ru-O coordination numbers from 5.0 to 5.4 under applied potential. While they interpret this as evidence for adsorption of oxygenated reaction intermediates, alternative explanations are not adequately addressed or ruled out. This change could potentially arise from other structural modifications or experimental artifacts. A more rigorous analysis considering multiple scenarios is necessary.

Response:

We genuinely thank the reviewer for the valuable concern. To prove that the increase in the coordination number of Ru-O during the *in-situ* XAFS measurements arose from the adsorption of oxygenated reaction intermediates, we recorded the *in-situ* Ru *K*-edge XAFS spectrum of Ru₁Ir_v/CoOOH as it returned to OCP. The *in-situ* Ru *K*-edge XANES spectra and the corresponding local magnification exhibited the absorption edge shift to higher energy when the applied voltage increased from OCP to 1.65 V, indicating an elevated Ru valence state (Fig. R21a). Notably, when the applied potential was reversed back to OCP (B-OCP), the absorption edge of the Ru *K*-edge shifted to an energy position close to the OCP. The shift may originate from the desorption of oxygenated intermediates (Fig. R21b). *In-situ* EXAFS spectra demonstrated only one prominent peak at about 1.5 Å assigned to Ru-O bonding, confirming the single-atom structure of Ru species (Fig. R21c). Further EXAFS fitting results showed the coordination numbers of Ru-O increased from 5.0 to 5.4 as the applied voltage increased from OCP to 1.65 V, due to the adsorption of oxygenated intermediates (Fig. R21d and Table R3). Remarkably, when the applied potential was reversed to OCP, the coordination numbers of Ru-O decreased from 5.4 to 5.1, in response to the desorption of oxygenated intermediates out of OER conditions. The above results proved the Ru single atoms serve as the active sites for adsorption of oxygenated reaction intermediates. The correlated data and discussions have been added to the revised manuscript.

Fig. R21 | *In-situ* spectroscopic characterizations. **a.** *In-situ* Ru *K*-edge XANES spectra of Ru_TIr_V/CoOOH. **b.** The relative energy of absorption edge in Ru *K*-edge XANES under different applied potentials. **c.** Experimental and fitting *in-situ* EXAFS spectra of Ru_TIr_V/CoOOH at the Ru *K*-edge under different applied potentials. The experimental and fitting results are indicated as circles and solid lines, respectively. The inset atomic models are the configurations of Ru single atoms during OER. The red and blue spheres represent O and Ru atoms, respectively. **d.** The coordination numbers of Ru single atoms during the *in-situ* XAFS measurements under different applied potentials.

Applied voltage	Path	R (Å)	CNs	σ^2 (10^{-3})	ΔE_0 (eV)	R -factor
OCP	Ru-O	2.00 ± 0.02	5.0 ± 1.2	6.4	-2.0	0.02
1.45 V	Ru-O	2.00 ± 0.02	5.1 ± 1.0	5.6	-3.2	0.01
1.55 V	Ru-O	2.00 ± 0.02	5.3 ± 0.8	6.1	-2.6	0.01
1.65 V	Ru-O	2.00 ± 0.01	5.4 ± 0.8	6.0	-2.2	0.01
Back to OCP	Ru-O	2.00 ± 0.02	5.1 ± 0.9	5.9	-2.8	0.01

Table R3 | Fitting results of *in-situ* Ru *K*-edge EXAFS spectra for Ru_TIr_V/CoOOH. R , the distance between the absorber and backscatter atoms. The R value was phase corrected during fitting process; CNs, coordination numbers; σ^2 , Debye-Waller factors; ΔE_0 , the inner potential correction that accounts for the difference in the inner potential between the sample and the

references; S_0^2 , the amplitude reduction factor, the S_0^2 for the *in-situ* Ru *K*-edge EXAFS spectra fitting was determined to be 0.89; *R*-factor, the goodness of fit.

3. Inconsistencies in structural characterization: There are discrepancies between the microscopy and diffraction data presented. TEM images (e.g., Fig. 1f, 1g, S6, S11) suggest a relatively crystalline substrate, yet the corresponding XRD patterns show weak, broad peaks. This inconsistency is not addressed in the manuscript. Furthermore, some XRD patterns (particularly the (002) peak in Fig. S3) do not align well with the reference CoOOH pattern (PDF #26-1107). These discrepancies raise questions about the true nature and uniformity of the support material.

Response:

We genuinely thank the reviewer for raising this issue. Generally, atomic images captured by HAADF-STEM represent the arrangement of atoms on the catalyst's surface. XRD was used for determining the periodic arrangement of atoms in the entire crystal, which were strongly dependent on the crystallinity and thickness of the sample. In this work, CoOOH was synthesized via an electrochemical deposition method under mild conditions, which resulted in poor crystallinity of the sample, and consequently weak and broad peaks in XRD patterns. A similar phenomenon was also found in previous literature (*Angew. Chem. Int. Ed.* **57**, 2672-2676 (2018); *J. Solid State Chem.* **323**, 124048 (2023)). To evaluate the thickness of CoOOH, we performed atomic force microscope (AFM) measurement. According to the AFM image of CoOOH, the thickness of the nanosheets was approximately 3.6 nm (**Fig. R22a** and **22b**). The thinness of the catalysts leads to reduced scattering signals, resulting in weak, broad peaks in the XRD patterns.

Fig. R22 | Morphology characterization. a. AFM image of CoOOH. b. The height profiles correspond to the AFM image of CoOOH.

To obtain more accurate XRD patterns, the scan rate of the tests was reduced to 2°/min. The results demonstrated that all samples exhibited characteristic peaks that emerged at about 20.2°, 36.3°, 37.8° and 65.2°, which attributed to the (002), (100), (101), and (110) facets of CoOOH (PDF #26-1107), respectively (**Fig. R23a-e**). The experimental results confirmed that the support material for all the as-prepared single-atom catalysts was CoOOH.

Fig. R23 | XRD characterizations. a-e. XRD patterns of CoOOH (a), Ru_T/CoOOH (b), Ir_V/CoOOH (c), Ru_TIr_V/CoOOH (d), and Ir_VRu_T/CoOOH (e).

4. Overinterpretation of DFT results: The computational studies appear to be based on idealized models that may not accurately represent the experimental system, given the uncertainties in the actual atomic structure. Without a more robust structural characterization, the relevance and validity of the DFT calculations are questionable.

Response:

We honestly thank this reviewer for the insightful comments. We first investigate the anchoring sites of Ru and Ir single atoms by analyzing the synthesis mechanisms. Subsequently, we performed HAADF-STEM simulations and captured experimental HAADF-STEM images to identify the anchoring sites of Ru and Ir single atoms. Then, the configuration of site-specific single atoms was precisely investigated by combining EXAFS results and the structure of anchoring sites. Based on above results, we constructed structural models of the samples to explore the site-specific synergy in heterogenous Ru and Ir single atoms for oxygen evolution.

Firstly, according to the synthesis mechanism, single-atom precursors with different electronegativity can be selectively anchored onto the sites with opposite electronegativity. In this work, the wet-chemical synthesis strategy was employed to anchor Ru single atoms. During synthesis, the positively charged Ru³⁺ ions in the solution were selectively anchored onto the negatively charged three-fold fcc hollow sites via electrostatic adsorption. As a result, Ru single atoms should be anchored at the three-fold fcc hollow sites. In contrast, an electrochemical deposition strategy was applied to anchor Ir single atoms. In the synthesis process, the negatively charged Ir(OH)₆²⁻ ions in the electrolyte were selectively anchored onto the positively charged V_O sites through electrostatic adsorption. Consequently, Ir single atoms were likely anchored at the V_O sites.

Subsequently, we analyzed the simulated HAADF-STEM images of Ru_T/CoOOH and Ir_V/CoOOH to identify the anchoring sites of Ru and Ir single atoms. The Ru atoms at three-fold fcc hollow sites almost coincide with the Co atomic column in the simulated HAADF-STEM image of Ru_T/CoOOH from [-111] projection (**Fig. R24a**). In contrast, in the simulated HAADF-STEM image of Ir_V/CoOOH from the [-111] projection, the Ir atoms at V_O sites were positioned at the interstice of three triangular Co columns (**Fig. R24b**). Therefore, the anchoring sites of Ru and Ir single atoms can be precisely determined from the experimental HAADF-STEM images. In the experimental HAADF-STEM image of Ru_T/CoOOH, Ru single atoms nearly overlap with the Co column, which suggested that Ru single atoms were anchored at the three-fold fcc hollow sites (**Fig. R24c**). The result suggests that Ru single atoms were anchored at the three-fold fcc hollow sites. In addition, bright spots can be discerned in the interstice of three triangular lattice sites in the experimental HAADF-STEM image of Ir_V/CoOOH, which were ascribed to Ir single atoms at the V_O sites (**Fig. R24d**). The above results demonstrated that the Ru and Ir single atoms were anchored at three-fold fcc hollow sites and V_O sites, respectively.

Fig. R24 | Imaging of the anchoring sites of Ru and Ir single atoms on CoOOH. a. b. Simulated HAADF-STEM images of Ru and Ir single atoms at the three-fold fcc hollow site (**a**) and V_O site (**b**) of CoOOH from [-111] projection. **c. d.** Experimental HAADF-STEM images of Ru_T/CoOOH (**c**) and Ir_VCoOOH (**d**). Ru and Ir single atoms were indicated by blue and yellow circles, respectively.

Afterward, the configurations of the Ru and Ir single atoms were revealed by analyzing the results of the EXAFS fitting and the structure of the anchoring sites. The EXAFS fitting results at the Ru *K*-edge demonstrated the Ru single atoms were coordinated with five oxygen atoms on both Ru_T/CoOOH and Ru_TIr_V/CoOOH. Therefore, based on the coordination numbers of Ru

single atoms and the structure of three-fold fcc hollow sites, the configuration of the Ru single atoms can be described as follows: Ru single atoms were stabilized by the three oxygen atoms at the three-fold fcc hollow sites, while the remaining two coordinated oxygen atoms were suspended at its surface as dangling bonds (**Fig. R25a**). Subsequently, the configurations of Ir single atoms at V_O sites were also elucidated. The Ir L_3 -edge EXAFS fitting results indicated the Ir single atoms were coordinated with six oxygen atoms on the Ir_v/CoOOH and Ru_TIr_v/CoOOH. Consequently, according to the coordination numbers of Ir single atoms and structure of V_O sites, the configuration of the Ir single atoms on the V_O sites can be deduced as follows: one apex oxygen of the IrO₆ octahedral structure was inserted into the V_O sites, while four side OH⁻ of the octahedra formed hydrogen bonding with adjacent oxygen atoms on the CoOOH surface to stabilize the structure (**Fig. R25b**).

Fig. R25 | Atomic structural analysis of the Ru and Ir single atoms at diverse sites. a. b. Structural models of Ru single atoms at the three-fold fcc hollow sites (a) and Ir single atoms at the V_O sites (b).

5. Limited comparison with state-of-the-art catalysts: While the authors claim superior performance, a comprehensive comparison with other leading OER catalysts in the field is lacking. This makes it difficult to assess the true significance of the reported activity improvements.

Response:

We genuinely thank the reviewer for the valuable suggestions. We have added the comparison of the catalytic performance for Ru_TIr_v/CoOOH with some state-of-the-art catalysts to the revised manuscript as **Supplementary Figure 27** and **Supplementary Table 5**. The overpotential required for Ru_TIr_v/CoOOH to reach a current density of 10 mA cm⁻² was lower than most other catalysts, demonstrating its excellent OER activity (**Fig. R26**). Furthermore, the lower Tafel slope indicated the faster kinetics of Ru_TIr_v/CoOOH compared to other catalysts. The results showed the performance of Ru_TIr_v/CoOOH was on par with the best records of currently reported OER catalysts.

Fig. R26 | Comparison of overpotential at a current density of 10 mA cm⁻² and Tafel slopes for recently reported catalysts in alkaline electrolyte.

6. Insufficient mechanistic insights: The proposed synergistic effects between Ru and Ir atoms are not thoroughly elucidated. The manuscript lacks a clear, molecularly-detailed mechanism explaining how the specific atomic arrangements lead to enhanced catalytic activity.

Response:

We genuinely thank the reviewer for the insightful comment. Firstly, to investigate whether the enhanced OER activity of Ru_TIr_V/CoOOH was attributed to the optimization of Ru single atoms' electronic structure, we characterized the electronic structure of Ru single atoms before and after anchoring Ir single atoms. According to the Ru *K*-edge XANES, the absorption edge of Ru_TIr_V/CoOOH overlapped with that of Ru_T/CoOOH, presenting similar valence states of Ru single atoms. The similar valence states suggested that the electronic structure of the Ru single atoms exhibited negligible changes before and after the anchoring of the Ir single atoms.

Subsequently, we performed an in-depth analysis of the reaction pathways for Ru_T/CoOOH and Ru_TIr_V/CoOOH by theoretical calculations to investigate the synergistic effect between Ru and Ir single atoms. According to the results of *in-situ* attenuated total reflection surface-enhanced infrared absorption spectroscopy (ATR-SEIRAS) and ¹⁸O isotope-labeling DEMS, the absorbate evolution mechanism with four concerted proton-electron transfer steps was considered for both samples. Based on the results of *in-situ* XAFS spectra, the Ru single atoms at the three-fold fcc hollow sites were considered active sites for the adsorption of reaction key intermediates. For Ru_T/CoOOH, the reaction started from the adsorption of OH⁻ ions on the Ru site, followed by the sequential deprotonation to form *O, O-O bonding formation to generate OOH*, and desorption to produce O₂(g). Remarkably, for Ru_TIr_V/CoOOH, as the reaction proceeds to the *OOH step, the distance between *OOH intermediates (H atom) on Ru single atom and coordinated hydroxide radicals of Ir single atom (O atom) was determined to be 2.37 Å, indicating the presence of hydrogen bonding interaction. The hydrogen bonding interaction significantly stabilized the *OOH intermediates, reducing the theoretical overpotential by 0.62 V. In addition, the hydrogen bonding interaction was further confirmed by the *in-situ* ATR-SEIRAS, where the wavenumber of *OOH species on Ru_T/CoOOH shifted from 1051 cm⁻¹ to 1094 cm⁻¹ on

Ru_TIr_V/CoOOH. Electrochemical evaluation demonstrated that the formation of hydrogen bonding interactions significantly reduced the overpotential from 220 mV for Ru_T/CoOOH to 180 mV for Ru_TIr_V/CoOOH at a current density of 10 mA cm⁻².

7. Concerns about reproducibility: Given the complexity of the synthesis process and the sensitivity of single-atom catalysts to preparation conditions, more detailed information on the reproducibility of the catalyst synthesis and performance is needed.

Response:

We deeply appreciate the reviewer's professional and constructive comments. The CoOOH, Ru_T/CoOOH, Ir_V/CoOOH, Ru_TIr_V/CoOOH, and Ir_TRu_V/CoOOH were re-synthesized to confirm the reproducibility of the catalyst synthesis. TEM images of the re-prepared samples exhibited nanosheet morphologies similar to the samples in the manuscript (**Fig. R27a, 28a, 29a, 30a, and 31a**). All characteristic peaks in the XRD patterns of re-prepared samples were attributed to CoOOH (PDF #26-1107), indicating the absence of Ru- and Ir-based metals or metal oxides (**Fig. R27b, 28b, 29b, 30b, and 31b**). HAADF-STEM images identified isolated dispersion of Ru and Ir atoms (**Fig. 28c, 29c, 30c, and 31c**). EDX elemental mapping images showed the uniform metal distribution across the reprepared samples (**Fig. 28d, 29d, 30d, and 31d**). The results validated the reproducibility of the catalyst synthesis.

Fig. R27 | Morphology and structure characterizations of re-prepared CoOOH. a. TEM image. b. XRD pattern.

Fig. R28 | Morphology and structure characterizations of re-prepared Ru₇/CoOOH. a. TEM image. **b.** XRD pattern. **c.** HAADF-STEM image. **d.** EDX elemental mapping.

Fig. R29 | Morphology and structure characterizations of re-prepared Ir_V/CoOOH. a. TEM image. **b.** XRD pattern. **c.** HAADF-STEM image. **d.** EDX elemental mapping.

Fig. R30 | Morphology and structure characterizations of re-prepared Ru_TIr_V/CoOOH. a. TEM image. **b.** XRD pattern. **c.** HAADF-STEM image. **d.** EDX elemental mapping.

Fig. R31 | Morphology and structure characterizations of re-prepared Ir_TRu_V/CoOOH. a. TEM image. **b.** XRD pattern. **c.** HAADF-STEM image. **d.** EDX elemental mapping.

In addition, we performed electrocatalytic evaluations of these samples to verify the reproducibility of their catalytic performance. A Hg/HgO electrode was employed as the reference electrode. The polarization curves of these re-prepared samples were almost identical to those in the manuscript. (**Fig. R32a** and **R32b**). The overpotentials of the re-prepared samples at a current density of 10 mA cm^{-2} were also close to those of the samples presented in the manuscript (**Fig. R32c** and **R32d**).

Fig. R32 | Comparison of the electrocatalytic performance of the re-prepared samples and the samples in the manuscript towards oxygen evolution. a. b. Polarization curves of re-prepared samples towards oxygen evolution in 1.0 M KOH electrolyte. The polarization curves of samples in the manuscript were used as references. **c. d.** Overpotentials of re-prepared samples at a current density of 10 mA cm^{-2} . The overpotentials of samples in the manuscript were used as references.

To further ensure the reproducibility of the electrocatalytic performance, the polarization curves were recorded three times for those samples. The results exhibited inconspicuous differences in the polarization curves across the three independent tests for each sample (**Fig. R33a** and **R33b**). Furthermore, the differences in overpotentials required at a current density of 10 mA cm^{-2} were minimal across the three measurements (**Fig. R33c** and **R33d**). The above results demonstrated excellent experimental reproducibility.

Fig. R33 | Electrocatalytic performance of re-prepared samples towards oxygen evolution in three independent experiments. a. b. Polarization curves of samples towards oxygen evolution in three independent experiments. **c. d.** Overpotentials of samples in three independent experiments at a current density of 10 mA cm⁻² with error bars.

In light of these significant concerns, I recommend that the manuscript be rejected in its current form. Substantial additional experimental evidence, particularly direct atomic-scale imaging and more rigorous spectroscopic analyses, would be required to support the authors' claims. A major revision addressing these points might be considered for resubmission, but the current work falls short of the standards expected for publication in Nature Communications.

Response:

We genuinely thank this reviewer for the insightful questions and valuable suggestions. We also appreciate the doubts and efforts of the reviewer, which certainly contributed to improving the quality of our manuscript. We have revised the manuscript based on the suggestions. Specifically, the atomic-scale HAADF-STEM images of Ru₇/CoOOH and Ir_V/CoOOH were provided. Then we combined the experimental HAADF-STEM images with simulated HAADF-STEM images to confirm that Ru and Ir single atoms were anchored at the three-fold fcc hollow sites and Vo sites, respectively. Subsequently, we recorded the *in-situ* XAFS spectrum at the Ru *K*-edge of Ru₇Ir_V/CoOOH as it returned to OCP. As the voltage returned to the OCP, the coordination number of Ru single atoms decreased, corresponding to the desorption of reaction intermediates. The results proved the Ru single atoms serve as the active sites for adsorption of oxygenated reaction intermediates. Additionally, the samples were all re-prepared and their OER performance was also evaluated to ensure the reproducibility of the catalyst synthesis and performance. We hope that the revisions incorporated into the manuscript will adequately address the concerns.

Reviewer #3 (Remarks to the Author):

The work by P. Ma et al. reports a relatively novel method to construct heterogeneous single-atom systems and reveals the site-specific synergy in heterogeneous single atoms for oxygen evolution. The as-formed catalysts show decent OER activity and long durability. However, I have concerns regarding numerous explanations throughout the paper that appear to be illogical or not sufficiently justified. Considering rather unclear expression of reaction mechanism design and performance, this manuscript cannot be accepted for publication before the following issues have been addressed. Please find the specific comments below:

Response:

We would like to thank the reviewer for the detailed analysis of the manuscript and truly insightful suggestions. To strengthen the claims and conclusions of this manuscript, we have further enhanced the analysis of the main points in this work and supplemented the key information raised. Specifically, a comprehensive characterization of Ir_TRu_V/CoOOH was carried out to investigate the intrinsic reasons for site-specific synergy in heterogeneous Ru and Ir single atoms. Then, we refitted the EXAFS of Ru and Ir single atoms in the manuscript to ensure the reasonableness of the fitting results. Additionally, the S_0^2 of all EXAFS fitting results has been added to the revised manuscript. Meanwhile, the specific activities of those samples were compared by normalizing the current densities against ECSAs. Furthermore, we have extended the time for the stability test. The activity and stability of Ru_TIr_V/CoOOH were also evaluated by assembling the sample into a membrane electrode assembly (MEA) water electrolyzer. In addition, we have analyzed the changes in the bond length of the Ru-O bonding during OER by the simulated structural model. Moreover, we conducted the *in-situ* ¹⁸O isotope-labeling DEMS experiments to validate the OER mechanism of the samples. Subsequently, we analyzed the hydrogen bonding interactions induced by Ir single atoms. Through addressing these issues, we hope that the revised manuscript will be favorably considered.

1. The paper asserts the site-specific synergy and claims that Ru_TIr_V/CoOOH shows the better OER performance than Ir_TRu_V/CoOOH. However, the studies about Ir_TRu_V/CoOOH are relatively deficient. Whether it is more suitable to compare the difference between Ru_TIr_V/CoOOH and Ir_TRu_V/CoOOH for the topic of “site-specific”, rather than do excessive efforts on Ru_T/CoOOH and Ir_V/CoOOH samples.

Response:

We genuinely thank the reviewer for the valuable comment and agree that the Ir_TRu_V/CoOOH should be analyzed in detail and compared with Ru_TIr_V/CoOOH. We performed a comprehensive structure characterization and mechanistic study of Ir_TRu_V/CoOOH to investigate the intrinsic reasons of site-specific synergy in heterogeneous Ru and Ir single atoms. The Ir_TRu_V/CoOOH was prepared by anchoring Ir single atoms onto the three-fold fcc hollow sites and Ru single atoms onto the V_O sites, respectively. TEM image showed the Ir_TRu_V/CoOOH presented nanosheet morphology (**Fig. R34a**). All characteristic peaks in the XRD patterns were attributed to CoOOH (PDF #26-1107), indicating the absence of Ir- and Ru-based metals or metal oxides

(Fig. R34b). HAADF-STEM image of Ir_TRu_V/CoOOH identified isolated bright spots relative to the support, corresponding to Ir or Ru atoms (Fig. R34c). EDX elemental mapping images showed the uniform distribution of Ir and Ru elements across the Ir_TRu_V/CoOOH (Fig. R34d). The quantitative analysis by ICP-AES demonstrated that the contents of Ir and Ru elements were 3.9 wt% and 4.0 wt%, respectively.

Fig. R34 | Morphology and structure characterizations of Ir_TRu_V/CoOOH. a. TEM image. **b.** XRD pattern. **c.** HAADF-STEM image. **d.** EDX elemental mapping.

Subsequently, the XANES and EXAFS results were analyzed to investigate the detailed electronic structures and coordination environments of Ir single atoms on Ir_TRu_V/CoOOH and Ru_TIr_V/CoOOH. In the Ir *L*₃-edge XANES spectra, the white line intensity of Ir_TRu_V/CoOOH was higher than that of Ru_TIr_V/CoOOH, indicating that the valence state of Ir single atoms anchored at three-fold fcc hollow sites was higher than that of anchored at V_O sites (Fig. R35a). In the Ir *L*₃-edge EXAFS spectra, both Ir_TRu_V/CoOOH and Ru_TIr_V/CoOOH exhibited only one characteristic peak at about 1.6 Å assigned to Ir-O bonding, which substantiated the isolated dispersion of Ir atoms. Further EXAFS fitting results demonstrated that the CNs of Ir-O for Ir_TRu_V/CoOOH and Ru_TIr_V/CoOOH were both about 6.0 (Fig. R35b and Table R4).

Fig. R35 | Atomic structural analysis of the Ir_TRu_V/CoOOH and Ru_TIr_V/CoOOH. a. XANES spectra at the Ir *L*₃-edge of Ir_TRu_V/CoOOH. **b.** Experimental and fitting EXAFS spectra at the Ir *L*₃-edge of Ir_TRu_V/CoOOH. The inset atomic models are the first-shell coordination of Ir single atoms at three-fold fcc hollow sites. The red and yellow spheres represent O and Ir atoms, respectively. Ir foil and IrO₂ were used as references.

Samples	Path	R (Å)	CNs	σ^2 (10 ⁻³)	ΔE_0 (eV)	R -factor
Ir _T Ru _V /CoOOH	Ir-O	2.00 ± 0.02	6.0 ± 1.1	6.5	9.8	0.01
Ru _T Ir _V /CoOOH	Ir-O	2.00 ± 0.01	6.0 ± 1.1	7.9	10.0	0.02

Table R4 | Fitting results of Ir *L*₃-edge EXAFS spectra for Ir_TRu_V/CoOOH and Ru_TIr_V/CoOOH. *R*, the distance between the absorber and backscatter atoms. The *R* value was phase corrected during fitting process; *CNs*, coordination numbers; σ^2 , Debye-Waller factors; ΔE_0 , the inner potential correction that accounts for the difference in the inner potential between the sample and the references; S_0^2 , the amplitude reduction factor, the S_0^2 for the Ir *L*₃-edge EXAFS spectra fitting was determined to be 1.05; *R*-factor, the goodness of fit.

Then, the electronic structures and coordination environments of Ru single atoms on Ir_TRu_V/CoOOH and Ru_TIr_V/CoOOH were investigated. In the Ru *K*-edge XANES spectra, the absorption edge of Ir_TRu_V/CoOOH shifted to a higher energy than that of Ru_TIr_V/CoOOH, indicating an elevated valence state of Ru single atoms (**Fig. R36a**). In the Ru *K*-edge EXAFS spectra, only one prominent peak was exhibited at about 1.5 Å for Ir_TRu_V/CoOOH, confirming the atomic dispersion of individual Ru atoms (**Fig. R36b**). By fitting the EXAFS spectra, the first-shell coordination of Ir_TRu_V/CoOOH was determined to be Ru-O with *CNs* of about 6.0, which varied from the Ru single atoms at three-fold fcc hollow sites that coordinated with five oxygen atoms (**Table R5**).

Fig. R36 | Atomic structural analysis of the Ir_TRu_V/CoOOH and Ru_TIr_V/CoOOH. a. XANES spectra at the Ru *K*-edge of Ir_TRu_V/CoOOH. **b.** Experimental and fitting EXAFS spectra at the Ru *K*-edge of Ir_TRu_V/CoOOH. The inset atomic models are the first-shell coordination of Ru single atoms at V_O sites. The red and blue spheres represent O and Ru atoms, respectively. Ru foil and RuO₂ were used as references.

Samples	Path	R (Å)	CNs	σ^2 (10 ⁻³)	ΔE_0 (eV)	R -factor
Ir _T Ru _V /CoOOH	Ru-O	2.00 ± 0.02	6.1 ± 1.3	3.8	-4.8	0.02
Ru _T Ir _V /CoOOH	Ru-O	2.00 ± 0.02	5.0 ± 0.8	5.0	-0.4	0.01

Table R5 | Fitting results of Ru *K*-edge EXAFS spectra for Ir_TRu_V/CoOOH and Ru_TIr_V/CoOOH. *R*, the distance between the absorber and backscatter atoms. The *R* value was phase corrected during fitting process; *CNs*, coordination numbers; σ^2 , Debye-Waller factors; ΔE_0 , the inner potential correction that accounts for the difference in the inner potential between the sample and the references; S_0^2 , the amplitude reduction factor, the S_0^2 for the Ru *K*-edge EXAFS spectra fitting was determined to be 0.74; *R*-factor, the goodness of fit.

To investigate the synergy in site-specific Ru and Ir single atoms, we evaluated the catalytic performance of Ru_TIr_V/CoOOH and Ir_TRu_V/CoOOH toward oxygen evolution. The polarization curves demonstrated the Ru_TIr_V/CoOOH showed dramatically improved current densities relative to the Ir_TRu_V/CoOOH (**Fig. R37a**). The overpotential required to reach a current density of 10 mA cm⁻² of Ru_TIr_V/CoOOH was 180 mV, which was 90 mV lower than that of Ir_TRu_V/CoOOH (**Fig. R37b**). The above results demonstrated the synergy in Ru and Ir single atoms was site-specific. The effective synergy between the Ru single atoms at the three-fold fcc hollow sites and the Ir single atoms at the V_O sites significantly enhanced the OER performance. In contrast, exchanging the anchoring sites of Ru and Ir single atoms resulted in a remarkably attenuated synergy. Above all, this work not only proposed a synthesis strategy for constructing heterogeneous single atoms but also revealed the site-specific synergy in heterogeneous Ru and Ir single atoms for oxygen evolution. The correlated data and discussions have been added to the revised manuscript.

Fig. R37 | Electrochemical performance of site-specific single-atom catalysts towards oxygen evolution. a. Polarization curves of Ru_xIr_y/CoOOH and Ir_xRu_y/CoOOH towards oxygen evolution in 1.0 M KOH electrolyte. **b.** Overpotentials of Ru_xIr_y/CoOOH and Ir_xRu_y/CoOOH at a current density of 10 mA cm⁻².

2. For EXAFS fitting, usually, for the fitting of transition metals in the first shell, the inner potential correction is generally $-10 \text{ eV} \leq \Delta E_0 \leq 10 \text{ eV}$; the disorder degree (σ^2) is generally considered to be acceptable at 0.003-0.007; also, the amplitude reduction factor (S_0^2) should be firstly provided. Therefore, it is recommended that the authors check the data in Supplementary Tables and refit EXAFS more carefully.

Response:

Thanks for the reviewer's suggestions. To ensure the reliability of our conclusion, we have refitted the XAFS spectra for Ru and Ir single atoms in the manuscript. Furthermore, in response to the insightful suggestions, the S_0^2 of all EXAFS fitting results have been incorporated into our revised manuscript. For the EXAFS spectra fitting at the Ru K -edge, the value of ΔE_0 were evaluated to range from -4.8 to -0.4 eV, while the σ^2 were evaluated to range from 0.0038 to 0.0064 (**Table R6** and **R7**). The value of S_0^2 for the *ex-situ* and *in-situ* Ru K -edge EXAFS spectra fitting were determined to be 0.74 and 0.89, respectively. All the fitting parameters of Ru K -edge EXAFS were located at reasonable range, confirming the reliability of the fitting results.

For the Ir L_3 -edge EXAFS spectra fitting, the value of ΔE_0 were evaluated to range from 9.8 to 10.0 eV (**Table R8** and **R9**). The value of S_0^2 for the Ir L_3 -edge EXAFS spectra fitting was determined to be 1.05. Thus, the ΔE_0 and S_0^2 of Ir L_3 -edge EXAFS were located at reasonable range. The σ^2 for the EXAFS fitting of Ir species was evaluated to range from 0.0069 to 0.01, slightly exceeding the range of 0.003-0.007. This can be attributed to the greater degree of disorder induced by the incorporation of heteroatoms, which consequently resulted in an increased σ^2 value (*Phys. Rev. B* **48**, 585 (1993); *J. Mater. Chem.* **14**, 102-110 (2004)). According to previous literature, the σ^2 value was influenced by the degree of disorder induced by the incorporated heteroatoms (*Nat Commun.* **13**, 7754 (2022); *Adv. Mater.* **33**, 2007056 (2021)). The maximum value of σ^2 for the EXAFS spectra fitting at the Ir L_3 -edge was 0.01 in this manuscript, which was within a reasonable range. The correlated data and discussions have been added to the revised manuscript.

Samples	Path	R (Å)	CNs	σ^2 (10^{-3})	ΔE_0 (eV)	R-factor
Ru _T /CoOOH	Ru-O	2.00 ± 0.02	5.0 ± 0.8	5.3	-1.2	0.01
Ru _T Ir _V /CoOOH	Ru-O	2.00 ± 0.02	5.0 ± 0.8	5.0	-0.4	0.01
Ir _T Ru _V /CoOOH	Ru-O	2.00 ± 0.02	6.1 ± 1.3	3.8	-4.8	0.02

Table R6 | Fitting results of Ru K-edge EXAFS spectra for Ru_T/CoOOH, Ru_TIr_V/CoOOH, and Ir_TRu_V/CoOOH. R , the distance between the absorber and backscatter atoms. The R value was phase corrected during fitting process; CNs, coordination numbers; σ^2 , Debye-Waller factors; ΔE_0 , the inner potential correction that accounts for the difference in the inner potential between the sample and the references; S_0^2 , the amplitude reduction factor, the S_0^2 for the Ru K-edge EXAFS spectra fitting was determined to be 0.74; R-factor, the goodness of fit.

Applied voltage	Path	R (Å)	CNs	σ^2 (10^{-3})	ΔE_0 (eV)	R-factor
OCP	Ru-O	2.00 ± 0.02	5.0 ± 1.2	6.4	-2.0	0.02
1.45 V	Ru-O	2.00 ± 0.02	5.1 ± 1.0	5.6	-3.2	0.01
1.55 V	Ru-O	2.00 ± 0.02	5.3 ± 0.8	6.1	-2.6	0.01
1.65 V	Ru-O	2.00 ± 0.01	5.4 ± 0.8	6.0	-2.2	0.01
Back to OCP	Ru-O	2.00 ± 0.02	5.1 ± 0.9	5.9	-2.8	0.01

Table R7 | Fitting results of *in-situ* Ru K-edge EXAFS spectra for Ru_TIr_V/CoOOH. R , the distance between the absorber and backscatter atoms. The R value was phase corrected during fitting process; CNs, coordination numbers; σ^2 , Debye-Waller factors; ΔE_0 , the inner potential correction that accounts for the difference in the inner potential between the sample and the references; S_0^2 : the amplitude reduction factor, the S_0^2 for the *in-situ* Ru K-edge EXAFS spectra fitting was determined to be 0.89; R-factor, the goodness of fit.

Samples	Path	R (Å)	CNs	σ^2 (10^{-3})	ΔE_0 (eV)	R-factor
Ir _V /CoOOH	Ir-O	2.00 ± 0.01	6.1 ± 1.3	10.0	10.0	0.02
Ru _T Ir _V /CoOOH	Ir-O	2.00 ± 0.01	6.0 ± 1.1	7.9	10.0	0.02
Ir _T Ru _V /CoOOH	Ir-O	2.00 ± 0.02	6.0 ± 1.1	6.5	9.8	0.01

Table R8 | Fitting results of Ir L_3 -edge EXAFS spectra for Ir_v/CoOOH, Ru_TIr_v/CoOOH, and Ir_TRu_v/CoOOH. R , the distance between the absorber and backscatter atoms. The R value was phase corrected during fitting process; CNs , coordination numbers; σ^2 , Debye-Waller factors; ΔE_0 , the inner potential correction that accounts for the difference in the inner potential between the sample and the references; S_0^2 , the amplitude reduction factor, the S_0^2 for the Ir L_3 -edge EXAFS spectra fitting was determined to be 1.05; R -factor, the goodness of fit.

Applied voltage	Path	R (Å)	CNs	σ^2 (10^{-3})	ΔE_0 (eV)	R -factor
OCP	Ir-O	2.00 ± 0.01	6.0 ± 1.1	9.6	10.0	0.02
1.45 V	Ir-O	2.00 ± 0.01	6.0 ± 1.1	8.7	10.0	0.01
1.55 V	Ir-O	2.00 ± 0.01	6.0 ± 1.1	8.1	10.0	0.02
1.65 V	Ir-O	2.00 ± 0.01	6.0 ± 1.1	6.9	10.0	0.02

Table R9 | Fitting results of *in-situ* Ir L_3 -edge EXAFS spectra for Ru_TIr_v/CoOOH. R , the distance between the absorber and backscatter atoms. The R value was phase corrected during fitting process; CNs , coordination numbers; σ^2 , Debye-Waller factors; ΔE_0 , the inner potential correction that accounts for the difference in the inner potential between the sample and the references; S_0^2 : the amplitude reduction factor, the S_0^2 for the *in-situ* Ir L_3 -edge EXAFS spectra fitting was determined to be 1.05; R -factor, the goodness of fit.

3. Please add some values in Fig. 3e for intuitively evaluating the OER performance. In addition, please consider ECSA- or BET- corrected performance for the further comparisons.

Response:

We sincerely thank the reviewer for the valuable comment. The specific activities of those samples were compared by normalizing the current densities against electrochemical active surface areas (ECSAs) (**Fig. R38**). The specific activities delivered an increasing trend from CoOOH, Ru_T/CoOOH, Ir_v/CoOOH, to Ru_TIr_v/CoOOH at all potentials (**Fig. R39a and 39b**). Notably, at an overpotential of 250 mV, the specific activities of CoOOH, Ru_T/CoOOH, Ir_v/CoOOH, and Ru_TIr_v/CoOOH were calculated to be 5.6, 49.6, 152.6, and 625.2 mA cm⁻², respectively (**Fig. R39c**). Therefore, the specific activity of Ru_TIr_v/CoOOH was 111.6, 12.6, and 4.1 times higher than that of CoOOH, Ru_T/CoOOH, Ir_v/CoOOH at this overpotential, respectively. Additionally, the specific activity of Ir_TRu_v/CoOOH was 11.7 mA cm⁻² at an overpotential of 250 mV, which was 53.4 times lower than that of Ru_TIr_v/CoOOH (**Fig. R39d**). The above results demonstrated the efficient synergy between Ru single atoms at three-fold fcc hollow sites and Ir single atoms at V_o sites for oxygen evolution.

Fig. R38 | CV curves and charging current density differences of CoOOH, Ru_T/CoOOH, Ir_V/CoOOH, Ir_TRu_V/CoOOH, and Ru_TIr_V/CoOOH. a-e. CV curves of CoOOH (a), Ru_T/CoOOH (b), Ir_V/CoOOH (c), Ir_TRu_V/CoOOH (d), and Ru_TIr_V/CoOOH (e), respectively. f-j. Charging current density differences of CoOOH (f), Ru_T/CoOOH (g), Ir_V/CoOOH (h), Ir_TRu_V/CoOOH (i), and Ru_TIr_V/CoOOH (j), respectively.

Fig. R39 | Specific activities of the samples normalized against ECSAs. a. b. Polarization curves of catalysts towards oxygen evolution in 1.0 M KOH electrolyte against ECSAs. **c. d.** Intrinsic activities of catalysts normalized against ECSAs at an overpotential of 250 mV.

In addition, specific values of the samples have been added to Fig. 3e for performance comparison (**Fig. R40**). The correlated data and discussions have been added to the revised manuscript.

Fig. R40 | Intrinsic OER activity comparison. The blue, purple, and red columns indicated Ru₇/CoOOH, Ir₇/CoOOH, and Ru₇Ir₇/CoOOH, respectively.

4. There are no obvious potential decays can be observed in Figs. 3f and 3i, please extend the operating times.

Response:

Many thanks for the constructive suggestion from the reviewer. We have extended the operating times to estimate the durability of Ru_TIr_V/CoOOH. As shown in **Fig. R41a**, Ru_TIr_V/CoOOH achieved a lifetime of 500 h at a current density of 10 mA cm⁻². Furthermore, the OER current density of Ru_TIr_V/CoOOH showed no apparent attenuation for over 260 h at a current density of 100 mA cm⁻². The results demonstrated the excellent stability of Ru_TIr_V/CoOOH towards oxygen evolution. To further evaluate the performance of Ru_TIr_V/CoOOH as an OER electrocatalyst for industrial water splitting, we adopted an AEM electrolyzer with the cathodic reaction supported by Pt/C for hydrogen evolution. The polarization curves indicated that the catalyst lowered the cell voltage to 1.9 and 2.3 V at current densities of 0.5 and 1.0 A cm⁻², respectively (**Fig. 3h**). The AEM water electrolyzer was continuously operated over 380 h and 220 h at a current density of 0.5 A cm⁻² and 1.0 A cm⁻², respectively, demonstrating its excellent stability for industrial water splitting (**Fig. R41b**). The correlated data and discussions have been added to the revised manuscript.

Fig. R41 | Stability evaluation of the Ru_TIr_V/CoOOH. a. Chronopotentiometry curves of Ru_TIr_V/CoOOH towards OER at a current density of 10 mA cm⁻² and 100 mA cm⁻². b. Chronopotentiometry test at a current density of 0.5 A cm⁻² and 1.0 A cm⁻² in the AEM water electrolyzer.

5. Please provide the table of catalytic activity of reported electrocatalysts for AEM water electrolyzer to compare with other electrocatalysts. Moreover, we suggest that the author adopt the method of catalyst coated membrane (CCM) to fabricate membrane electrode assembly (MEA), which is expected to reduce the internal resistance of proton coupled electron transfer at the catalyst-membrane interface, improving reactor performance.

Response:

We genuinely thank the reviewer for the valuable suggestion. We have added the comparison of the AEM water electrolyzer performance for Ru_TIr_V/CoOOH with some other reported electrocatalysts to the revised manuscript as **Supplementary Figure 30** and **Supplementary Table 6**. The overpotential at a current density of 1.0 A cm⁻² and stability time of Ru_TIr_V/CoOOH were comparable to those of recently reported catalysts for AEM water electrolyzers (**Fig. R42**).

Fig. R42 | Comparison of cell voltage and stability time at a current density of 1.0 A cm⁻² for recently reported catalysts for AEM water electrolyzers.

The activity and stability of Ru₁Ir_x/CoOOH were also evaluated by assembling the sample into a MEA electrolyzer (**Fig. R43a** and **43b**). The polarization curves showed the Ru₁Ir_x/CoOOH exhibit a cell voltage of 2.3 V at current densities of 1.0 A cm⁻² (**Fig. R43c**). Stability evaluation demonstrated the MEA incorporating Ru₁Ir_x/CoOOH as the anode catalysts operated for over 87 h at a current density of 1.0 A cm⁻², demonstrating its remarkable stability for industrial water splitting (**Fig. R43d**).

Fig. R43 | MEA water electrolyzer test. a. b. Optical image of the MEA water electrolyzer (**a**) and the Ru₁Ir_x/CoOOH coated membrane (**b**). **c.** Polarization curves of Ru₁Ir_x/CoOOH towards OER in the MEA water electrolyzer. **d.** Chronopotentiometry curves of Ru₁Ir_x/CoOOH towards OER at a current density of 1.0 A cm⁻² in the MEA water electrolyzer.

6. For *in-situ* EXAFS spectra of Ru, the coordination numbers of Ru-O increased from 5.0 to 5.4 as the applied voltage increased from OCP to 1.65 V, but the average distance between Ru and O (R) keep unchanged, why? In my opinion, under the OER conditions, the average R is comprised of two different types, which is related with Ru atoms connected with the lattice oxygen and Ru atoms connected with intermediates.

Response:

We would like to express our gratitude to the reviewer for the valuable comments and insightful discussions. We fully agree with the reviewer’s point that the average bond length of Ru-O was comprised of different types, including Ru atoms connected with the lattice oxygen and Ru atoms connected with intermediates as well as the surface dangling bonds. Nevertheless, the Ru-O bond lengths determined by *in-situ* XAFS represent an average value, which cannot assess changes in Ru-O bond lengths of different types. Therefore, we performed theoretical calculations to evaluate the bond lengths of these types of Ru-O bonds under OER conditions. For the Ru-O bonding of Ru atoms connected with the lattice oxygen, the average Ru-O bond lengths for the initial, *OH, *O, and *OOH steps were about 2.18, 2.16, 2.19, and 2.18 Å, respectively (Fig. R44). For the Ru-O bonding of Ru atoms connected with intermediates and the surface dangling bonds, the average Ru-O bond length for the initial, *OH, *O, and *OOH steps were about 1.87, 1.85, 1.80, and 1.84 Å, respectively. Thus, the average Ru-O bond lengths for the initial, *OH, *O, and *OOH steps were 2.05, 2.01, 2.00, and 2.01 Å, respectively. The variation in the average bond lengths across each step was minimal, leading to an insignificant alteration of the Ru-O bond length in the *in-situ* XAFS spectra.

Fig. R44 | Evaluation of the average Ru-O bond length at different reaction steps.

7. It is not sufficient to determine the AEM pathway by the evolution of *OOH species from the *in-situ* ATR-SEIRAS, LOM pathway can co-exist in alkaline water splitting, especially for these kinds of substrates [(MOOH or M(OH)_x]. DEMS is recommended.

Response:

Many thanks to the reviewers for giving valuable advice. We conducted *in-situ* ¹⁸O isotope-labeling DEMS experiments on Ru_T/CoOOH, Ir_v/CoOOH, Ir_TRu_v/CoOOH, and Ru_TIr_v/CoOOH to validate their OER mechanism. Two steps of DEMS experiments were designed, utilizing H₂¹⁸O and H₂¹⁶O as the supporting electrolyte (1.0 M KOH). Initially, the Ru_T/CoOOH, Ir_v/CoOOH, Ir_TRu_v/CoOOH, and Ru_TIr_v/CoOOH were labeled with ¹⁸O isotopes by performing cyclic voltammetry. In the second step, the samples were rinsed with abundant water and then operated in the H₂¹⁶O electrolyte. Cyclic voltammetry cycles were carried out

with the simultaneous detection of signals for $^{36}\text{O}_2$ ($^{18}\text{O}^{18}\text{O}$), $^{34}\text{O}_2$ ($^{18}\text{O}^{16}\text{O}$), and $^{32}\text{O}_2$ ($^{16}\text{O}^{16}\text{O}$), respectively. The mass signal for $^{36}\text{O}_2$ ($^{18}\text{O}^{18}\text{O}$) was barely detectable in all samples, precluding the oxygen-oxygen coupling mechanism (OPM) as reaction pathway for all samples (**Fig. R45a-d**). Additionally, a trace amount of $^{34}\text{O}_2$ ($^{18}\text{O}^{16}\text{O}$) in the OER products was detected, which can be attributed to the natural abundance of ^{18}O in deionized water used in the experiments, suggesting that the lattice oxygen-mediated mechanism (LOM) did not occur over those samples. Furthermore, the predominant mass signal corresponding to $^{32}\text{O}_2$ ($^{16}\text{O}^{16}\text{O}$) was observed for all samples, indicating that $\text{Ru}_\text{T}/\text{CoOOH}$, $\text{Ir}_\text{V}/\text{CoOOH}$, $\text{Ir}_\text{T}\text{Ru}_\text{V}/\text{CoOOH}$, and $\text{Ru}_\text{T}\text{Ir}_\text{V}/\text{CoOOH}$ follow the adsorbate evolution mechanism for oxygen evolution. The correlated data and discussions have been added to the revised manuscript.

Fig. R45 | *In-situ* ^{18}O isotope-labeling DEMS measurements. a-d. *In-situ* DEMS signals of $^{36}\text{O}_2$, $^{34}\text{O}_2$, and $^{32}\text{O}_2$ for $\text{Ru}_\text{T}/\text{CoOOH}$ (a), $\text{Ir}_\text{V}/\text{CoOOH}$ (b), $\text{Ir}_\text{T}\text{Ru}_\text{V}/\text{CoOOH}$ (c), and $\text{Ru}_\text{T}\text{Ir}_\text{V}/\text{CoOOH}$ (d).

8. The section about mechanistic studies is insufficient. It is recommended that the authors disclose the effects of site-specific on the OER performance more deeply. For example, the difference of the distortions by the heterogeneous atoms' assembly, which is directly influence the efficiency of charge transfer between active sites and reactants. Please pay more attention to the improvements of intrinsic activity induced by the specific atom assemble.

Response:

We genuinely thank the reviewer for the insightful comment. Firstly, to investigate whether the enhanced OER activity of $\text{Ru}_\text{T}\text{Ir}_\text{V}/\text{CoOOH}$ was attributable to the optimization of the electronic

structure of Ru single atoms, we examined the impact of the anchored Ir single atoms on the electronic structure of Ru single atoms. According to the Ru *K*-edge XANES, the absorption edge of Ru_TIr_V/CoOOH overlapped with that of Ru_T/CoOOH, presenting similar valence states of Ru single atoms. The result indicated the anchoring of Ir single atoms did not alter the electronic structure of Ru single atoms.

Subsequently, we performed a comprehensive analysis of the reaction pathways for Ru_T/CoOOH and Ru_TIr_V/CoOOH using theoretical calculations to explore the synergistic effect between Ru and Ir single atoms. According to the results of *in-situ* ATR-SEIRAS and ¹⁸O isotope-labeling DEMS, the absorbate evolution mechanism with four concerted proton-electron transfer steps was considered for both samples. Based on the *in-situ* XAFS spectra results, the Ru single atoms at the three-fold fcc hollow sites were identified as active sites for adsorption of key intermediates. For Ru_T/CoOOH, the reaction started from the adsorption of OH⁻ on the Ru site, followed by the sequential deprotonation to form *O, O-O bonding formation to generate OOH*, and desorption to produce O₂(g). Remarkably, for Ru_TIr_V/CoOOH, as the reaction proceeds to the *OOH step, the coordinated hydroxide radicals of Ir single atoms formed hydrogen bonding with the *OOH intermediates on the Ru sites. The hydrogen bonding interaction significantly stabilized the *OOH intermediates, reducing the theoretical overpotential by 0.62 V. OER performance evaluation demonstrated that the overpotential required to reach a current density of 10 mA cm⁻² for Ru_TIr_V/CoOOH was 180 mV, which was 40 mV lower than 220 mV for Ru_T/CoOOH. In addition, the hydrogen bonding interaction was further confirmed by the *in-situ* ATR-SEIRAS, where the wavenumber of *OOH species on Ru_T/CoOOH shifted from 1051 cm⁻¹ to 1094 cm⁻¹ on Ru_TIr_V/CoOOH.

To evaluate the intrinsic activity of Ru_T/CoOOH and Ru_TIr_V/CoOOH, the turnover frequency (TOF) was obtained by normalizing the current density against the number of Ru species. The TOF of Ru_TIr_V/CoOOH was calculated to be 247588.7 s⁻¹ at an overpotential of 250 mV, which was 13.0 times that of Ru_T/CoOOH (19087.5 s⁻¹), indicating the Ir single atoms at V_O sites significantly elevated the OER performance of Ru_T/CoOOH (**Fig. R46**). The enhanced OER activity was also reflected in its faster charge transfer efficiency. The charge transfer efficiency of the samples was characterized via electrochemical impedance spectra (EIS) (**Fig. R47**). The smallest semicircle diameter of Ru_TIr_V/CoOOH suggested its fastest charge transfer at the interface, which was beneficial to accelerate OER kinetics.

Fig. R46 | Comparison of turnover frequencies between Ru_T/CoOOH and Ru_TIr_V/CoOOH toward OER at an overpotential of 250 mV.

Fig. R47 | EIS of CoOOH, Ru_T/CoOOH, Ir_V/CoOOH, and Ru_TIr_V/CoOOH.

9. Ru single atoms at the three-fold fcc hollow sites serve as adsorption sites for key reaction intermediates. Meanwhile, Ir single atoms at V_O sites stabilized the *OOH intermediates on the Ru single atoms via hydrogen bonding interactions. But, the current experiments cannot support this point. What is hydrogen bonding and how to confirm this point? What is the effective distance of hydrogen bonding interaction?

Response:

We genuinely thank the reviewer for the insightful questions and valuable suggestions. The hydrogen bonding interaction is a type of intermolecular force other than van der Waals force. It arises from the partial electrostatic attraction between a hydrogen atom and an electronegative atom or group. The effective distance of hydrogen bonding is lower than 3.5 Å when both ligands are oxygen (*Biochemistry*, **48**, 7986-7995 (2009)).

To prove the enhanced OER activity of Ru_TIr_V/CoOOH compared to Ru_T/CoOOH was attributable to the hydrogen bonding interactions R45 rather than the optimization of the Ru single

atoms' electronic structure, we investigated the electronic structure of Ru single atoms after anchoring Ir single atoms. According to the Ru *K*-edge XANES, the absorption edge of Ru_TIr_V/CoOOH overlapped with that of Ru_T/CoOOH, presenting similar valence states of Ru single atoms. The similar Ru valence state indicated the anchoring of Ir single atoms did not alter the electronic structure of Ru single atoms.

Subsequently, we performed *in-situ* ATR-SEIRAS and theoretical calculations to confirm the existence of hydrogen bonding interactions and the possible effect on the adsorption of intermediates. As shown in **Fig. 4g**, with the applied potential on Ru_T/CoOOH increased from OCP to 1.65 V, an absorption band at about 1051 cm⁻¹ showed a potential-dependent behavior, which can be assigned to the adsorption of *OOH species. By comparison, the wavenumber of *OOH species on Ru_TIr_V/CoOOH was located at about 1094 cm⁻¹ (**Fig. 4h**). The blue shift of the *OOH characteristic peak in Ru_TIr_V/CoOOH indicated an enhanced O-O bond vibration, which results from the weaker adsorption of *OOH intermediates modulated by additional forces. Furthermore, the configuration of the reaction intermediates was analyzed by theoretical calculations to investigate the origin of the weaker adsorption of *OOH intermediates. As the reaction proceeds to the *OOH step, the distance between *OOH intermediates (H atom) on Ru single atom and coordinated hydroxide radicals of Ir single atom (O atom) was 2.37 Å, indicating the presence of hydrogen bonding interaction. The hydrogen bonding interaction significantly stabilized the *OOH intermediates, reducing the theoretical overpotential by 0.62 V, thereby enhancing the OER performance of Ru_TIr_V/CoOOH.

Reviewer #4 (Remarks to the Author):

In this article, the authors introduced a precision synthesis strategy for constructing heterogeneous single-atom catalysts. They proposed that the synergy effect in heterogeneous single atoms was strongly related to the single-atom anchoring sites. The systematic electrochemical evaluation demonstrated that the Ru_TIr_V/CoOOH with Ru single atoms at three-fold fcc hollow sites and Ir single atoms at oxygen vacancies exhibited superior OER performance compared to Ir_TRu_V/CoOOH. Overall experiments including a series of *in-situ* spectroscopic evidence and mechanistic studies further proved the intrinsic mechanism of the site-specific synergy in heterogeneous single atoms. In our personally opinion, the work is interesting and proposes a new insight into the rational design of highly efficient catalysts applied in OER. We think this manuscript is very worthy of publication in Nature Communications after some revisions. The following are our questions

Response:

We sincerely thank the reviewer for the positive comments and valuable suggestions on our manuscript, which certainly helped to improve its quality. We have revised the manuscript based on the suggestions. Specifically, we discussed the intrinsic mechanism of the selective combination between single atoms and anchoring sites. Then, we investigated the origins for the site-specific interactions of Ru and Ir single atoms with Co species by analyzing structural models of Ru_TIr_V/CoOOH. In addition, the OER mechanism of Ir_TRu_V/CoOOH was also be

investigated to elucidate the intrinsic reason for the site-specific synergy in heterogeneous Ru and Ir single atoms. Moreover, we conducted *in-situ* XAFS experiments to explore the changes in the electronic structure of Co species in Ru_TIr_V/CoOOH during OER. Additionally, we adopted ICP-MS to probe the dissolved Ru and Ir species in the electrolyte during the stability tests and characterized the structure of Ir_TRu_V/CoOOH after OER. The OER performance of Ir_T/CoOOH was also evaluated and the Tafel slopes of Ir_T/CoOOH and Ir_TRu_V/CoOOH were compared with Ru_TIr_V/CoOOH. We hope the revised manuscript will alleviate any concerns.

1. The authors proposed that the Ru and Ir single atoms were selectively anchored at the three-fold fcc hollow sites and oxygen vacancies, respectively. Please discuss the intrinsic mechanism of the selective combination between single atoms and anchoring sites.

Response:

We sincerely thank the reviewer for the valuable comment. The intrinsic mechanism of the selective combination between single atoms and anchoring sites was described as follows: the multiple topologies or defects on the surface of CoOOH provided us with diverse single-atom anchoring sites, including negatively charged three-fold fcc hollow sites of oxygen and positively charged V_O sites. As a result of the different electronegativities between single-atom precursors and anchoring sites, the single-atom precursors could selectively combine with the specific sites. In particular, a wet-chemical synthesis strategy was employed to fabricate Ru single atoms. During the fabrication process, the positively charged Ru^{3+} ions in the solution were selectively anchored onto negatively charged three-fold fcc hollow sites through electrostatic adsorption. In contrast, the Ir single atoms were anchored via an electrochemical deposition method. During synthesis, the negatively charged $Ir(OH)_6^{2-}$ ions in the electrolyte were selectively anchored onto the positively charged V_O sites by electrostatic adsorption. Therefore, based on the mechanism of selective combination between the single-atom precursors and anchoring sites, heterogeneous Ru and Ir single atoms can be selectively anchored onto specific sites. The correlated discussions have been added to the revised manuscript.

2. Multiple characterizations including XAFS, XPS, and XAS demonstrated that the Ru single atoms at three-fold fcc hollow sites exhibited stronger interactions with Co species than Ir single atoms at oxygen vacancies. Please clearly origins of the site-specific interactions.

Response:

We sincerely thank this reviewer for the insightful comment. According to the EXAFS fitting results, Ru single atoms at three-fold fcc hollow sites were stabilized by three oxygen atoms of the sites, while the remaining two coordinated oxygen atoms were suspended at its surface as dangling bonds. For Ir single atoms at V_O sites, one apex oxygen of the IrO_6 octahedral structure was inserted into the V_O sites, while four side OH^- of the octahedra formed hydrogen bonding with adjacent oxygen atoms on the CoOOH surface to stabilize the structure. Thus, the Ru and Ir single atoms at diverse sites interacted with Co species through oxygen atoms. Specifically, the Ru single atoms at the three-fold fcc hollow sites interacted with Co atoms through three O

atoms of CoOOH. The Ir single atoms at the V_O sites interacted with Co atoms through one O atom. The increased number of connected O atoms results in stronger interactions between Ru single atoms and Co species compared to those between Ir single atoms and Co species.

3. Through a series of *in-situ* experiments and theoretical calculations, the authors have well explained the intrinsic mechanism for the improved performance of $Ru_TIr_V/CoOOH$ compared to $Ru_T/CoOOH$. However, the reason for the weakened synergy in Ir single atoms at the three-fold fcc hollow sites and Ru single atoms at the oxygen vacancies was less clearly explained. It is recommended to discuss the attenuated synergy in detail.

Response:

We genuinely thank the reviewer for the insightful questions and valuable suggestions. Based on the characterization and theoretical calculation results presented in the manuscript, the enhanced performance of $Ru_TIr_V/CoOOH$ compared to $Ru_T/CoOOH$ can be attributed to the interactions between Ir single atoms at the V_O sites and the *OOH intermediates adsorbed on the Ru single atoms at the three-fold fcc hollow sites. Specifically, as the reaction proceeds to the *OOH step, the coordinated hydroxide radicals of Ir single atoms formed hydrogen bonding with the *OOH intermediates, thus significantly stabilized the *OOH intermediates. The hydrogen bonding interaction caused a decrease in the theoretical overpotential of $Ru_TIr_V/CoOOH$ by 0.62 V compared to $Ru_T/CoOOH$. Moreover, the presence of hydrogen bonding interactions was further confirmed by *in-situ* ATR-SEIRAS, where the wavenumber of *OOH species on $Ru_T/CoOOH$ shifted from 1051 cm^{-1} to 1094 cm^{-1} on $Ru_TIr_V/CoOOH$.

To investigate the weakened synergy between Ir single atoms at the three-fold fcc hollow sites and Ru single atoms at the V_O sites, *in-situ* ATR-SEIRAS was performed on $Ir_T/CoOOH$ and $Ir_TRu_V/CoOOH$. The results showed the wavenumber of *OOH on $Ir_T/CoOOH$ and $Ir_TRu_V/CoOOH$ were positioned at about 1040 cm^{-1} and 1038 cm^{-1} , respectively (**Fig. R48a and 48b**). The close wavenumber suggested that the Ru single atoms at the V_O sites did not significantly influence the adsorption of *OOH intermediates on the Ir single atoms at the three-fold fcc hollow sites. Therefore, the synergistic effect in the heterogeneous Ru and Ir single atoms on $Ir_TRu_V/CoOOH$ was weaker compared to that on $Ru_TIr_V/CoOOH$. In addition, the weakened synergy was also reflected by the OER performance of $Ir_T/CoOOH$ and $Ir_TRu_V/CoOOH$ (**Fig. R49**). The polarization curves showed that the overpotential required to reach a current density of 10 mA cm^{-2} for $Ir_T/CoOOH$ and $Ir_TRu_V/CoOOH$ was 280 and 270 mV, respectively, reflecting a minimal improvement in OER performance. The correlated data and discussions have been added to the revised manuscript.

Fig. R48 | In-situ spectroscopic analysis. a. b. In-situ ATR-SEIRAS of Ir_T/CoOOH (a) and Ir_TRu_V/CoOOH (b).

Fig. R49 | Electrocatalytic performance of Ir_T/CoOOH and Ir_TRu_V/CoOOH towards oxygen evolution. Polarization curves of Ir_T/CoOOH and Ir_TRu_V/CoOOH towards oxygen evolution in 1.0 M KOH electrolyte. The inset figure is the overpotentials of Ir_T/CoOOH and Ir_TRu_V/CoOOH at a current density of 10 mA cm⁻².

4. The working electrode is not represented in the optical image of the *in-situ* ATR-SEIRAS experiment (Supplementary Fig. 22b). This optical image should be re-shooting.

Response:

We honestly thank the reviewer for raising this important issue and apologize for the negligence here. In the original optical image of the *in-situ* ATR-SEIRAS measurements, the working electrode was positioned at the front of the cell and was not captured in the images. The optical image of the *in-situ* ATR-SEIRAS experiment was re-shot and added to the revised manuscript (Fig. R50).

Fig. R50 | *In-situ* spectroscopic characterizations. Optical image of the *in-situ* ATR-SEIRAS measurements.

5. *In-situ* Raman spectra of Ru_TIrv/CoOOH were performed to explore the potential structure evolution of the CoOOH under oxygen evolution conditions. It is recommended the author conduct characterizations such as *in-situ* XAFS to discover the potential evolution in the valence state of Co species.

Response:

We sincerely thank the reviewer for the valuable comment. *In-situ* XAFS measurements were conducted to investigate the potential evolution of Co species in Ru_TIrv/CoOOH under OER conditions (**Fig. R51a** and **51b**). The Co *K*-edge XANES spectra showed negligible shifts in the absorption edge from OCP to 1.65 V, indicating that the valence states of the Co species only slightly increased under oxidation potentials (**Fig. R51c**). The EXAFS spectra of *in-situ* Co *K*-edge showed two prominent peaks at about 1.4 and 2.4 Å, which were attributed to Co-O and Co-Co bonding, respectively (**Fig. R51d**). The characteristic peaks of Co-O and Co-Co coordination in the EXAFS showed insignificant changes with increasing voltage, indicating the stable structure of Ru_TIrv/CoOOH during OER. The correlated data and discussions have been added to the revised manuscript.

Fig. R51 | *In-situ* spectroscopic characterizations. **a.** Schematic diagram of the *in-situ* XAFS electrolytic cell. **b.** Optical image of the *in-situ* Co K-edge XAFS tests. **c. d.** *In-situ* Co K-edge XANES (**c**) and EXAFS (**d**) spectra of Ru_TIr_V/CoOOH under applied potentials ranging from OCP to 1.65 V.

6. Whether Ru and Ir single atoms on the Ru_TIr_V/CoOOH surface dissolved into the electrolyte during the OER process.

Response:

We genuinely appreciate the insightful comment provided by the reviewer. We have quantified the content of dissolved Ru and Ir species for Ru_TIr_V/CoOOH using ICP-MS during the stability test at a current density of 10 mA cm⁻². As shown in **Fig. R52**, merely 5.9 and 7.0 wt% of Ru and Ir species were dissolved during 500 h stability test. The insignificant dissolution of single atoms indicated the excellent stability of Ru_TIr_V/CoOOH towards OER. The correlated data and discussions have been added to the revised manuscript.

Fig. R52 | The dissolved percentage of Ru and Ir species during the stability test. The dissolved Ru and Ir species were detected by ICP-MS during the stability test.

7. The structure characterizations of Ru_TIr_V/CoOOH after OER should be also added to the manuscript, such as XRD pattern, TEM image, and HAADF-STEM image.

Response:

We genuinely thank this reviewer for the valuable suggestions. The morphology and structure of Ru_TIr_V/CoOOH after OER were characterized. TEM image showed that Ru_TIr_V/CoOOH preserved the nanosheet morphology after OER (**Fig. R53a**). The XRD pattern of Ru_TIr_V/CoOOH after OER was attributed to the CoOOH (PDF #26-1107), no extra phase was identified in the XRD pattern compared with that before OER (**Fig. R53b**). In the HAADF-STEM image, the atomic dispersion of Ru and Ir single atoms were preserved after the test (**Fig. R53c**). EDX elemental mapping images still showed the uniform metal distribution of Ru and Ir elements across the Ru_TIr_V/CoOOH (**Fig. R53d**). The above results demonstrated the excellent stability of Ru_TIr_V/CoOOH for oxygen evolution. The correlated data and discussions have been added to the revised manuscript.

Fig. R53 | Morphology and structure characterizations of Ru_TIr_V/CoOOH after OER. a. TEM image. **b.** XRD pattern. **c.** HAADF-STEM image. **d.** EDX elemental mapping.

8. The oxygen evolution performance of Ir_T/CoOOH should be evaluated and added to the manuscript. In addition, the Tafel slopes of Ir_T/CoOOH and Ir_TRu_V/CoOOH should be compared with Ru_TIr_V/CoOOH.

Response:

We sincerely thank the reviewer for the insightful comments. The OER performance of Ir_T/CoOOH was evaluated in a standard three-electrode system in 1.0 M KOH electrolyte. According to the polarization curve, the overpotential required to reach a current density of 10 mA cm⁻² for Ir_T/CoOOH and Ir_TRu_V/CoOOH was 280 and 270 mV, which was 100 and 90 mV higher than that of Ru_TIr_V/CoOOH, respectively (**Fig. R54a** and **54b**). In addition, the Tafel slopes of Ir_T/CoOOH and Ir_TRu_V/CoOOH were also assessed. Ir_T/CoOOH and Ir_TRu_V/CoOOH exhibited Tafel slope values of 67 and 64 mV dec⁻¹, which was 26 and 23 mV dec⁻¹ higher than that of Ru_TIr_V/CoOOH, respectively (**Fig. R54c**). The lowest overpotential and Tafel slope for Ru_TIr_V/CoOOH indicated its fastest kinetics among the catalysts. The correlated data and discussions have been added to the revised manuscript.

Fig. R54 | Electrocatalytic performance towards oxygen evolution. a. Polarization curves of Ir_T/CoOOH, Ir_TRu_v/CoOOH, and Ru_vIr_T/CoOOH towards oxygen evolution in 1.0 M KOH electrolyte. **b. c.** Overpotentials at a current density of 10 mA cm⁻² (**b**) and Tafel slopes (**c**) of Ir_T/CoOOH, Ir_TRu_v/CoOOH, and Ru_vIr_T/CoOOH.